# THINK WHILE YOU GENERATE: DISCRETE DIFFUSION WITH PLANNED DENOISING

**Sulin Liu**[1*]**, Juno Nam**[1]**, Andrew Campbell**[2]**, Hannes Stärk**[1]**, Yilun Xu**[3]**,
Tommi Jaakkola**[1]**, Rafael Gómez-Bombarelli**[1]
[1]Massachusetts Institute of Technology,  [2]University of Oxford,  [3]NVIDIA Research

## ABSTRACT

Discrete diffusion has achieved state-of-the-art performance, outperforming or approaching autoregressive models on standard benchmarks. In this work, we introduce *Discrete Diffusion with Planned Denoising* (DDPD), a novel framework that separates the generation process into two models: a planner and a denoiser. At inference time, the planner selects which positions to denoise next by identifying the most corrupted positions in need of denoising, including both initially corrupted and those requiring additional refinement. This plan-and-denoise approach enables more efficient reconstruction during generation by iteratively identifying and denoising corruptions in the optimal order. DDPD outperforms traditional denoiser-only mask diffusion methods, achieving superior results on language modeling benchmarks such as `text8`, `OpenWebText`, and token-based generation on `ImageNet` $256 \times 256$. Notably, in language modeling, DDPD significantly reduces the performance gap between diffusion-based and autoregressive methods in terms of generative perplexity. Code is available at [github.com/liusulin/DDPD](github.com/liusulin/DDPD).

## 1 INTRODUCTION

Generative modeling of discrete data has recently seen significant advances across various applications, including text generation [5], biological sequence modeling [25, 4], and image synthesis [8, 9]. Autoregressive transformer models have excelled in language modeling but are limited to sequential sampling, with performance degrading without annealing techniques such as nucleus (top-p) sampling [20]. In contrast, diffusion models offer more flexible and controllable generation, proving to be more effective for tasks that lack natural sequential orderings, such as biological sequence modeling [25, 1] and image token generation [8]. In language modeling, the performance gap between discrete diffusion and autoregressive models has further narrowed recently, thanks to improved training strategies [29, 36, 34, 13], however, a gap still remains on some tasks [36, 29].

State-of-the-art discrete diffusion methods train a denoiser (or score) model that determines the transition rate (or velocity) from the current state to predicted values. During inference, the generative process is discretized into a finite number of steps. At each step, the state values are updated based on the transition probability, which is obtained by integrating the transition rate over the timestep period.

In order to further close the performance gap with autoregressive models, we advocate for a rethinking of the standard discrete diffusion design methodology. We propose a new framework **Discrete Diffusion with Planned Denoising** (DDPD), that divides the generative process into two key components: a planner and a denoiser, facilitating a more adaptive and efficient sampling procedure. The process starts with a sequence of tokens initialized with random values. At each timestep, the planner model examines the sequence to identify the position most likely to be corrupted and in need of denoising. The denoiser then predicts the value for the selected position, based on the current noisy sequence. The key insight behind our plan-and-denoise approach is that the generative probability at each position can be factorized into two components: 1) **planning**: the probability that a position is corrupted, and 2) **denoising**: the probability of denoising it according to the data distribution.

Our framework offers two primary advantages:

---
*Corresponding to `sulinliu@mit.edu`.

1. **Simplified Learning:** By decomposing the generative task into planning and denoising, each sub-task becomes easier to learn. Planning, in particular, is significantly less complex than denoising. Traditional uniform diffusion models, in contrast, require a single neural network to handle both tasks simultaneously, which can lead to inefficiencies.

2. **Improved Sampling Algorithm:** The plan-and-denoise framework enables an adaptive sampling procedure that dynamically adjusts based on the planner's predictions.

As shown in Fig. 1, this sampling process provides enhanced robustness: **1. Adaptive Time Discretization:** When the planner identifies that the sequence is noisier than expected at a given timestep, it increases the number of denoising moves within the remaining time to ensure thorough refinement. **2. Error Correction:** The planner can identify errors introduced in earlier steps. If necessary, it adjusts time backward, allowing the denoiser to revisit and correct mistakes. This ensures all corrupted tokens are reconstructed accurately, yielding higher-quality generations.

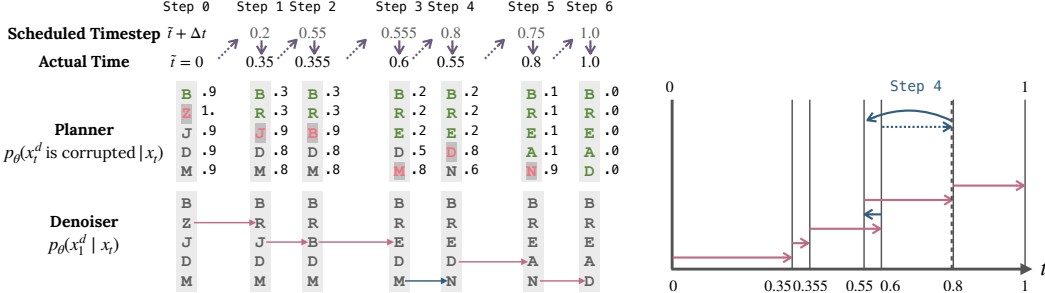

Figure 1: An example generation trajectory from $t = 0$ to $1$ of a 5-letter sequence. At each step, the *planner* estimates the probability of token corruption (as indicated by the numbers), selects a position, and the *denoiser* predicts the token. The *actual time* progression — adaptively re-calibrated by the planner's noise level assessment — may deviate from the scheduled timestep. For instance, in step 2, minimal improvement results in slower time progression, while in step 4, an error causes a backward step in time. Sampling continues until all corrupted tokens are reconstructed.

Our contributions are as follows:

- We introduce Discrete Diffusion with Planning and Denoising (DDPD), a novel framework for discrete generative modeling that decomposes the generation process into planning and denoising.

- Our proposed plan-and-denoise framework introduces an adaptive sampling scheme, guided by the planner's output, enabling continuous self-correction of errors in an optimal order. This results in improved generation by scaling the number of sampling steps.

- We derive simple and effective training objectives for the planner and denoiser models, grounded in maximizing the Evidence Lower Bound (ELBO) for discrete diffusion processes.

- In experiments on GPT-2 scale language modeling and $256 \times 256$ image token generation, DDPD significantly outperforms its mask diffusion counterparts when using the same denoiser. Furthermore, we demonstrate that incorporating a planner substantially enhances generation quality, even when using a smaller or weaker denoiser model compared to baseline methods.

## 2 PRELIMINARIES

We begin by introducing the problem setup and notations. Following [7, 13], we then explain the Continuous Time Markov Chain (CTMC) framework [31], which is used to define the forward and reverse processes of discrete diffusion, and how the discrete timestep version is derived from it.

**Setup and Notations.** We aim to model discrete data where a sequence $x \in \{1, \cdots, S\}^D$ is $D$-dimensional and each element $x^d$ takes $S$ possible states. $x^{\backslash d}$ denotes all dimensions except $d$. For clarity in presentation, we assume $D = 1$ and all results hold for $D > 1$ (see Section 3.1). We use $p(x)$ to denote the probability mass function (PMF). The $\delta\{i, j\}$ is the Kronecker delta function, which is 1 when $i = j$ and 0 otherwise.

**Continuous Time Markov Chains and Discretization.** We adopt the CTMC framework to define the discrete diffusion process. A realization of the CTMC dynamics is defined by a trajectory $x_t$ over time $t \in [0, 1]$ that makes jumps to another state after a random waiting period known as the holding time. The transition rates and the next states are governed by the rate matrix $R_t \in \mathbb{R}^{S \times S}$ (analogous to the velocity field $\nu_t$ in continuous state spaces), where the off-diagonal elements, representing jumps to different states, are non-negative. For an infinitesimal timestep $\mathrm{d}t$, the probability of transitioning from $x_t$ to a different state $j$ is given by $R_t(x_t, j)\mathrm{d}t$.

In practice, the trajectory is simulated using finite time intervals $\Delta t$. As a result, the transition probability follows a categorical distribution with the PMF [39]:

$$p_{t+\Delta t|t}(j|x_t) = \delta\{x_t, j\} + R_t(x_t, j)\Delta t, \tag{1}$$

where we denote this as $\mathrm{Cat}(\delta\{x_t, j\} + R_t(x_t, j)\Delta t)$, and $R_t(x_t, x_t) := -\sum_{s \neq x_t} R_t(x_t, s)$ to ensure that the transition probabilities sum to 1.

**Forward Corruption Process.** Following [7, 34, 13], which were inspired by flow matching in continuous state space [26, 27, 2], we construct the forward process by interpolating from noise $p_0(x_0) = p_{\mathrm{noise}}(x_0)$ to clean data $p_1(x_1) = p_{\mathrm{data}}(x_1)$. Common choices for the noise distribution include: (i) $p_{\mathrm{noise}}^{\mathrm{unif}}(x_t) = 1/S$, a uniform distribution over $\{1, \cdots, S\}$; and (ii) $p_{\mathrm{noise}}^{\mathrm{mask}}(x_t) = \delta\{\mathbb{M}, x_t\}$, a delta PMF concentrated on an artificially introduced mask state $\mathbb{M}$. Let $\alpha_t$ be the noise schedule that introduces noise to the data over time $t$. For example, a linear schedule is given by $\alpha_t = t$. The conditional marginal $p_{t|1}(x_t|x_1)$ is given in closed form:

$$p_{t|1}^{\mathrm{unif}}(x_t|x_1) = \mathrm{Cat}(\alpha_t\delta\{x_1, x_t\} + (1 - \alpha_t)\frac{1}{S}), \tag{2}$$

$$p_{t|1}^{\mathrm{mask}}(x_t|x_1) = \mathrm{Cat}(\alpha_t\delta\{x_1, x_t\} + (1 - \alpha_t)\delta\{\mathbb{M}, x_t\}). \tag{3}$$

At $t = 1$, the conditional marginal converges to the datapoint $x_1$, i.e. $\delta\{x_1, x_t\}$. At $t = 0$, the conditional marginal converges to the noise distribution $p_{\mathrm{noise}}(x_t)$.

**Reverse Generation Process.** Sampling from $p_{\mathrm{data}}$ is achieved by learning a generative rate matrix $R_t(x_t, j)$ to reverse simulate the process from $t = 0$ to $t = 1$ using Eq. (1), such that we begin with samples of $p_{\mathrm{noise}}$ and end with samples of $p_{\mathrm{data}}$. The datapoint conditional reverse rate $R_t(x_t, j|x_1)$ for $j \neq x_t$[1] under the uniform or mask noise distributions is given by [7, 13]:

$$R_t^{\mathrm{unif}}(x_t, j|x_1) = \frac{\dot{\alpha}_t}{1 - \alpha_t}\delta\{x_1, j\}(1 - \delta\{x_1, x_t\}), \quad R_t^{\mathrm{mask}}(x_t, j|x_1) = \frac{\dot{\alpha}_t}{1 - \alpha_t}\delta\{x_1, j\}\delta\{x_t, \mathbb{M}\}.$$

[7, 13] show that the rate we aim to learn for generating $p_{\mathrm{data}}$ is the expectation of the data-conditional rate, taken over the denoising distribution, i.e., $R_t(x_t, j) := \mathbb{E}_{p_{1|t}(x_1|x_t)}[R_t(x_t, j|x_1)]$:

$$R_t^{\mathrm{unif}}(x_t, j) = \frac{\dot{\alpha}_t}{1 - \alpha_t}p_{1|t}(x_1 = j|x_t), \quad R_t^{\mathrm{mask}}(x_t, j) = \frac{\dot{\alpha}_t}{1 - \alpha_t}\delta\{x_t, \mathbb{M}\}p_{1|t}(x_1 = j|x_t). \tag{4}$$

The goal is to approximate this rate using a neural network. During inference, the generative process is simulated with the learned rate by taking finite timesteps, as described in Eq. (1).

## 3 METHOD

### 3.1 DECOMPOSING GENERATION INTO PLANNING AND DENOISING

Recent state-of-the-art discrete diffusion methods [39, 29, 7, 36, 34, 13] have converged on parameterizing the generative rate using a denoising neural network and deriving cross-entropy-based training objectives. This enables simplified and effective training, leading to better and SOTA performance in discrete generative modeling compared to earlier approaches [3, 6]. In Table 1, we summarize the commonalities and differences in the design choices across various methods.

Based on the optimal generative rate in Eq. (4), we propose a new approach to parameterizing the generative rate by dividing it into two distinct components: planning and denoising. We begin

---

[1]For simplicity, we only derive the rates for $j \neq x_t$. The rate for $j = x_t$ can be computed as $R_t(x_t, x_t|x_1) := -\sum_{s \neq x_t} R_t(x_t, s|x_1)$.

Table 1: Specific design choices employed by different discrete diffusion models.

| | SDDM [39] | SEDD [29] | DFM [7] / Discrete FM [13] | MDLM [34] / MD4 [36] | Ours ('DDPD') |
|---|---|---|---|---|---|
| **Sampling (Section 3.2)** | | | | | |
| Method | Tau-leaping | Tau-leaping | Tau-leaping | Tau-leaping | Adaptive Gillespie |
| Time steps $t_i$ | $i/N$ | $i/N$ | $i/N$ | $i/N$ | $t_\theta(x_i) + \tau_i,$ (†) $\tau_i \sim \mathrm{Exp}\left(\lambda_\theta(x_i)\right)$ |
| Noise schedule $\alpha_t$ | $\alpha_t$ | $\alpha_t$ (‡) | $\alpha_t$ (*) | $\alpha_t$ | |
| **Generative rate parameterization (Section 3.1)** | | | | | |
| Masking | – | $\frac{\dot{\alpha}_t}{\alpha_t}\delta\left\{x_t^d, M\right\}\cdot$ $s_\theta\left(x_t\right)_{x_t^d \to j}$ | $\dot{\alpha}_t \delta\{x_t^d, M\}/{1-\alpha_t}\cdot$ $p_\theta\left(x_1^d = j\vert x_t\right)$ | $\dot{\alpha}_t\delta\{x_t^d,M\}/{1-\alpha_t}\cdot$ $p_\theta\left(x_1^d = j\vert x_t\right)$ | – |
| Uniform | $\frac{\dot{\alpha}_t}{S\alpha_t}\cdot$ $\frac{p_\theta\left(x_t^d=j\vert x_t^{\backslash d}\right)}{p_\theta\left(x_t^d=x_t^d\vert x_t^{\backslash d}\right)}$ | $\frac{\dot{\alpha}_t}{S\alpha_t}\cdot$ $s_\theta\left(x_t\right)_{x_t^d \to j}$ | $\dot{\alpha}_t/{1-\alpha_t}\cdot$ $p_\theta\left(x_1^d = j\vert x_t\right)$ | – | $\dot{\alpha}_t/{1-\alpha_t}p_\theta(z_t^d = N\vert x_t)\cdot$ $p_\theta\left(x_1^d = j\vert x_t, z_t^d = N\right)$ |

‡ DFM assumes a linear schedule. * MD4 also supports learnable schedule of $\alpha_t$.
† $\lambda_\theta(x_i)$ is the total rate of jump determined by the planner.

by examining how the generative rate in mask diffusion can be interpreted within our framework, followed by a derivation of the decomposition for uniform diffusion.

**Mask Diffusion.** For mask diffusion, the planning part is assigning probability of $\frac{\dot{\alpha}_t}{1-\alpha_t}\delta\left\{x_t, \mathbb{M}\right\}\Delta t$ for the data to be denoised with an actual value. $\delta\left\{x_t, \mathbb{M}\right\}$ tells if the data is noisy ($\mathbb{M}$) or clean. $\frac{\dot{\alpha}_t}{1-\alpha_t}$ is the rate of denoising, which is determined by the remaining time according to the noise schedule. The denoiser $p_{1\vert t}$ assigns probabilities to the possible values to be filled in if this transition happens.

$$R_t^{\mathrm{mask}}(x_t, j)\Delta t = \underbrace{\frac{\dot{\alpha}_t}{1-\alpha_t}\delta\left\{x_t, \mathbb{M}\right\}\Delta t}_{\text{rate of making correction}}\underbrace{p_{1\vert t}\left(x_1 = j\vert x_t\right)}_{\text{prob. of denoising}} \tag{5}$$

**Uniform Diffusion.** Similarly, we would want to decompose the transition probability into two parts: the planning probability based on if the data is corrupted and the denoising probability that determines which value to change to. But in contrast to the mask diffusion case, the noise/clean state of the data is not given to us during generation. We use $z_t^d \in \{N, D\}$ as a latent variable to denote if a dimension is corrupted, with $N$ denoting noise and $D$ denoting data.

From Bayes rule, for $j \neq x_t$, since $p_{1\vert t}\left(x_1 = j\vert x_t, z_t = D\right) = \frac{p(x_t\vert x_1=j, z_t=D)p(x_1=j)}{p(x_t\vert z_t=D)} = 0$

$$p_{1\vert t}\left(x_1 = j\vert x_t\right) = \sum_{z_t \in \{N, D\}} p(z_t\vert x_t)p_{1\vert t}(x_1 = j\vert x_t, z_t) = \underbrace{p(z_t = N\vert x_t)}_{\text{prob. of being corrupted}}\underbrace{p_{1\vert t}(x_1 = j\vert x_t, z_t = N)}_{\text{prob. of denoising}} \tag{6}$$

The first part of the decomposition is the posterior probability of $x_t$ being corrupted, and the second part gives the denoising probability to recover the value of $x_t$ if $x_t$ is corrupted. Plugging Eq. (6) into Eq. (4), we arrive at:

$$R_t^{\mathrm{unif}}(x_t, j)\Delta t = \underbrace{\frac{\dot{\alpha}_t}{1-\alpha_t}p(z_t = N\vert x_t)\Delta t}_{\text{rate. of making correction}}\underbrace{p_{1\vert t}(x_1 = j\vert x_t, z_t = N)}_{\text{prob. of denoising}} \tag{7}$$

By comparing Eq. (7) and Eq. (5), we find that they share the same constant part $\frac{\dot{\alpha}_t}{1-\alpha_t}$ which represents the rate determined by the current time left according to the noise schedule. The main difference is in the middle part that represents the probability of $x_t$ being corrupted. In mask diffusion case, this can be readily read out from the $\mathbb{M}$ token. But in the uniform diffusion case, we need to compute/approximate this probability instead. The last part is the denoising probability conditioned on $x_t$ being corrupted, which again is shared by both and needs to be computed/approximated.

**Generative Rate for Multi-Dimensions.** The above mentioned decomposition extends to $D > 1$. We have the following reverse generative rate [7, 13] for mask diffusion:

$$R_t^{\mathrm{mask}}(x_t, j^d)\Delta t = \frac{\dot{\alpha}_t}{1-\alpha_t}\Delta t\,\delta\left\{x_t^d, \mathbb{M}\right\}p_{1\vert t}\left(x_1^d = j^d\vert x_t\right), \quad \forall j^d \neq x_t^d, \tag{8}$$

and we derive the following decomposition result for uniform diffusion (proof in Appendix A.1):

**Proposition 3.1.** *The reverse generative rate at $d$-th dimension can be decomposed into the product of recovery rate, probability of corruption and probability of denoising:*

$$R_t^{\text{unif}}\left(x_t, j^d\right) \Delta t = \frac{\dot{\alpha}_t}{1-\alpha_t} p_{1|t}\left(x_1^d = j^d | x_t\right) \Delta t \tag{9}$$

$$= \underbrace{\frac{\dot{\alpha}_t}{1-\alpha_t}}_{\text{noise removal rate}} \underbrace{p\left(z_t^d = N | x_t\right)}_{\text{prob. of corruption}} \underbrace{p_{1|t}\left(x_1^d = j^d | x_t, z_t^d = N\right)}_{\text{prob. of denoising}} \Delta t, \quad \forall j^d \neq x_t^d$$

$$\text{where} \quad p\left(z_t^d = N | x_t\right) = 1 - p_{1|t}\left(x_1^d = x_t^d | x_t\right) \frac{\alpha_t}{\alpha_t + (1-\alpha_t)/S} \tag{10}$$

$$p_{1|t}\left(x_1^d = j^d | x_t, z_t^d = N\right) = \frac{p_{1|t}\left(x_1^d = j^d | x_t\right)}{p\left(z_t^d = N | x_t\right)} \tag{11}$$

We observe that the term $p\left(z_t^d = N | x_t\right)$ determines how different dimensions are reconstructed at different rates, based on how likely the dimension is clean or noise given the current context.

**Previous Parameterization.** In the case of mask diffusion, as studied in recent works [29, 7, 36, 34, 13], the most effective parameterization for learning is to directly model the denoising probability with a neural network, as this is the only component that needs to be approximated.

In the case of uniform diffusion, the conventional approach uses a single model to approximate the generative rate as a whole, by modeling the posterior $p_{1|t}\left(x_1^d = j^d | x_t\right)$ as shown in Eq. (9). However, despite its theoretically greater flexibility – allowing token values to be corrected throughout sampling, akin to the original diffusion process in the continuous domain – its performance has not always outperformed mask diffusion, particularly in tasks like image or language modeling.

**Plan-and-Denoise Parameterization.** Based on the observation made in Proposition 3.1, we take the view that generation should consist of two models: a planner model for deciding which position to denoise and a denoiser model for making the denoising prediction for a selected position.

$$R_{t,\text{jump}}^{\text{unif}}\left(x_t, j^d\right) \Delta t = \underbrace{\frac{\dot{\alpha}_t}{1-\alpha_t}}_{\text{noise removal rate}} \Delta t \underbrace{p_\theta\left(z_t^d = N | x_t\right)}_{\text{planner}} \underbrace{p_{1|t}^\theta\left(x_1^d = j^d | x_t, z_t^d = N\right)}_{\text{denoiser}}, \quad \forall x_t \neq j^d \tag{12}$$

This allows us to utilize the planner's output to design an improved sampling algorithm that optimally identifies and corrects errors in the sequence in the most effective denoising order. Additionally, the task decomposition enables separate training of the planner and denoiser, simplifying the learning process for each neural network. Often, a pretrained denoiser is already available, allowing for computational savings by only training the planner, which is generally faster and easier to train.

**Remark 3.2.** *Under this perspective, masked diffusion (Eq. (8)) can be interpreted as a denoiser-only modeling paradigm with a fixed planner, i.e., $p_\theta\left(z_t^d = N | x_t\right) = \delta\left\{x_t^d, \mathbb{M}\right\}$, which assumes that mask tokens represent noise while actual tokens represent clean data. This planner is optimal under the assumption of a perfect denoiser, which rarely holds in practice. When the denoiser makes errors, this approach does not provide a mechanism for correcting those mistakes.*

Next, we demonstrate how our plan-and-denoise framework enables an improved sampling algorithm that effectively leverages the planner's predictions. From this point forward, we assume uniform diffusion by default and use $R_{t,\text{jump}}$ to denote the reverse jump rate, unless explicitly stated otherwise.

## 3.2 SAMPLING

**Prior Works: Tau-leaping Sampler.** The reverse generative process is a CTMC that consists of a sequence of jumps from $t = 0$ to $t = 1$. The most common way [6, 39, 29, 36, 34, 13] is to discretize it into equal timesteps and simulate each step following the reverse generative rate using an approximate simulation method called tau-leaping. During step $[t, t + \Delta t]$, all the transitions happening according to Eq. (1) are recorded first and simultaneously applied at the end of the step. When the discretization is finer than the number of denoising moves required, some steps may be wasted when no transitions occur during those steps. In such cases when no transition occurs during $[t - \Delta t, t]$, a neural network forward pass can be saved for step $[t, t + \Delta t]$ by using the cached $p_{1|t}(x_1^d | x_{t-\Delta t})$ from the previous step [10, 34], assuming the denoising probabilities remain

unchanged during $[t - \Delta t, t]$. However, as discussed in literature [6, 39, 29, 7], such modeling of the reverse denoiser is predicting single dimension transitions but not joint transitions of all dimensions. Therefore, the tau-leaping simulation will introduce approximation errors if multiple dimensions are changed during the same step.

**Gillespie Sampler.** Instead, we adopt the Gillespie algorithm [15, 16, 41], a simulation method that iteratively repeats the following two-step procedure: (1) sampling $\Delta t$, the holding time spent at current state until the next jump and (2) sampling which state transition occurs. In the first step, the holding time $\Delta t$ is drawn from an exponential distribution, with the rate equal to the total jump rate, defined as the sum of all possible jump rates in Eq. (12): $R_{t,\text{total}}(x_t) := \sum_d \sum_{j^d \neq x_t^d} R_{t,\text{jump}}(x_t, j^d)$. For the second step, the event at the jump is sampled according to $R_{t,\text{jump}}(x_t, j^d) / R_{t,\text{total}}(x_t)$, such that the likelihood of the next state is proportional to the rate from the current position. A straightforward way is using ancestral sampling by first selecting the dimension $\bar{d}$, followed by sampling the value to jump to $j^{\bar{d}}$:

$$\bar{d} \sim \text{Cat}\left(\sum_{j^{\bar{d}} \neq x_t^{\bar{d}}} R_{t,\text{jump}}(x_t, j^{\bar{d}}) / R_{t,\text{total}}(x_t)\right), \quad j^{\bar{d}} \sim \text{Cat}\left(R_{t,\text{jump}}(x_t, j^{\bar{d}}) / \sum_{j^{\bar{d}} \neq x_t^{\bar{d}}} R_{t,\text{jump}}(x_t, j^{\bar{d}})\right).$$

We find we can simplify the calculation of the total jump rate and next state transition by introducing the possibility of self-loop jumps into the CTMC. These allow the trajectory to remain in the current state after a jump occurs. This modification results in an equivalent but simpler simulated process with our plan-and-denoise method, formalized in the following proposition:

**Proposition 3.3.** *The original CTMC defined by the jump rate $R_{t,\text{jump}}$ given by Eq. (12) has the same distribution over trajectories as the modified self-loop CTMC with rate matrix*

$$\tilde{R}_t(x_t, j^d) = \frac{\dot{\alpha}_t}{1 - \alpha_t} p_\theta(z_t^d = N | x_t) p_{1|t}^\theta(x_1^d = j^d | x_t, z_t^d = N), \quad \forall x_t, j^d$$

*For this self-loop Gillespie algorithm, the total jump rate and next state distribution have the form*

$$\sum_{d, j^d} \tilde{R}_t(x_t, j^d) = \frac{\dot{\alpha}_t}{1 - \alpha_t} \sum_d p_\theta(z_t^d = N | x_t)$$

$$\bar{d} \sim \text{Cat}\left(p_\theta(z_t^{\bar{d}} = N | x_t)\right), \quad j^{\bar{d}} \sim \text{Cat}\left(p_{1|t}^\theta(x_1^{\bar{d}} = j^{\bar{d}} | x_t, z_t^{\bar{d}} = N)\right).$$

Intuitively, the modification preserves the inter-state jump rates, ensuring that the distribution of effective jumps remains unchanged. A detailed proof is provided in Appendix A.3.

**Remark 3.4.** *The Gillespie algorithm sets $\Delta t$ adaptively, which is given by the holding time until the next transition. This enables a more efficient discretization of timesteps, such that one step leads to one token denoised (if denoising is correct). In contrast, tau-leaping with equal timesteps can result in either no transitions or multiple transitions within the same step. Both scenarios are suboptimal: the former wastes a step, while the latter introduces approximation errors.*

**Adaptive Time Correction.** According to the sampled timesteps $\Delta t$, the sampling starts from noise at $t = 0$ and reaches data at 1. However, in practice, the actual time progression can be faster or slower than scheduled. For example, sometimes the progress is faster in the beginning when the starting sequence contains some clean tokens. More often, later in the process, the denoiser makes mistakes, and hence the time progression is slower than scheduled or even negative for some steps.

This raises the question: can we leverage the signal from the planner to make adaptive adjustments? For example, even if scheduled time reaches $t = 1.0$, but according to the planner $10\%$ of the data remains corrupted, the actual time progression under a linear schedule should be closer to $t = 0.9$. The reasonable approach is to assume that the process is not yet complete and continue the plan-and-denoise sampling. Under this 'time correction' mechanism, the stopping criterion is defined as continuing the sampling procedure until the planner determines that all positions are denoised, i.e., when $p_\theta(z_t^d = N | x_t) \approx 0$. In practice, we don't need to know the exact time; instead, we can continue sampling until either the stopping criterion is satisfied or the maximum budget of steps, $T$, is reached. The pseudo-algorithm for our proposed sampling method is presented in Algorithm 1. In cases where the denoiser use time information as input, we find it helpful to use the estimated time $\tilde{t}$ from the planner. At time $t$, from the noise schedule, we expect there to be $(1 - \alpha_t)D$ noised positions. The estimate of the number of noised positions from the denoiser is $\sum_{d'} p_\theta(z_t^{d'} = N | x_t)$. Therefore, the planner's estimate of the corruption time is $\tilde{t} = \alpha_t^{-1}(1 - \sum_{d'} p_\theta(z_t^{d'} = N | x_t)/D)$

where $\alpha_t^{-1}$ is the inverse noise schedule. This adjustment better aligns the time-data pair with the distribution that the denoiser was trained on.

Based on the decomposition of the generation into planning and denoising, the proposed sampling method maximally capitalizes on the available sampling steps budget. The Gillespie-based plan-and-denoise sampler allows for exact simulation and ensures no step is wasted by prioritizing the denoising of noisy tokens first. The time correction mechanism enables the planner to identify both initial and reintroduced noisy tokens, continuously denoising them until all are corrected. This mechanism shares similarities with the stochastic noise injection-correction step in EDM [22]. Instead of using hyperparameters for deciding how much to travel back, our time correction is based on the planner's estimate of the noise removal progress.

---

**Algorithm 1** DDPD Sampler

---

1: **init** $i \leftarrow 0, x_0 \sim p_0$, planner $p_\theta$, denoiser $p_{1|t}^\theta$, maximum steps $T$, stopping criteria $\epsilon$
2: **while** $i < T$ **or** $p_\theta(z_i^d = N | x_i) < \epsilon, \forall d$ **do**
3:     $\boxed{\text{Plan}}$ sample dimension $\bar{d} \sim \text{Cat}\left(p_\theta(z_i^{\bar{d}} = N | x_i)\right)$
4:     **if** denoiser uses time as input **then**
5:         $t \leftarrow \alpha_t^{-1}\left(1 - \sum_{d'} p_\theta(z_t^{d'} = N | x_t)/D\right)$
6:     $\boxed{\text{Denoise}}$ sample $j^{\bar{d}} \sim \text{Cat}\left(p_{1|t}^\theta(x_1^{\bar{d}} = j^{\bar{d}} | x_i, z_i^{\bar{d}} = N)\right), x_{i+1}^{\bar{d}} \leftarrow j^{\bar{d}}$
7:     $i \leftarrow i + 1$
8: **return** $x_i$

---

**Utilizing a Pretrained Mask Diffusion Denoiser.** In language modeling and image generation, mask diffusion denoisers have been found to be more accurate than uniform diffusion counterparts [3, 29], with recent efforts increasingly focused on training mask diffusion denoisers [36, 34, 13]. The following proposition offers a principled way to sample from the uniform denoiser by leveraging a strong pretrained mask diffusion denoiser, coupled with a separately trained planner.

**Proposition 3.5.** *From marginalization over $z_t$, which indicates if tokens are noise or data:*

$$p_{1|t}\left(x_1^d | x_t, z_t^d = N\right) = \sum_{z_t} p\left(z_t | x_t, z_t^d = N\right) p_{1|t}(x_1^d | x_t, z_t), \tag{13}$$

*samples from $p_{1|t,uniform}\left(x_1^d | x_t, z_t^d = N\right)$ can be drawn by first sampling $z_t$ from $p(z_t | x_t, z_t^d = N)$ and then using a mask diffusion denoiser to sample $x_1^d$ with $p_{1|t,mask}\left(x_1^d | \tilde{x}_t\right)$, where $\tilde{x}_t$ is the masked version of $x_t$ according to $z_t$.*

In practice, we can approximately sample $z^{\backslash d}$ from $p(z_t | x_t, z_t^d = N) \approx \prod_{d' \neq d} p_\theta(z_t^{d'} | x_t)$. This approximation becomes exact if $p(z_t^d | x_t)$ is either very close to 0 or 1, which holds true for most dimensions during generation. We validated this holds most of time in language modeling in Appendix E.5. Even if approximation errors in $z_t$ occasionally lead to increased denoising errors, our sampling algorithm can effectively mitigate this by using the planner to identify and correct these unintentional errors. In our controlled experiments, we validate this and observe improved generative performance by replacing the uniform denoiser with a mask diffusion denoiser trained on the same total number of tokens, while keeping the planner fixed.

## 4   Training

**Training objectives.** Our plan-and-denoise parameterization in Eq. (12) enables us to use two separate neural networks for modeling the planner and the denoiser. Alternatively, both the planner and denoiser outputs can be derived from a single uniform diffusion model, $p_{1|t}^\theta(x_1^d | x_t)$, as described in Proposition 3.1. This approach may offer an advantage on simpler tasks, where minimal approximation errors for neural network training can be achieved, avoiding the sampling approximation introduced in Proposition 3.5. However, in modern generative AI tasks, training is often constrained by neural network capacity and available training tokens, making approximation errors inevitable. By using two separate networks, we can better decompose the complex task, potentially enabling faster training – especially since planning is generally easier than denoising.

The major concern with decomposed modeling is that joint modeling could introduce unnecessarily coupled training dynamics, hindering effective backpropagation of the training signal across different models. However, as we prove in Theorem 4.1, the evidence lower bound (ELBO) of discrete diffusion decomposes neatly, allowing the use of direct training signals from the noise corruption process for independent training of the planner and denoiser (proof in Appendix A.2).

**Theorem 4.1.** *Let $x_1$ be a clean data point, and $x_t, z_t$ represent a noisy data point and its state of corruption drawn from $p_{t|1}(x_t, z_t|x_1)$. The ELBO for uniform discrete diffusion simplifies into the sum of the following separate cross-entropy-type objectives. The training of the planner $p_\theta$ reduces to a binary classification, where the goal is to estimate the corruption probability by maximizing:*

$$\mathcal{L}_{planner} = \mathbb{E}_{\mathcal{U}(t;0,1)p_{\mathrm{data}}(x_1)p_{t|1}(z_t,x_t|x_1)} \left[ \frac{\dot{\alpha}_t}{1-\alpha_t} \sum_{d=1}^{D} \log p_\theta \left( z_t^d|x_t \right) \right]. \tag{14}$$

*The denoiser $p_{1|t}^\theta$ is trained to predict clean data reconstruction distribution:*

$$\mathcal{L}_{denoiser} = \mathbb{E}_{\mathcal{U}(t;0,1)p_{\mathrm{data}}(x_1)p(x_t,z_t|x_1)} \left[ \frac{\dot{\alpha}_t}{1-\alpha_t} \sum_{d=1}^{D} \delta\{z_t^d, N\} \log p_{1|t}^\theta(x_1^d|x_t, z_t^d = N) \right]. \tag{15}$$

Standard transformer architectures can be used to parameterize both the denoiser and the planner, where the denoiser outputs $S$ logits and the planner outputs a single logit per dimension.

## 5 RELATED WORK

**Discrete Diffusion/Flow Models.** Previous discrete diffusion/flow methods, whether in discrete time [3, 21, 34] or continuous time [6, 39, 29, 7, 36, 13], adopt the denoiser-only or score-modeling perspective. In contrast, we introduce a theoretically grounded decomposition of the generative process into planning and denoising. DFM [7] and the Reparameterized Diffusion Model (RDM) [46] introduce stochasticity into the reverse flow/diffusion process, allowing for adjustable random jumps between states. This has been shown to improve the generation quality by providing denoiser more opportunities to correct previous errors. Additionally, RDM uses the denoiser's prediction confidence as a heuristic [14, 35, 8] for determining which tokens to denoise first. Lee et al. [24] introduces another heuristic in image generation that aims at correcting previous denoising errors by first generating all tokens and then randomly regenerating them in batches.

**Self-Correction Sampling.** Predictor-corrector sampling methods are proposed for both continuous and discrete diffusion [37, 6] that employ MCMC steps for correction after each predictor step. However, for continuous diffusion, this approach has been found to be less effective compared to the noise-injection stochasticity scheme [22, 42]. In the case of discrete diffusion, an excessively large number of corrector steps is required, which limits the method's overall effectiveness.

## 6 EXPERIMENT

Before going into details, we note that DDPD incurs 2 NFE v.s. 1 NFE per step in denoiser-only approaches, an extra cost we pay for planning. To ensure a fair comparison, we also evaluate denoiser-only methods that are either $2\times$ large or use $2\times$ steps. Our findings show that spending compute on planning is more effective than doubling compute on denoising when cost is a factor.

**Text8.** We first evaluate DDPD on the small-scale character-level text modeling benchmark, text8 [30], which consists of 100 million characters extracted from Wikipedia, segmented into chunks of 256 letters. Our experimental setup follows that of [7]. Methods for comparison include 1) autoregressive model 2) DFM: discrete flow model (and $2\times$ param. version) [7], the best available discrete diffusion/flow model for this task, 3) DFM-Uni: original DFM uniform diffusion using tau-leaping, 4) DDPD-DFM-Uni: DDPD using uniform diffusion model as planner and denoiser, 5) DDPD-UniD: DDPD with separately trained planner and uniform denoiser, 6) DDPD-MaskD: DDPD with separately planner and mask denoiser. Details in the sampling differences are summarized in Table 3. All models are of same size (86M) and trained for $750k$ iterations of batch size 2048, except for autoregressive model, which requires fewer iterations to converge. Generated samples are evaluated using the negative log-likelihood (NLL) under the larger language model GPT-J-6B [40]. Since NLL can be manipulated by repeating letters, we also measure token distribution entropy. High-quality samples should have both low NLL and entropy values close to the data distribution.

Fig. 2 shows the performance of various methods with different sampling step budgets. DFM methods use tau-leaping while DDPD methods use our proposed adaptive Gillespie sampler. The original mask diffusion (DFM, $\eta = 0$) and uniform diffusion (DFM-Uni) perform similarly, and adding stochasticity ($\eta = 15$) improves DFM's sample quality. Our proposed plan-and-denoise DDPD sampler consistently enhances the quality vs. diversity trade-off and significantly outperforms significantly outperforms DFM with $2\times$ parameters. Moreover, DDPD makes more efficient use of the inference-time budget (Fig. 2, middle), continuously refining the generated sequences.

We observe that DDPD with a single network (DDPD-DFM-Uni) outperforms using separately trained planner and denoiser, as the task simplicity allows all models to achieve $\geq 90\%$ denoising accuracy at $t = 0.85$. The benefit of reducing each model's burden is outweighed by compounded approximation errors (Section 3.2). The weaker performance of $\tau$-leaping (P$\times$MaskD in Fig. 7) confirms this issue lies in approximation errors, not the sampling scheme. To emulate a practical larger-scale task where the models are undertrained due to computational or capacity budget limitations, we reduced training steps from $750k$ to $20k$ (Fig. 2, right). With only $20k$ steps, using separate planner and denoiser performs comparably to DFM at $750k$, while the single model suffers mode collapse due to larger approximation errors, highlighting the benefits of faster separate learning.

Further ablation studies on imperfect training of either the planner or denoiser (Figs. 5 and 6) show that performance remains robust to varying levels of denoiser imperfection, thanks to the self-correction mechanism in the sampling process. An imperfect planner has a greater impact, shifting the quality-diversity Pareto front. In this case, training separate models proves more robust in preserving diversity and preventing mode collapse compared to training a single model. The ablation in Fig. 7 examines the individual effects of the modifications introduced in the DDPD sampler. We also measure the denoising error terms in the ELBO for the planner + denoiser setup vs. the single neural network approach, as shown in Tables 4 to 6 of Appendix E.1.3, which further validates the performance difference of various design choices. In Appendix E.2, we tested how quickly the sampling converges and reported statistics on how many dimensions are adjusted during the process.

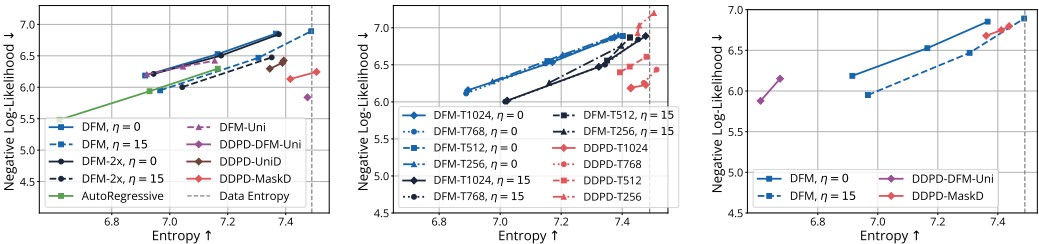

Figure 2: Negative log-likelihood measured with GPT-J versus sample entropy (in terms of tokens), with logit temperatures of the denoiser swept over $\{0.8, 0.9, 1.0\}$. *Left*: DDPD v.s. SOTA baselines. *Middle*: Varying sampling steps from 256 to 1024; both DFM and DDPD use the same mask-based denoiser. *Right*: DDPD single-neural-network v.s. DDPD planner + mask denoiser, both trained for $20k$ iterations. DFM at $750k$ iter.

**OpenWebText Language Modeling.** In Fig. 3, we compare DDPD with SEDD [29], both trained on the larger OpenWebText dataset [18]. We maintained the same experimental settings as in SEDD, with token vocabulary size $S = 50257$ and $D = 1024$, to validate whether planning improves generative performance under controlled conditions. We use the same pretrained SEDD-small or SEDD-medium score model as a mask diffusion denoiser, based on the conversion relationship outlined in Table 1. A separate planner network, with the same configuration as SEDD-small (90M), is trained for $400k$ iterations with batch size 512. We evaluated the quality of unconditional samples using generative perplexity, measured by larger language models GPT-2-L (774M) [33] and GPT-J (6B) [40]. Both SEDD and DDPD were simulated using 1024 to 4096 steps, with top-p $= 1.0$. SEDD used tau-leaping, while DDPD employed our newly proposed sampler. We also include GPT-2 [33] as the autoregressive baseline, with top-p sweeping from $0.7$ to $1.0$. We experimented DDPD sampler with both softmax selection (Fig. 9) and proportional selection (Fig. 10).

In contrast to SEDD, DDPD leveraged planning to continuously improve sample quality, with the highest improvements in the early stages and diminishing returns in later steps as the sequence is largely corrected. This shows that the planner optimally selects the denoising order and adaptively corrects accumulated mistakes.

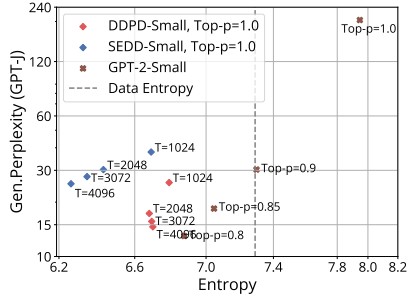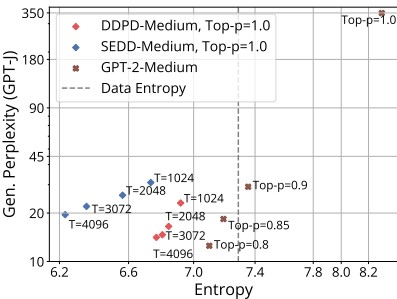

Figure 3: Generative perplexity ↓ v.s. entropy ↑ (both plotted in log-scale) of SEDD, DDPD and GPT-2.

**ImageNet** $256 \times 256$ **Generation with Discrete Tokens.** We represent images with discrete-valued tokens using a pre-trained tokenizer and decoder from Yu et al. [45], resulting in a token length $D = 128$. Our focus is comparing DDPD against existing sampling methods rather than achieving state-of-the-art performance.

We compare DDPD to two mask diffusion baselines: standard mask diffusion and MaskGIT [8]. MaskGIT selects tokens based on denoiser confidence logits with annealed random noise to avoid overly greedy selections. Experiments span from 8 to 128 sampling steps, employing parallel sampling for steps below 128. Results measured by FID scores [19] appear in Table 2, with additional metrics and samples in Appendices E.4 and F.3. Sampling in DDPD consists of two stages: parallel sampling followed by adaptive, noise-level-based refinement.

Standard mask diffusion performs poorly due to low denoiser accuracy (3% vs. 60% for language modeling at $t = 0.85$). MaskGIT achieves better results but sacrifices diversity for quality, becoming excessively greedy at higher steps. Conversely, DDPD achieves optimal results once sufficient steps correct early-stage errors without degrading FID at higher steps. Ablation studies on refinement steps are in Fig. 12.

We tested all methods with logit temperature annealing, finding $\tau = 0.6$ optimal among tested values, significantly enhancing standard mask diffusion but reducing diversity and increasing FID in MaskGIT. Another annealing method from Yu et al. [45], linearly reducing $\tau$ from 1.0 to 0.0, further improved mask diffusion but had minimal impact on DDPD. Additional evaluations with heuristics and classifier-free guidance are discussed in Appendix E.4.1. Notably, DDPD is the only method that performs consistently with or without the additional heuristics or CFG.

Table 2: FID score (↓) on ImageNet $256 \times 256$. MaskD refers to mask diffusion. The denoiser and parallel sampling schedule are kept the same as [45], without classifier-free guidance.

| Steps $T$ | No Logit Annealing | | | Logit temp $0.6$ | | | Logit temp $1.0 \rightarrow 0.0$ | | |
|---|---|---|---|---|---|---|---|---|---|
| | MaskD | MaskGIT | DDPD | MaskD | MaskGIT | DDPD | MaskD | MaskGIT | DDPD |
| 8 | 38.06 | 5.51 | 6.8 | 5.69 | 10.02 | 5.71 | 4.99 | 8.53 | 5.99 |
| 16 | 32.44 | 6.66 | 5.12 | 4.85 | 11.24 | 4.92 | 4.69 | 9.21 | 5.03 |
| 32 | 29.12 | 8.09 | 4.75 | 4.86 | 11.93 | 4.91 | 4.62 | 9.9 | 4.98 |
| 64 | 27.54 | 9.08 | 4.73 | 4.98 | 12.26 | 5.14 | 4.6 | 10.35 | 5.26 |
| 128 | 26.83 | 9.34 | 4.89 | 5.13 | 12.52 | 5.39 | 4.89 | 10.2 | 5.54 |

# 7 CONCLUSION

We introduced Discrete Diffusion with Planned Denoising (DDPD), a novel framework that decomposes the discrete generation process into planning and denoising. We propose a new adaptive sampler that leverages the planner for more effective and robust generation by adjusting time step sizes and prioritizing the denoising of the most corrupted positions. Additionally, it simplifies the learning process by allowing each model to focus specifically on either planning or denoising. The incorporation of planning makes the generative process more robust to errors made by the denoiser during generation. On GPT-2 scale language modeling and ImageNet $256 \times 256$ token generation, DDPD enables a significant performance boost compared to denoiser-only discrete diffusion models.

## 8 REPRODUCIBILITY STATEMENT

To facilitate reproducibility, we provide comprehensive details of our method in the main paper and Appendix D. This includes the model designs, hyper-parameters in training, sampling schemes and evaluation protocols for all the experiments. We further provide PyTorch pseudocode for the proposed adaptive Gillespie sampling algorithm.

## 9 ETHICS STATEMENT

This work raises ethical considerations common to deep generative models. While offering potential benefits such as generating high-quality text/image contents, these models can also be misused for malicious purposes like creating deepfakes or generating spam and misinformation. Mitigating these risks requires further research into guardrails for reducing harmful contents and collaboration with socio-technical experts.

Furthermore, the substantial resource costs associated with training and deploying deep generative models, including energy and water consumption, present environmental concerns. This work can save cost on training by reusing pre-trained denoiser and just focusing on training different planner models for slightly different tasks. At inference time, our newly proposed sampler is able to generate at better quality as compared to existing methods that use same amount of compute.

### ACKNOWLEDGMENTS

We thank Jiaxin Shi for valuable discussions on evaluation of ELBO and anonymous reviewers for their helpful feedback. SL and RGB acknowledge funding from MIT-IBM Watson AI Lab and NSF Award 2209892. JN acknowledges support from Toyota Research Institute. AC acknowledges support from the EPSRC CDT in Modern Statistics and Statistical Machine Learning (EP/S023151/1). This research used compute resources of the National Energy Research Scientific Computing Center (NERSC), a Department of Energy Office of Science User Facility using award GenAI@NERSC-m4737.

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

# Appendix

## A  PROOFS

### A.1  PROOF OF PROPOSITION 3.1

**Part 1: Calculate $p\left(z_t^d = N | x_t\right)$ in Eq. (10).**

We first derive how to calculate $p\left(z_t^d = N | x_t\right)$ in Eq. (10) using $p_{1|t}$.

First, from law of total probability,

$$p\left(z_t^d = N | x_t\right) = \sum_{\bar{j}^d} p_{1|t}\left(x_1^d = \bar{j}^d | x_t\right) p\left(z_t^d = N | x_1^d = \bar{j}^d, x_t\right) \tag{16}$$

Next we derive closed form of $p\left(z_t^d = N | x_1^d = \bar{j}^d, x_t\right)$.

According to the noise schedule,

$$p\left(z_t^d, x_t^d | x_1\right) = \begin{cases} \alpha_t & \text{if } z_t^d = D, x_t^d = x_1^d \\ 0 & \text{if } z_t^d = D, x_t^d \neq x_1^d \\ (1-\alpha_t)/S & \text{if } z_t^d = N, x_t^d = x_1^d \\ (1-\alpha_t)/S & \text{if } z_t^d = N, x_t^d \neq x_1^d \end{cases} \tag{17}$$

Using Bayes rule $p\left(z_t^d | x_t^d, x_1\right) = p\left(z_t^d, x_t^d | x_1\right) / p\left(x_t^d | x_1\right)$, we have

$$p\left(z_t^d | x_t^d, x_1\right) = \begin{cases} \frac{\alpha_t}{\alpha_t + (1-\alpha_t)/S} & \text{if } z_t^d = D, x_t^d = x_1^d \\ 0 & \text{if } z_t^d = D, x_t^d \neq x_1^d \\ \frac{(1-\alpha_t)/S}{\alpha_t + (1-\alpha_t)/S} & \text{if } z_t^d = N, x_t^d = x_1^d \\ 1 & \text{if } z_t^d = N, x_t^d \neq x_1^d \end{cases} \tag{18}$$

Plugging Eq. (18) into Eq. (16), we have

$$p\left(z_t^d = N | x_t\right) = \sum_{\bar{j}^d} p\left(x_1^d = \bar{j}^d | x_t\right) p\left(z_t^d = N | x_1^d = \bar{j}^d, x_t\right) \tag{19}$$

$$= \sum_{\bar{j}^d \neq x_t^d} p\left(x_1^d = \bar{j}^d | x_t\right) \cdot 1 + p\left(x_1^d = x_t^d | x_t\right) \cdot \frac{(1-\alpha_t)/S}{\alpha_t + (1-\alpha_t)/S} \tag{20}$$

$$= \sum_{\bar{j}^d \neq x_t^d} p\left(x_1^d = \bar{j}^d | x_t\right) \cdot 1 + p\left(x_1^d = x_t^d | x_t\right) \cdot \left(1 - \frac{\alpha_t}{\alpha_t + (1-\alpha_t)/S}\right) \tag{21}$$

$$= 1 - p\left(x_1^d = x_t^d | x_t\right) \frac{\alpha_t}{\alpha_t + (1-\alpha_t)/S} \tag{22}$$

which gives us the form in Eq. (10).

**Part 2: Calculate $p_{1|t}\left(x_1^d = j^d | x_t, z_t^d = N\right)$ in Eq. (11).**

From Bayes rule, we have

$$p_{1|t}\left(x_1^d = j^d | x_t, z_t^d = N\right) = \frac{p\left(x_1^d = j^d, z_t^d = N | x_t\right)}{p\left(z_t^d = N | x_t\right)} \tag{23}$$

$$= \frac{p\left(x_1^d = j^d | x_t\right) p\left(z_t^d = N | x_t, x_1^d = j^d\right)}{p\left(z_t^d = N | x_t\right)} \tag{24}$$

$$= \frac{p\left(x_1^d = j^d | x_t\right)}{p\left(z_t^d = N | x_t\right)}, \quad \forall x_t^d \neq j^d \tag{25}$$

Eq. (25) is by plugging in the following from Eq. (18) that $p\left(z_t^d = N | x_t, x_1^d = j^d\right) = 1$ if $x_t^d \neq j^d$.

**Part 3: Verify equivalence to the original optimal rate in Eq. (9).**

By plugging in Eq. (10) and Eq. (11) into Eq. (9), we have

$$R_t^{\text{unif}}\left(x_t, j^d\right) = \underbrace{\frac{\dot{\alpha}_t}{1 - \alpha_t}}_{\text{recovery rate}} \underbrace{p\left(z_t^d = N | x_t\right)}_{\text{prob. of corruption}} \underbrace{p_{1|t}\left(x_1^d = j^d | x_t, z_t^d = N\right)}_{\text{prob. of denoising}} = \underbrace{\frac{\dot{\alpha}_t}{1 - \alpha_t}}_{\text{recovery rate}} \underbrace{p_{1|t}\left(x_1^d = j^d | x_t\right)}_{\text{composed denoising prob.}}$$

(26)

## A.2 PROOF OF THEOREM 4.1: DERIVING THE ELBO FOR TRAINING

This derivation follows the Continuous Time Markov Chain framework. We refer the readers to Appendix C.1 of Campbell et al. [7] for a primer on CTMC. Here we provide the proof for $D = 1$ case. The result holds for $D > 1$ case following same arguments from Appendix E of Campbell et al. [7].

Let $W$ be a CTMC trajectory, fully described by its jump times $T_1, \cdots, T_n$ and state values between jumps $W_0, W_{T_0}, \cdots, W_{T_n}$. At time $T_k$, the state jumps from $W_{T_{k-1}}$ to $W_{T_k}$. With its path measure properly defined, we start with the following result from Campbell et al. [7] Appendix C.1, the ELBO of $\mathbb{E}_{p_{\text{data}}(x_1)}\left[\log p_\theta(x_1)\right]$ is given by introducing the corruption process as the variational distribution:

$$\mathbb{E}_{p_{\text{data}}(x_1)}\left[\log p_\theta(x_1)\right] \geq \int p_{\text{data}}(\mathrm{d}x_1)\mathbb{Q}^{|x_1}(\mathrm{d}\omega) \log \frac{\mathrm{d}\mathbb{P}^\theta}{\mathrm{d}\mathbb{Q}^{|x_1}}(\omega),$$

(27)

where

$$\frac{\mathrm{d}\mathbb{P}^\theta}{\mathrm{d}\mathbb{Q}^{|x_1}}(\omega) = \frac{p_0(W_0) \exp\left(-\int_{t=0}^{t=1} R_t^\theta(W_t^-)\mathrm{d}t\right) \prod_{t:W_t \neq W_t^-} R_t^\theta(W_t^-, W_t)}{p_{0|1}(W_0 | x_1) \exp\left(-\int_{t=0}^{t=1} R_t(W_t^- | x_1)\mathrm{d}t\right) \prod_{t:W_t \neq W_t^-} R_t(W_t^-, W_t | x_1)}.$$

(28)

Intuitively, the measure (probability) of a trajectory is determined by: the starting state from the prior distribution $p_0(W_0)$, and the product of the probability of waiting from $T_{k-1}$ to $T_k$ (which follows an Exponential distribution) and the transition rate of the jump from $W_t^-$ to $W_t$.

For our method, when simulating the data corruption process, we augment $W_t^{\text{aug}}$ to record both state values $W_t$ and its latent value $Z_t \in \{N, D\}^D$. The jump is defined to happen when the latent value jumps from $Z_{k-1}$ to $Z_k$ with one of the dimensions corrupted and the state value jumps from $W_{k-1}$ to $W_k$. Similarly, the ELBO of $\mathbb{E}_{p_{\text{data}}(x_1)}\left[\log p_\theta(x_1)\right]$ can be defined as:

$$\mathbb{E}_{p_{\text{data}}(x_1)}\left[\log p_\theta(x_1)\right] \geq \int p_{\text{data}}(\mathrm{d}x_1)\mathbb{Q}^{|x_1}(\mathrm{d}\omega^{\text{aug}}) \log \frac{\mathrm{d}\mathbb{P}^\theta}{\mathrm{d}\mathbb{Q}^{|x_1}}(\omega^{\text{aug}})$$

(29)

where the Radon-Nikodym derivative is given by:

$$\frac{\mathrm{d}\mathbb{P}^\theta}{\mathrm{d}\mathbb{Q}^{|x_1}}(\omega^{\text{aug}}) = \frac{p_0\left(W_0\right) \exp\left(-\int_{t=0}^{t=1} R_t^\theta\left(W_t^-\right)\mathrm{d}t\right) \prod_t R_t^\theta\left(W_t^-, W_t\right)}{p_{0|1}\left(W_0, Z_0 \mid x_1\right) \exp\left(-\int_{t=0}^{t=1} R_t\left(W_t^-, Z_t^- \mid x_1\right)\mathrm{d}t\right) \prod_t R_t\left(W_t^-, W_t, Z_t^-, Z_t \mid x_1\right)}$$

(30)

By plugging in Eq. (30) into Eq. (29), we arrive at

$$\mathcal{L}_{\text{ELBO}} = \int p_{\text{data}}\left(dx_1\right) \int_{\omega^{\text{aug}}:\mathbb{Q}^{|x_1}(\omega^{\text{aug}})>0} \mathbb{Q}^{|x_1}(d\omega^{\text{aug}}) \left\{ -\int_{t=0}^{t=1} R_t^\theta\left(W_t^-\right) dt + \sum_t \log R_t^\theta\left(W_t^-, W_t\right) \right\}$$

$$= \int p_{\text{data}}\left(dx_1\right) \int_{\omega^{\text{aug}}:\mathbb{Q}^{|x_1}(\omega^{\text{aug}})>0} \mathbb{Q}^{|x_1}(d\omega^{\text{aug}}) \left\{ -\int_{t=0}^{t=1} \frac{\dot{\alpha}_t}{1-\alpha_t} p_\theta\left(Z_t^- = N|W_t^-\right) \right.$$

$$\left. + \sum_t \log \frac{\dot{\alpha}_t}{1-\alpha_t} p_\theta\left(Z_t^- = N|W_t^-\right) p_{1|t}^\theta\left(x_1 = W_t|W_t^-, Z_t^- = N\right) \right\}$$

$$= \int p_{\text{data}}\left(dx_1\right) \int_{\omega^{\text{aug}}:\mathbb{Q}^{|x_1}(\omega^{\text{aug}})>0} \mathbb{Q}^{|x_1}(d\omega^{\text{aug}}) \left\{ -\int_{t=0}^{t=1} \frac{\dot{\alpha}_t}{1-\alpha_t} p_\theta\left(Z_t^- = N|W_t^-\right) \right.$$

$$\left. + \sum_{t\in\{T_1,\cdots,T_N\}} \left( \log \frac{\dot{\alpha}_t}{1-\alpha_t} p_\theta\left(Z_t^- = N|W_t^-\right) + \log p_{1|t}^\theta\left(x_1 = W_t|W_t^-, Z_t^- = N\right) \right) \right\}$$

$$(31)$$

Eq. (31) contain terms that depend on the planner $p_\theta\left(z_t = N|x_t\right)$ and the denoiser $p_{1|t}^\theta\left(x_1 = j|x_t, z_t = N\right)$. Next, we show those terms can be separated into two parts:

$$\mathcal{L}_{\text{denoiser}} = \mathbb{E}_{\mathcal{U}(t;0,1)p_{\text{data}}(x_1)p(x_t,z_t|x_1)} \left[ \frac{\dot{\alpha}_t}{1-\alpha_t} \delta\{z_t, N\} \log p_{1|t}^\theta(x_1|x_t, z_t = N) \right] \quad (32)$$

$$\mathcal{L}_{\text{planner}} = \mathbb{E}_{\mathcal{U}(t;0,1)p_{\text{data}}(x_1)p_{t|1}(z_t,x_t|x_1)} \left[ \frac{\dot{\alpha}_t}{1-\alpha_t} \log p_\theta\left(z_t|x_t\right) \right] \quad (33)$$

**First part: cross-entropy loss on $x_1$ denoising.**

All terms associated with $p_{1|t}^\theta\left(x_1 = W_t|W_t^-, Z_t^- = N\right)$ are:

$$\mathcal{L}_{\text{denoising}} = \int p_{\text{data}}\left(dx_1\right) \int_{\omega^{\text{aug}}} \mathbb{Q}^{|x_1}(d\omega^{\text{aug}}) \sum_t \log p_{1|t}^\theta\left(x_1 = W_t|W_t^-, Z_t^- = N\right)$$

$$= \int p_{\text{data}}\left(dx_1\right) \int_{\omega^{\text{aug}}} \mathbb{Q}^{|x_1}(d\omega^{\text{aug}}) \int_{t=0}^{t=1} \sum_{(y,u)} R_t\left((W_t, Z_t), (y, u)|x_1\right) \log p_{1|t}^\theta\left(x_1 = W_t|W_t^-, Z_t^- = N\right) \text{ Dynkin}$$

$$= \int p_{\text{data}}\left(dx_1\right) \int_{\omega^{\text{aug}}} \mathbb{Q}^{|x_1}(d\omega^{\text{aug}}) \int_{t=0}^{t=1} \sum_{(y,u)} \frac{\dot{\alpha}_t}{1-\alpha_t} \delta\{Z_t, N\}\delta\{u, D\}\delta\{y, x_1\} \log p_{1|t}^\theta\left(x_1 = W_t|W_t^-, Z_t^- = N\right)$$

$$= \int \int_{t=0}^{t=1} p_{\text{data}}\left(dx_1\right) \int_{\omega^{\text{aug}}} \mathbb{Q}^{|x_1}(d\omega^{\text{aug}}) \frac{\dot{\alpha}_t}{1-\alpha_t} \delta\{Z_t, N\} \log p_{1|t}^\theta\left(x_1 = W_t|W_t^-, Z_t^- = N\right)$$

$$= \mathbb{E}_{\mathcal{U}(t;0,1)p_{\text{data}}(x_1)p_{t|1}(z_t,x_t|x_1)} \left[ \frac{\dot{\alpha}_t}{1-\alpha_t} \delta\{z_t, N\} \log p_{1|t}^\theta\left(x_1|x_t, z_t = N\right) \right]$$

At the second equation, we use Dynkin's formula

$$\int p_{\text{data}}\left(dx_1\right) \mathbb{Q}^{|x_1}(d\omega) \sum_{t:\text{all jump times}} f\left(W_t^-, W_t\right) = \int p_{\text{data}}\left(dx_1\right) \mathbb{Q}^{|x_1}(d\omega) \int_{t=0}^{t=1} \sum_y R_t\left(W_t, y \mid x_1\right) f\left(W_t, y\right) dt$$

which allows us to switch from a sum over jump times into a full integral over time interval weighted by the probability of the jump happening and the next state the jump goes to.

**Second part: cross-entropy loss on $z_t = N$ prediction.**

The remaining terms are associated with $\frac{\dot{\alpha}_t}{1-\alpha_t} p_\theta\left(Z_t^- = N|W_t^-\right)$ which is the jump rate at $W_t^-$, i.e. $R_{t,\text{jump}}^\theta(W_t^-) = \sum_j R_{t,\text{jump}}^\theta(W_t^-, j) = \frac{\dot{\alpha}_t}{1-\alpha_t} p_\theta\left(Z_t^- = N|W_t^-\right)$ according to Eq. (12). In this proof,

we will use $R_t^\theta$ in short for $R_{t,\text{jump}}^\theta$. The remaining loss terms

$$\mathcal{L}_{\text{planner}} = \int p_{\text{data}}\,(\mathrm{d}x_1) \int_{\omega^{\text{aug}}} \mathbb{Q}^{|x_1}(\mathrm{d}\omega^{\text{aug}}) \left\{ -\int_{t=0}^{t=1} R_t^\theta\left(W_t^-\right)\mathrm{d}t + \sum_t \log R_t^\theta\left(W_t^-\right) \right\}$$

$$= \int p_{\text{data}}\,(\mathrm{d}x_1) \int_{\omega^{\text{aug}}} \mathbb{Q}^{|x_1}(\mathrm{d}\omega^{\text{aug}}) \left\{ -\int_{t=0}^{t=1} R_t^\theta\left(W_t\right)\mathrm{d}t + \int_{t=0}^{t=1} \sum_{(y,u)} R_t\left((W_t, Z_t), (y,u)|x_1\right) \log R_t^\theta\left(W_t\right)\mathrm{d}t \right\}$$

$$\text{Dynkin}$$

$$= \int p_{\text{data}}\,(\mathrm{d}x_1) \int_{\omega^{\text{aug}}} \mathbb{Q}^{|x_1}(\mathrm{d}\omega^{\text{aug}}) \left\{ -\int_{t=0}^{t=1} R_t^\theta\left(W_t\right)\mathrm{d}t + \int_{t=0}^{t=1} \frac{\dot{\alpha}_t}{1 - \alpha_t} \delta\{Z_t, N\} \log R_t^\theta\left(W_t\right)\mathrm{d}t \right\}$$

$$= \mathbb{E}_{\mathcal{U}(t;0,1)p_{\text{data}}(x_1)p(x_t, z_t|x_1)} \left[ -R_t^\theta\left(x_t\right) + \frac{\dot{\alpha}_t}{1 - \alpha_t} \delta\{z_t, N\} \log R_t^\theta\left(x_t\right) \right]$$

We first rewrite the loss as

$$\mathbb{E}_{\mathcal{U}(t;0,1)p(x_t)p(x_1, z_t|x_t)} \left[ -R_t^\theta\left(x_t\right) + \frac{\dot{\alpha}_t}{1 - \alpha_t} \delta\{z_t, N\} \log R_t^\theta\left(x_t\right) \right]$$

$$= \mathbb{E}_{\mathcal{U}(t;0,1)p(x_t)p(x_1, z_t|x_t)} \left[ -\frac{\dot{\alpha}_t}{1 - \alpha_t} p_\theta\left(z_t = N|x_t\right) + \frac{\dot{\alpha}_t}{1 - \alpha_t} \delta\{z_t, N\} \log \frac{\dot{\alpha}_t}{1 - \alpha_t} p_\theta\left(z_t = N|x_t\right) \right]$$

$$\tag{34}$$

If $x_t$ is given fixed, by taking the derivative of $p_\theta\left(z_t = N|x_t\right)$ and setting it to zero, we have the optimal solution to be:

$$p_\theta\left(z_t = N|x_t\right) = \mathbb{E}_{p(x_1, z_t|x_t)} \delta\{z_t, N\}$$

This is equivalent to optimizing the cross-entropy loss, which has the same optimal solution:

$$\mathbb{E}_{p(x_1, z_t|x_t)} \left[ \delta\{z_t, N\} \log p_\theta\left(z_t = N|x_t\right) + (1 - \delta\{z_t, N\}) \log\left(1 - p_\theta\left(z_t = N|x_t\right)\right) \right]$$

Plugging this $x_t$-conditional loss back to Eq. (34), we arrive at the cross entropy training loss for the planner:

$$\mathcal{L}_{\text{planner}} = \mathbb{E}_{\mathcal{U}(t;0,1)p_{\text{data}}(x_1)p_{t|1}(z_t, x_t|x_1)} \left[ \frac{\dot{\alpha}_t}{1 - \alpha_t} \log p_\theta\left(z_t|x_t\right) \right].$$

### A.3 Proof of Proposition 3.3: Continuous Time Markov Chains with Self-Connections

We first describe the stochastic process that includes self-loops as this differs slightly from the standard CTMC formulation. We have a rate matrix $\tilde{R}(i,j)$ that is non-negative at all entries, $\tilde{R}(i,j) \geq 0$. To simulate this process, we alternate between waiting for an exponentially distributed amount of time and sampling a next state from a transition distribution. The waiting time is exponentially distributed with rate $\sum_k \tilde{R}(i,k)$. The next state transition distribution is $\tilde{P}(j|i) = \frac{\tilde{R}(i,j)}{\sum_k \tilde{R}(i,k)}$. Note that $\tilde{P}(i|i)$ can be non-zero due to the self-loops in this style of process.

We can find an equivalent CTMC without self-loops that has the same distribution over trajectories as this self-loop process. To find this, we look at the infinitesimal transition distribution from time $t$ to time $t + \Delta t$. We let $J$ denote the event that the exponential timer expires during the period $[t, t + \Delta t]$. We let $\bar{J}$ denote the no jump event. For the self-loop process, the infinitesimal transition distribution is

$$p_{t+\Delta t|t}(j|i) = \mathbb{P}(J, j|i) + \mathbb{P}(\bar{J}, j|i)$$
$$= \mathbb{P}(J|i)\mathbb{P}(j|J, i) + \mathbb{P}(\bar{J}|i)\mathbb{P}(j|\bar{J}, i)$$

We have the following relations

$$\mathbb{P}(J|i) = \sum_k \tilde{R}(i,k)\Delta t \quad \text{property of exponential distribution}$$

$$\mathbb{P}(j|J,i) = \frac{\tilde{R}(i,j)}{\sum_k \tilde{R}(i,k)}$$

$$\mathbb{P}(j|\bar{J},i) = \delta\{j = i\}$$

Our infinitesimal transition distribution therefore becomes

$$p_{t+\Delta t|t}(j|i) = \left(\sum_k \tilde{R}(i,k)\Delta t\right) \frac{\tilde{R}(i,j)}{\sum_k \tilde{R}(i,k)} + \left(1 - \Delta t \sum_k \tilde{R}(i,k)\right) \delta\{j = i\}$$

$$= \Delta t \tilde{R}(i,j) + \delta\{j = i\} - \delta\{j = i\}\Delta t \sum_k \tilde{R}(i,k)$$

$$= \delta\{i = j\} + \Delta t \left(\tilde{R}(i,j) - \delta\{i = j\} \sum_k \tilde{R}(i,k)\right)$$

We now note that for a standard CTMC without self-loops and rate matrix $R(i,j)$, the infinitesimal transition probability is

$$p_{t+\Delta t|t}(j|i) = \delta\{i = j\} + \Delta t R(i,j)$$

Therefore, we can see our self-loop process is equivalent to the CTMC with rate matrix equal to

$$R(i,j) = \tilde{R}(i,j) \quad i \neq j$$

$$R(i,i) = -\sum_k \tilde{R}(i,k)$$

In other words, the CTMC rate matrix is the same as the self-loop matrix except simply removing the diagonal entries and replacing them with negative row sums as is standard.

In our case, the original CTMC without self-loops is defined by rate matrix

$$R_t(x_t, j^d) = \frac{\dot{\alpha}_t}{1 - \alpha_t} p_\theta(z_t^d = N|x_t) p_{1|t}^\theta(x_1^d = j^d|x_t, z_t^d = N), \qquad \forall x_t^d \neq j^d$$

We have free choice over the diagonal entries in our self-loop rate matrix and so we set the diagonal entries to be exactly the above equation evaluated at $x_t^d = j^d$.

$$\tilde{R}_t(x_t, j^d) = \frac{\dot{\alpha}_t}{1 - \alpha_t} p_\theta(z_t^d = N|x_t) p_{1|t}^\theta(x_1^d = j^d|x_t, z_t^d = N), \forall x_t^d, j^d$$

We can now evaluate the quantities needed for Gillespie's Algorithm. The first is the total jump rate

$$\sum_d \sum_{j^d} \tilde{R}_t(x_t, j^d) = \frac{\dot{\alpha}_t}{1 - \alpha_t} \sum_d \sum_{j^d} p_\theta(z_t^d = N|x_t) p_{1|t}^\theta(x_1 = j^d|x_t, z_t^d = N) \tag{35}$$

$$= \frac{\dot{\alpha}_t}{1 - \alpha_t} \sum_d p_\theta(z_t^d = N|x_t) \tag{36}$$

We now need to find the next state jump distribution. To find the dimension to jump to we use

$$\frac{\sum_{j^d} \tilde{R}_t(x_t, j^d)}{\sum_d \sum_{j^d} \tilde{R}_t(x_t, j^d)} = \frac{\frac{\dot{\alpha}_t}{1-\alpha_t} p_\theta(z_t^d = N|x_t)}{\frac{\dot{\alpha}_t}{1-\alpha_t}} = p_\theta(z_t^d = N|x_t) \tag{37}$$

To find the state within the chosen jump, the distribution is

$$\frac{\tilde{R}_t(x_t, j^d)}{\sum_{j^d} \tilde{R}_t(x_t, j^d)} = \frac{\frac{\dot{\alpha}_t}{1-\alpha_t} p_\theta(z_t^d = N|x_t) p_{1|t}^\theta(x_1^d = j^d|x_t, z_t^d = N)}{\frac{\dot{\alpha}_t}{1-\alpha_t} p_\theta(z_t^d = N|x_t)} \tag{38}$$

$$= p_{1|t}^\theta(x_1^d = j^d|x_t, z_t^d = N) \tag{39}$$

## B  GENERAL FORMULATION

### B.1  SCORE-ENTROPY BASED: SDDM [39], SEDD [29]

For coherence, we assume $t = 0$ is noise and $t = 1$ is data, while discrete diffusion literature consider a flipped notion of time. Following Campbell et al. [6], (see Sec. H.1 of Campbell et al. [7]), the conditional reverse rate of discrete diffusion considered in [39, 29] is defined to be:

$$R_t^{\text{diff}}\left(x_t, j^d \mid x_1^d\right) = R_t(j^d, x_t^d)\frac{p_{t|1}\left(j^d \mid x_1^d\right)}{p_{t|1}\left(x_t^d \mid x_1^d\right)} \tag{40}$$

with the forward corruption rate $R_t = \frac{\dot{\alpha}_t}{S\alpha_t}\left(\mathbb{1}\mathbb{1}^\top - S\mathbf{I}\right)$ for uniform diffusion or $R_t = \frac{\dot{\alpha}_t}{\alpha_t}\left(\mathbb{1}\mathbf{e}_{\mathbb{M}}^\top - \mathbf{I}\right)$ for mask diffusion, such that the corruption schedule is according to $\alpha_t$.

And the expected reverse rate for $x_t^d \neq j^d$ is given by:

$$R_t^{\text{diff}}\left(x_t, j^d\right) = \mathbb{E}_{p_{1|t}(x_1^d \mid x_t)} R_t^{\text{diff}}\left(x_t, j^d \mid x_1^d\right) \tag{41}$$

$$= \sum_{x_1^d} p_{1|t}\left(x_1^d \mid x_t\right) R_t(j^d, x_t^d)\frac{p_{t|1}\left(j^d \mid x_1^d\right)}{p_{t|1}\left(x_t^d \mid x_1^d\right)} \tag{42}$$

$$= R_t \sum_{x_1^d} p_{1|t}\left(x_1^d \mid x_t\right)\frac{p_{t|1}\left(j^d \mid x_1^d\right)}{p_{t|1}\left(x_t^d \mid x_1^d\right)} \tag{43}$$

$$= R_t \sum_{x_1^d} \frac{p\left(x_t^d \mid x_1^d, x_t^{\backslash d}\right) p\left(x_1^d \mid x_t^{\backslash d}\right)}{p\left(x_t^d \mid x_t^{\backslash d}\right)}\frac{p_{t|1}\left(j^d \mid x_1^d\right)}{p_{t|1}\left(x_t^d \mid x_1^d\right)} \tag{44}$$

$$= R_t \sum_{x_1^d} \frac{p\left(x_1^d \mid x_t^{\backslash d}\right)}{p\left(x_t^d \mid x_t^{\backslash d}\right)}p_{t|1}\left(j^d \mid x_1^d\right) \tag{45}$$

$P_t^\theta$ **Parameterization in SDDM.**   From here we can derive the rate derived in Eq. (16) in SDDM [39] which uses a neural network to parameterize $p^\theta\left(x_t^d \mid x_t^{\backslash d}\right)$

$$R_t^{\text{diff}}\left(x_t, j^d\right) = R_t\frac{\sum_{x_1^d} p\left(x_1^d \mid x_t^{\backslash d}\right) p_{t|1}\left(j^d \mid x_1^d\right)}{p\left(x_t^d \mid x_t^{\backslash d}\right)} = R_t\frac{p_t^\theta\left(j^d \mid x_t^{\backslash d}\right)}{p_t^\theta\left(x_t^d \mid x_t^{\backslash d}\right)}$$

$S_t^\theta$ **Parameterization in SEDD.**   SEDD [29] introduces the notion of score that directly models $\frac{p_t(j^d, x_t^{\backslash d})}{p_t(x_t^d, x_t^{\backslash d})}$ with $s_\theta\left(x_t\right)_{x_t^d \to j}$. Hence the reverse rate is parameterized by:

$$R_t^{\text{diff}}\left(x_t, j^d\right) = R_t\frac{p_t\left(j^d \mid x_t^{\backslash d}\right)}{p_t\left(x_t^d \mid x_t^{\backslash d}\right)} = R_t\frac{p_t\left(j^d, x_t^{\backslash d}\right)}{p_t\left(x_t^d, x_t^{\backslash d}\right)} = R_t s_\theta\left(x_t\right)_{x_t^d \to j}$$

$p_{1|t}^\theta$ **Parameterization in SDDM.**   In Eq. (24) of Sun et al. [39], the alternative parameterization uses a neural network to parameterize $p_{1|t}^\theta\left(x_1^d \mid x_t^{\backslash d}\right)$ and the rate is given by:

$$R_t^{\text{diff}}\left(x_t, j^d\right) = R_t\frac{p\left(j^d \mid x_t^{\backslash d}\right)}{p\left(x_t^d \mid x_t^{\backslash d}\right)}$$

$$= R_t\frac{\sum_{x_1^d} p_{1|t}^\theta\left(x_1^d \mid x_t^{\backslash d}\right) p_{t|1}\left(j^d \mid x_1^d\right)}{\sum_{x_1^d} p_{1|t}^\theta\left(x_1^d \mid x_t^{\backslash d}\right) p_{t|1}\left(x_t^d \mid x_1^d\right)}$$

**Connection to reverse rate in Campbell et al. [7].**    In mask diffusion case, the rate of SD-DM/SEDD coincides with the rate used in Eq. (42), i.e. rate of discrete diffusion and discrete flow formulation are the same for the mask diffusion case, $R_t^{\text{diff}} = R_t^{*,\text{DFM}}$. We have for $x_t^d = \mathbb{M}$ and $j^d \neq \mathbb{M}$:

$$
\begin{aligned}
R_t^{\text{diff}}\left(x_t, j^d\right) &= \sum_{x_1^d} p_{1|t}\left(x_1^d \mid x_t\right) R_t(j^d, x_t^d) \frac{p_{t|1}\left(j^d \mid x_1^d\right)}{p_{t|1}\left(x_t^d \mid x_1^d\right)} \\
&= R_t p_{1|t}\left(x_1^d = j^d \mid x_t\right) \frac{\alpha_t}{1 - \alpha_t} \\
&= \frac{\dot{\alpha}_t}{\alpha_t} p_{1|t}\left(x_1^d = j^d \mid x_t\right) \frac{\alpha_t}{1 - \alpha_t} \\
&= \dot{\alpha}_t \frac{1}{1 - \alpha_t} p_{1|t}\left(x_1^d = j^d \mid x_t\right)
\end{aligned}
$$

We can find that the parameterization of the generative rate used in DFM is only different from the SDDM/SEDD's parameterization by a scalar.

In the uniform diffusion case, the reverse rate used for discrete diffusion effectively generates the same marginal distribution $p_{t|1}$ and $p_t$, but the difference lies in that the rate used for discrete diffusion is the sum of the rate introduced in Campbell et al. [7] plus a special choice of the CTMC stochasticity that preserve detailed balance: $R_t^{\text{diff}} = R_t^{*,\text{DFM}} + R_t^{\text{DB}}$. Details are proved in H.1 in Campbell et al. [7]

## C    Additional technical details

### C.1    Evaluating the ELBO

Note that the ELBO values are only comparable between uniform diffusion methods or mask diffusion methods, since they have different marginal distribution $p_{t|1}$ and hence different trajectory path distribution $\mathbb{Q}(W \in d\omega)$. Based on Eq. (29), we write out the ELBO terms for mask diffusion and uniform diffusion. Results about log-likelihood in prior works [3, 7, 29, 36, 34] are reporting the (denoising) rate transitioning term only, i.e., $\log p_{1|t}^\theta\left(x_1^{d'} = W_t^{d'}|W_t^-\right)$.

#### C.1.1    Mask diffusion ELBO

**Term 1: Prior ratio** $\log \frac{p_0(W_0)}{p_{0|1}(W_0|x_1)} = 0$**.**

We observe that $\frac{p_0(W_0)}{p_{0|1}(W_0|x_1)} = 1$ since the starting noise distribution is the same. Hence $\log \frac{p_0(W_0)}{p_{0|1}(W_0|x_1)} = 0$.

**Term 2: Rate Matching** $\log \frac{\exp\left(-\int_{t=0}^{t=1} R_t^\theta(W_t^-)dt\right)}{\exp\left(-\int_{t=0}^{t=1} R_t(W_t^-|x_1)dt\right)} = 0$**.**

In the mask diffusion case, this term equals to 1, since

$$
R_t^\theta(W_t^-) = \frac{\dot{\alpha}_t}{1 - \alpha_t} \sum_{d=1}^D \delta\left\{W_t^{-,d}, \mathbb{M}\right\}, \quad R_t(W_t^-|x_1) = \frac{\dot{\alpha}_t}{1 - \alpha_t} \sum_{d=1}^D \delta\left\{W_t^{-,d}, \mathbb{M}\right\}
$$

For any trajectory $W_t, t \in [0, 1)$, $R_t^\theta(W_t^-) = R_t(W_t^-|x_1)$ and hence Term 2 equals to 0, i.e. $\log \frac{\exp\left(-\int_{t=0}^{t=1} R_t^\theta(W_t^-)dt\right)}{\exp\left(-\int_{t=0}^{t=1} R_t(W_t^-|x_1)dt\right)} = 0$.

**Term 3: Rate Transitioning** $\log \frac{R_t^\theta(W_t^-, W_t)}{R_t(W_t^-, W_t|x_1)}$**.**

Let the jump at $t$ happens at dimension $d'$, we have

$$
R_t^\theta(W_t^-, W_t) = \frac{\dot{\alpha}_t}{1 - \alpha_t} \delta\left\{W_t^{-,d'}, \mathbb{M}\right\} p_{1|t}^\theta\left(x_1^{d'} = W_t^{d'}|W_t^-\right), \quad R_t(W_t^-, W_t|x_1) = \frac{\dot{\alpha}_t}{1 - \alpha_t} \delta\left\{W_t^{-,d'}, \mathbb{M}\right\}
$$

Since before the jump $W_t^{-,d'}$ must be at mask state in order for jump to happen, hence this term simplifies to $\log \frac{R_t^\theta(W_t^-, W_t)}{R_t(W_t^-, W_t|x_1)} = \log p_{1|t}^\theta \left( x_1^{d'} = W_t^{d'}|W_t^- \right)$.

### C.1.2 Uniform diffusion ELBO

**Term 1: Prior ratio** $\log \frac{p_0(W_0)}{p_{0|1}(W_0|x_1)} = 0$.

We observe that $\frac{p_0(W_0)}{p_{0|1}(W_0|x_1)} = 1$ since the starting noise distribution is the same uniform distribution. Hence $\log \frac{p_0(W_0)}{p_{0|1}(W_0|x_1)} = 0$.

**Term 2: Rate Matching** $\log \frac{\exp\left(-\int_{t=0}^{t=1} R_t^\theta(W_t^-)dt\right)}{\exp\left(-\int_{t=0}^{t=1} R_t(W_t^-|x_1)dt\right)}$.

If the generative process is parameterized by $p_{1|t}^\theta(x_1^d|x_t)$ in Eq. (4):

$$R_t^\theta(W_t^-) = \frac{\dot{\alpha}_t}{1-\alpha_t} \sum_{d=1}^D p_{1|t}^\theta(x_1^d \neq W_t^{-,d}|x_t), \quad R_t(W_t^-|x_1) = \frac{\dot{\alpha}_t}{1-\alpha_t} \sum_{d=1}^D (1 - \delta\left\{W_t^{-,d}, x_1^d\right\})$$

If the reverse generative process is parameterized as our approach in Eq. (12):

$$R_t^\theta(W_t^-) = \frac{\dot{\alpha}_t}{1-\alpha_t} \sum_{d=1}^D p^\theta(z_t^{-,d} = N|x_t), \quad R_t(W_t^-, Z_t^-|x_1) = \frac{\dot{\alpha}_t}{1-\alpha_t} \sum_{d=1}^D \delta\left\{Z_t^{-,d}, N\right\}$$

Term 2 simplifies to:

$$\int_{t=0}^{t=1} R_t(W_t^-|x_1)dt - \int_{t=0}^{t=1} R_t^\theta(W_t^-)dt$$
$$= \int_{t=0}^{t=1} \left[R_t\left(W_t^-|x_1\right) - R_t^\theta\left(W_t^-\right)\right]dt$$
$$= \mathbb{E}_{\mathcal{U}(t;0,1)}\left[R_t\left(W_t^-|x_1\right) - R_t^\theta\left(W_t^-\right)\right]$$

Similarly, it simplifies to $\mathbb{E}_{\mathcal{U}(t;0,1)}\left[R_t\left(W_t^-, Z_t^-|x_1\right) - R_t^\theta\left(W_t^-\right)\right]$ for DDPD.

For a given $W_t$ or $W_t^{\text{aug}}$, we can approximate this term with Monte-Carlo samples from $t \sim \mathcal{U}(t;0,1)$.

**Term 3: Rate Transitioning** $\log \frac{R_t^\theta(W_t^-, W_t)}{R_t(W_t^-, W_t|x_1)}$.

If using parameterization $p_{1|t}^\theta(x_1^d|x_t)$ in Eq. (4):

$$R_t^\theta(W_t^-, W_t) = \frac{\dot{\alpha}_t}{1-\alpha_t} p_{1|t}^\theta\left(x_1^{d'} = W_t^{d'}|W_t^-\right), \quad R_t(W_t^-, W_t|x_1) = \frac{\dot{\alpha}_t}{1-\alpha_t}\left(1 - \delta\left\{W_t^{-,d'}, x_1^d\right\}\right)\delta\left\{W_t^{d'}, x_1^d\right\}$$

We know for the trajectory $W_t$, before the jump $W_t^{-,d'} \neq x_1^d$ and after the jump $W_t^{d'} = x_1^d$, therefore $R_t(W_t^-, W_t|x_1) = \frac{\dot{\alpha}_t}{1-\alpha_t}$. Hence the term simplifies to

$$\log \frac{R_t^\theta(W_t^-, W_t)}{R_t(W_t^-, W_t|x_1)} = \log p_{1|t}^\theta\left(x_1^{d'} = W_t^{d'}|W_t^-\right)$$

If using our parameterization in Eq. (12):

$$R_t^\theta(W_t^-, W_t) = \frac{\dot{\alpha}_t}{1-\alpha_t} p^\theta\left(z_t^{-,d'} = N|W_t^-\right) p_{1|t}^\theta\left(x_1^{d'} = W_t^{d'}|W_t^-, z_t^{-,d'} = N\right),$$

$$R_t(W_t^-, W_t, Z_t^-, Z_t|x_1) = \frac{\dot{\alpha}_t}{1-\alpha_t}\delta\left\{z_t^{-,d'}, N\right\}\delta\left\{z_t^{d'}, D\right\}\delta\left\{W_t^{d'}, x_1^d\right\} = \frac{\dot{\alpha}_t}{1-\alpha_t}$$

The term simplifies to

$$\log \frac{R_t^\theta(W_t^-, W_t)}{R_t(W_t^-, W_t|x_1)} = \log\left[p^\theta\left(z_t^{-,d'} = N|W_t^-\right) p_{1|t}^\theta\left(x_1^{d'} = W_t^{d'}|W_t^-, z_t^{-,d'} = N\right)\right]$$

# D   IMPLEMENTATION DETAILS

## D.1   TEXT8

**Models and training.**   We used the same transformer architecture from the DFM [7] for the denoiser model, with architectural details provided in Appendix I of [7]. For the planner, we modified the final layer to output a logit value representing the probability of noise. To prevent the planner model from exploiting the the current time step information to cheating on predicting the noise level, we find it necessary to not use time-embedding. Unlike the original DFM implementation, which uses self-conditioning inputs with previously predicted $x_1$, we omit self-conditioning in all of our trained models, as we found it had minimal impact on the results. When training the planner and denoiser, we implemented the optimization objectives in Theorem 4.1 as the cross entropy between target and predicted noise state and tokens, averaged over the corrupted dimensions. A linear noise schedule is used. We do not apply the time-dependent prefactor $\frac{\dot{\alpha}_t}{1-\alpha_t}$ to the training examples, as the signals from each corrupted token are independent. All models follow Campbell et al. [7] which is based on the smallest GPT2 architecture (768 hidden dimensions, 12 transformer blocks, and 12 attention heads) and have 86M parameters. We increase the model size to 176M parameters for DFM-2× with 1024 hidden dimensions, 14 transformer blocks, and 16 attention heads.

The following models were trained for text8:

- Autoregressive: $p(x^d|x^{1:d-1})$
- Uniform diffusion denoiser (DFM-Uni): $p_{1|t}(x_1^d|x_t)$
- Planner: $p(z_t^d|x_t)$
- Noise-conditioned uniform diffusion denoiser (UniD): $p_{1|t}(x_1^d|x_t, z_t^d = N)$
- Mask diffusion denoiser (MaskD): $p_{1|t}(x_1^d|x_t, x_t^d = \mathbb{M})$

We maintained the training procedure reported in [7], which we reproduce here for completeness. For all models, we used an effective batch size of 2048 with micro-batch 512 accumulated every 4 steps. For optimization, we used AdamW [28] with a weight decay factor of 0.1. Learning rate was linearly warmed up to $10^{-4}$ over 1000 steps, and decayed using a cosine schedule to $10^{-5}$ at 1M steps. We used the total training step budget of 750k steps. We saved checkpoints every 150k steps for ablation studies reported in Figs. 5 and 6. EMA was not used for text8 models. We trained our models on four A100 80GB GPUs, and it takes around 100 hours to finish training for $750k$ iterations.

Table 3: Sampling schemes used for text8 experiments.

| Method | Planner | Denoiser | Sampling | Options |
|---|---|---|---|---|
| DFM | N/A | MaskD | tau-leaping | stochasticity $\eta = 0, 15$ |
| DFM-Uni | DFM-Uni | | tau-leaping | |
| DDPD-DFM-Uni | DFM-Uni | DFM-Uni | Gillespie | A, A+B, A+B+C |
| P×UniD | Planner | UniD | tau-leaping | |
| P×MaskD | Planner | MaskD | tau-leaping | |
| DDPD-UniD | Planner | UniD | Gillespie | A, A+B, A+B+C |
| DDPD-MaskD | Planner | MaskD | Gillespie | A, A+B, A+B+C |

**Sampling schemes.**   The sampling schemes used for experiments in the main text are outlined in Table 3. Gillespie Algorithm options A, B, and C are defined as follows:

- A: Default DDPD Gillespie sampling in Algorithm 1
- +B: Continue sampling until the maximum time step budget is reached
- +C: Use the softmax of noise prediction logits (over the dimension axis) instead of normalized prediction values to select the dimensions that will be denoised

The implementation of these options when the uniform diffusion denoiser (DFM-Uni) is decomposed as a planner and a denoiser is presented in Listing 1. In Listing 2, we include the implementation of

Listing 1: Gillespie Algorithm sampling loop with uniform diffusion denoiser (DFM-Uni) decomposed as a planner and a denoiser.

```python
import torch
import torch.nn.functional as F

B = batch_size
D = num_dimensions
S = mask_token_id = vocab_size
eps = stopping_criteria

samples = torch.randint(0, S, (B, D), dtype=torch.int64)
time = torch.zeros(B, dtype=torch.float)
is_time_up = torch.zeros(B, dtype=torch.bool)

for i in range(timesteps):
    # Planning: compute probabilities of changing each dimension
    logits = model(samples, time)  # (B, D, S+1)
    logits[:, :, mask_token_id] = -1e4
    pt_x1_probs = F.softmax(logits, dim=-1)
    pt_x1_probs_at_xt = torch.gather(pt_x1_probs, -1, samples[:, :, None])  # (B, D, 1)
    p_if_change = 1 - pt_x1_probs_at_xt.squeeze()
    p_if_change = torch.clamp(p_if_change, min=1e-20, max=1.0)  # (B, D)
    total_noise = p_if_change.sum(-1)

    # Continue (Gillespie option B) or check stopping criteria
    if allow_time_backwards:
        pass
    else:
        is_time_up = (p_if_change < eps).all(-1)
        if is_time_up.all():
            break

    # Planning: get dimensions that change
    if use_softmax_for_dim_change:  # Use softmax instead (Gillespie option C)
        logits_dim_change = torch.logit(p_if_change.to(torch.float64), eps=1e-10)
        dim_change = torch.multinomial(
            torch.softmax(logits_dim_change, dim=-1), 1
        ).squeeze()  # (B,)
    else:
        dim_change = torch.multinomial(p_if_change, 1).squeeze()

    # Compute time input from planner output
    time = 1.0 - total_noise / D

    # Denoising: sample new values for the dimensions that change
    logits = model(samples, time)
    logits[:, :, mask_token_id] = -1e4
    pt_x1_probs = F.softmax(logits, dim=-1)  # (B, D, S+1)

    probs_change = pt_x1_probs[torch.arange(B), dim_change, :]  # (B, S+1)
    probs_change[torch.arange(B), samples[torch.arange(B), dim_change]] = 0.0
    x1_values = torch.multinomial(probs_change, 1).squeeze()  # (B,)
    samples[~is_time_up, dim_change[~is_time_up]] = x1_values[~is_time_up]
```

option B and option C when a separate planner and a separate denoiser are used. Option A of using a separate planner and a denoiser follows the same logic of Listing 1 except using separate output from the planner and the denoiser.

**Evaluation.** For each specified sampling scheme and sampling time step budget, we sampled 512 sequences with $D = 256$. Using the GPT-J (6B) model [40], we computed the average negative log-likelihood for each sequence, and using the same tokenization scheme (BPE in [33]), we calculated sequence entropy as the sum over all dimensions.

## D.2 OPENWEBTEXT

**Models and training.** We used the same model architectures from SEDD [29], which are based on the diffusion transformer (DiT) [32] and use rotary positional encodings [38]. We followed their training procedure closely for the OpenWebText experiments. Like the text8 models, we modified the final layer of DiT to serve as a noise probability logit predictor for the planner model. SEDD models

Listing 2: DDPD sampling loop with a separate planner and a denoiser.

```python
import torch
import torch.nn.functional as F

timesteps = T
B = batch_size
D = num_dimensions
S = mask_token_id = vocab_size
eps = stopping_criteria

samples = torch.randint(0, S, (B, D), dtype=torch.int64)
time = torch.zeros(B, dtype=torch.float)

for i in range(timesteps):
    # Planning: compute probabilities of each dimension being corrupted
    if_noise_logits = planner_model(samples)   # (B, D)
    # check for early stopping criteria: if every dimension is denoised
    prob_if_noise = torch.sigmoid(if_noise_logits)
    if (prob_if_noise < eps).all():
        break
    if use_softmax_for_dim_change: # Option C
        dim_change = torch.multinomial(
            torch.softmax(if_noise_logits, dim -1), 1
        ).squeeze()
    else: # Option B
        dim_change = torch.multinomial(prob_if_noise, 1).squeeze()

    # Denoising: sample new values for the dimensions that change
    # compute time input from planner output
    if use_mask_denoiser:
        mask = torch.bernoulli(prob_if_noise).bool().long() # sampling z_t
        mask[torch.arange(B), dim_change] = 1 # always mask the dimensions that are picked for denoising
        masked_sample = torch.where(mask, samples, mask_token_id)
        time = 1.0 - mask.sum(-1)/D
        logits = denoiser_model(masked_sample, time)
        logits[:, :, mask_token_id] = -1e4
    else:
        time = 1.0 - prob_if_noise.sum(-1) / D
        logits = denoiser_model(samples, time)
    pt_x1_probs = F.softmax(logits, dim=-1)   # (B, D, S+1)
    probs_change = pt_x1_probs[torch.arange(B), dim_change, :]   # (B, S+1)
    x1_values = torch.multinomial(probs_change, 1).squeeze()   # (B,)
    samples[torch.arange(B), dim_change] = x1_values
```

use the noise level $\sigma$ instead of time $t$ for the time embeddings. While we retain this model input by using their $\sigma(t)$, we replace it with zero when training the planner, similarly to the text8 models. All models were trained with a batch size of 128 and gradients were accumulated every 4 steps. We used AdamW [28] with a weight decay factor of 0, and the learning rate was linearly warmed up to $3 \times 10^{-4}$ over the first 2500 steps and then held constant. EMA with a decay factor of 0.9999 was applied to the model parameters. We validated the models on the OpenWebText dataset [17]. The mask denoisers are taken from the pretrained checkpoints of Lou et al. [29]. SEDD-small has 90M parameters and SEDD-medium has 320M parameters. We trained our planner models on nodes with four A100 80GB GPUs for $400k$ iterations. We only trained the planner models in the size of GPT-2-Small, which is 768 hidden dimensions, 12 layers, and 12 attention heads.

**Sampling and evaluation.** We employed Tweedie tau-leaping denoising scheme for SEDD, and adaptive Gillespie sampler for DDPD, and different nucleus sampling thresholds (top-p values of 0.8, 0.85, 0.9, and 1.0) for GPT-2. For all models and sampling schemes, we generated 200 samples of sequence length 1024 and evaluated the generative perplexity using the GPT-2 Large [33] and GPT-J [40] models.

### D.3 IMAGE GENERATION WITH TOKENS

**Models and training.** For tokenization and decoding of images, we use TiTok-S-128 model [45], which tokenizes $256 \times 256$ image into $D = 128$ tokens with the codebook size of $S = 4096$. Both mask diffusion denoiser and planner models use the U-Vit model architecture of MaskGIT [8] as implemented in the codebase of [45], with 768 hidden dimensions, 24 layers, and 16 attention heads. The mask denoisers are taken from pretrained checkpoints from Yu et al. [45]. The planner is trained with batch size 2048 for $400k$ iterations on 4 A100-80GB GPUs. We used AdamW [28] optimizer with a weight decay factor of 0.03, $\beta_1 = 0.9$, and $\beta_2 = 0.96$, and a learning rate of $2 \times 10^{-4}$. The learning rate schedule included a linear warmup over the first 10k steps, followed by cosine annealing down to a final learning rate of $10^{-5}$. EMA was applied with a decay factor of 0.999.

**Evaluation.** We utilize the evaluation code from ADM [11] to compute the FID scores [19] and inception scores. For this evaluation, 50,000 images are generated across all classes. Each image is produced by first generating tokens, followed by decoding with the TiTok-S-128 decoder.

## E ADDITIONAL RESULTS

### E.1 TEXT8

#### E.1.1 EFFECT OF APPROXIMATION ERRORS IN DENOISER AND PLANNER

We conducted experiments to measure the effect of approximation errors in denoiser and planner on the generation quality. Results are summarized in Figs. 4 to 6.

#### E.1.2 ABLATION OF CHANGES INTRODUCED IN DDPD SAMPLER

We conducted controlled experiment to measure the individual effect of the changes we introduced to the sampling process. Results are summarized in Fig. 7

#### E.1.3 MODEL LOG-LIKELIHOODS ON TEST DATA

Following ELBO terms derived in Appendix C.1, we calculate them for three different design choices:

- A single uniform diffusion neural network, but decomposed into planner and denoiser.
- Separate planner network and uniform diffusion denoiser network
- Separate planner network and mask diffusion denoiser network

In Table 4, we evaluate the ELBO terms for three methods both trained for $750k$ iterations (near optimality). We observe that the mask diffusoin denoiser has a better denoising performance even with mask approximation error introduced in the step of Proposition 3.5. In Table 5, We also observe

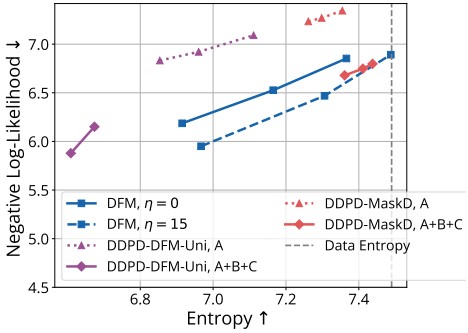

Figure 4: Comparing DDPD sampling under imperfect learning: 1) a single uniform diffusion model as planner + denoiser v.s. 2) separately trained planner + mask denoiser. The single uniform diffusion model converge slower in training and using DDPD sampler results in collapse in sample entropy. Using separate networks for planner and denoiser achieves results more close to SOTA methods in terms of quality v.s. diversity.

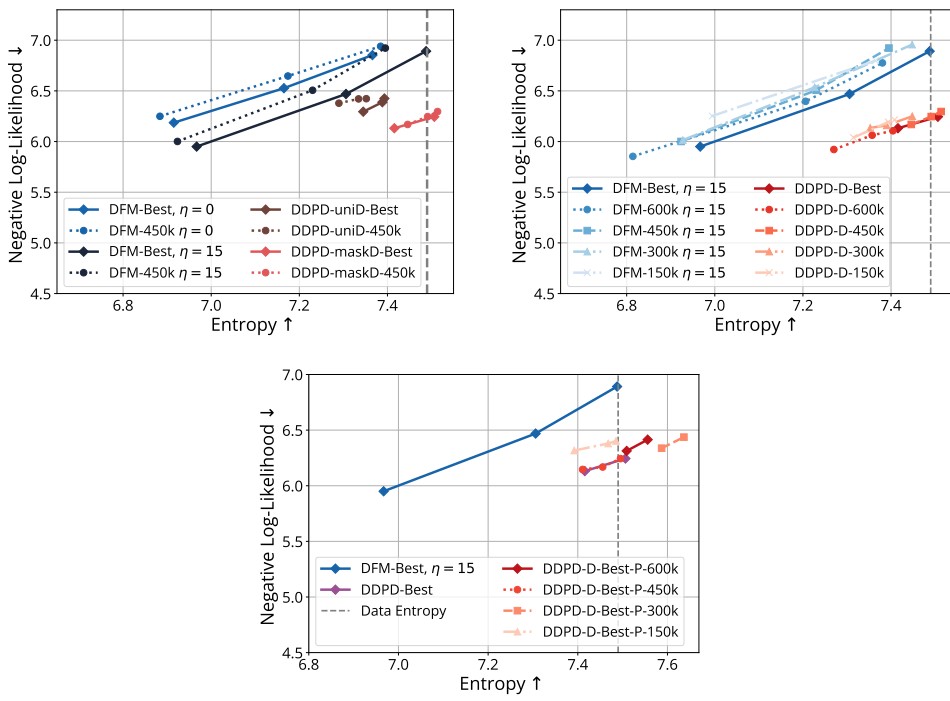

Figure 5: *Left*: Denoiser checkpoints at 450k v.s. 750k iterations. *Right*: Denoiser checkpoints at 150k, 300k, 450k, 600k, 750k iterations. DDPD is able to use an imperfect denoiser to achieve the same performance as the best possible.

mask diffusion denoiser performs better than uniform diffusion denoiser in terms the denoising log-likelihood.

## E.2 CONVERGENCE OF SAMPLING

We conducted experiments to see the convergence of sampling with regards to number of steps. The results are shown in Fig. 8.

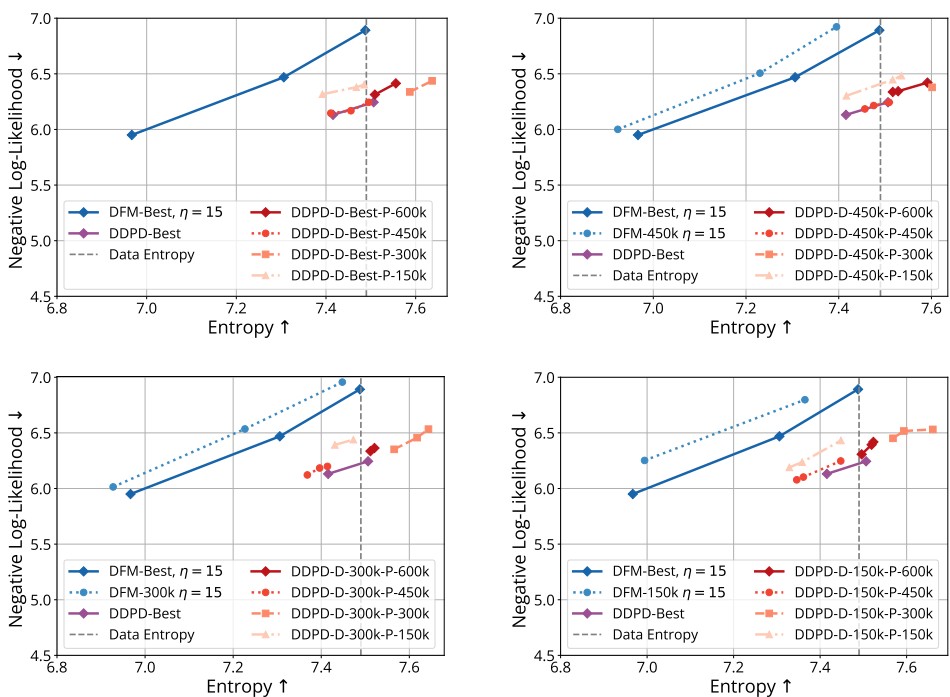

Figure 6: More ablation studies on pairing an imperfect denoiser with an imperfect planner.

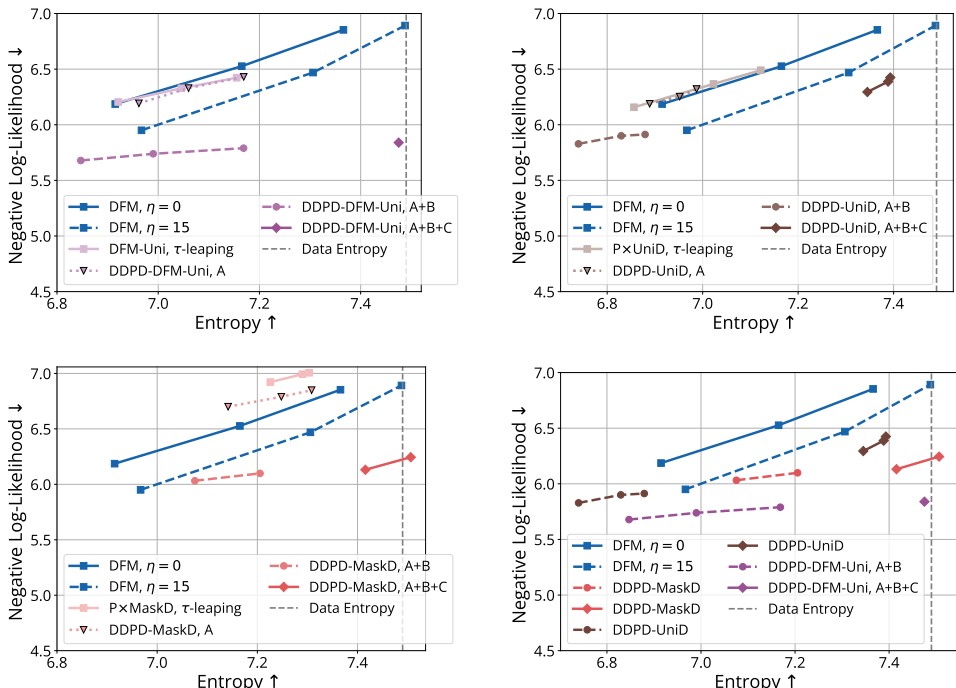

Figure 7: Ablation on introduced changes to discrete diffusion. A: Original Gillespie sampling. B: Time-adjustment based on the planner, continue sample until maximum number of steps is reached. C: Use softmax(`logit_if_noise`) instead of sigmoid(`logit_if_noise`) to pick which dimension to denoise next. The softmax trick makes the planning slightly more greedy than the original planning probability.

Table 4: ELBO terms computed on the test set of text8 in bits-per-character (BPC) with fully trained models. Denoising likelihood only evaluates the probability of correctly denoising, for Planer + Mask Diffusion Denoiser, a mask is first sampled according to the planner.

| Method | Rate Matching (BPC) | Transitioning (BPC) | Combined (BPC) |
|---|---|---|---|
| Uniform Diffusion | $\leq 0.0131$ | $\leq 2.252$ | $\leq 2.265$ |
| Planner + Uniform Diffusion Denoiser | $\leq 0.0176$ | $\leq 2.284$ | $\leq 2.244$ |
| Planner + Mask Diffusion Denoiser (given correct mask for denoising) | $\leq 0.0176$ | $\leq 2.226$ | $\leq 2.302$ |
| Planner + Mask Diffusion Denoiser (use planner-predicted mask for denoising) | $\leq 0.0176$ | $\leq 2.605$ | $\leq 2.623$ |

Table 5: Denoising performance in bits-per-character (BPC). Mask Denoiser v.s. Uniform Diffusion Denoiser. Note that those are not entirely comparable as ELBO terms for uniform diffusion and mask diffusion are different.

| Method | Denoising (BPC) | Denoising Accuracy at $\alpha_t = 0.85$ |
|---|---|---|
| Uniform Diffusion Denoiser | $\leq 2.063$ | 92.2% |
| Mask Diffusion Denoiser | $\leq 1.367$ | 96.8% |

Table 6: ELBO terms computed on the test set of text8 in bits-per-character (BPC) with imperfect models trained at $20k$ iterations.

| Method | Transitioning (BPC) | Combined (BPC) |
|---|---|---|
| Uniform Diffusion | $\leq 3.060$ | $\leq 3.076$ |
| Planner + Mask Diffusion Denoiser (given correct mask for denoising) | $\leq 2.854$ | $\leq 2.843$ |
| Planner + Mask Diffusion Denoiser (use planner-predicted mask for denoising) | $\leq 3.166$ | $\leq 3.155$ |

### E.3 OPENWEBTEXT

In Figs. 9 and 10, we measure generative perplexity of unconditional samples from GPT-2-small, GPT-2-medium, SEDD-small, SEDD-medium, DDPD-Small: Planner-small + SEDD-small-denoiser, DDPD-Medium: Planner-small + SEDD-small-denoiser. We also tested using sigmoid(`logit_if_noise`) and softmax(`logit_if_noise`) for planning. The difference is not as significant as in the text8 case. Using softmax(`logit_if_noise`) slightly increases entropy at the expense of perplexity. In Fig. 11, we find that DDPD using Planner-Small and SEDD-Denoiser-Small outperforms simply scaling up denoiser to SEDD-Medium.

### E.4 IMAGENET $256 \times 256$

We study the effect of planned denoising with an increased number of refinement steps in Table 9. The FID first increases and then converges. The inception score also improves with increased refinement steps and then converges. From the visualized samples, we can see that plan-and-denoise sampling is very effective at fixing errors without losing its original content.

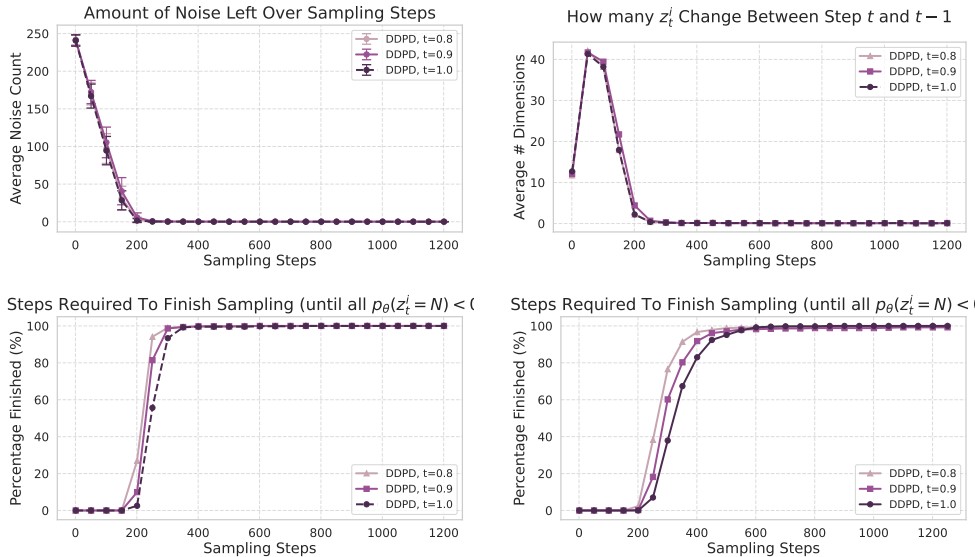

Figure 8: Convergence of sampling along number of sampling steps: *Left Upper:* amount of noise left. *Right Upper:* amount of dimensions that are predicted to haved different $z_t^i$ and $z_{t-1}^i$ between $t$ and $t-1$. *Left Down:* how many samples meet the criteria of stopping to sample for $p_\theta(z_t^i = N) < 0.05$). *Right Down:* how many samples meet the criteria of stopping to sample for $p_\theta(z_t^i = N) < 0.01$.

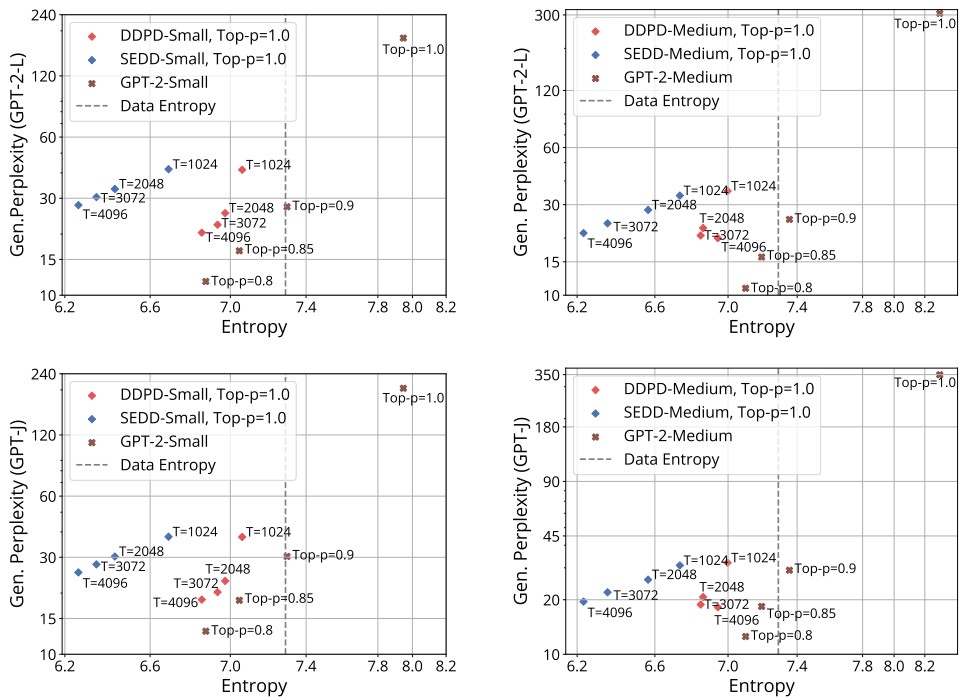

Figure 9: Using softmax(`logit_if_noise`) for planning. Generative perplexity evaluated with GPT-2 Large (GPT-2-L) and GPT-J: SEDD v.s. DDPD using the same denoiser.

### E.4.1 EFFECT OF CLASSIFIER-FREE GUIDANCE

We also studied the effect of applying classifier-free guidance in Table 10 for discrete mask diffusion. The effect of classifier-free guidance is similar to temperature annealing. In the case of no logit

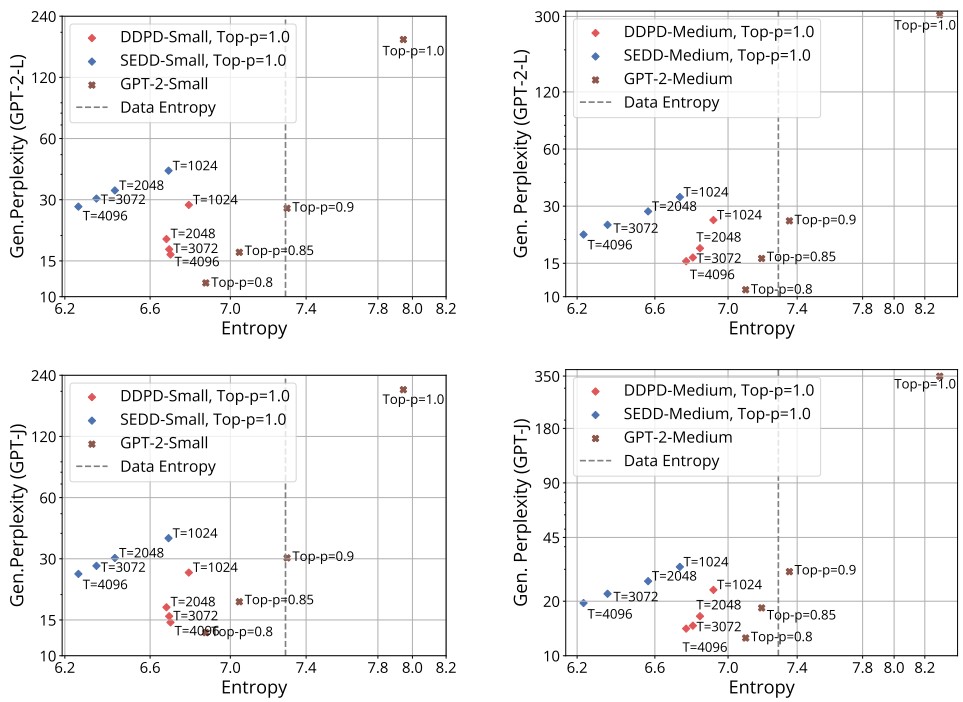

Figure 10: Using sigmoid(`logit_if_noise`) for planning. Generative perplexity evaluated with GPT-2 Large (GPT-2-L) and GPT-J: SEDD v.s. DDPD using the same denoiser.

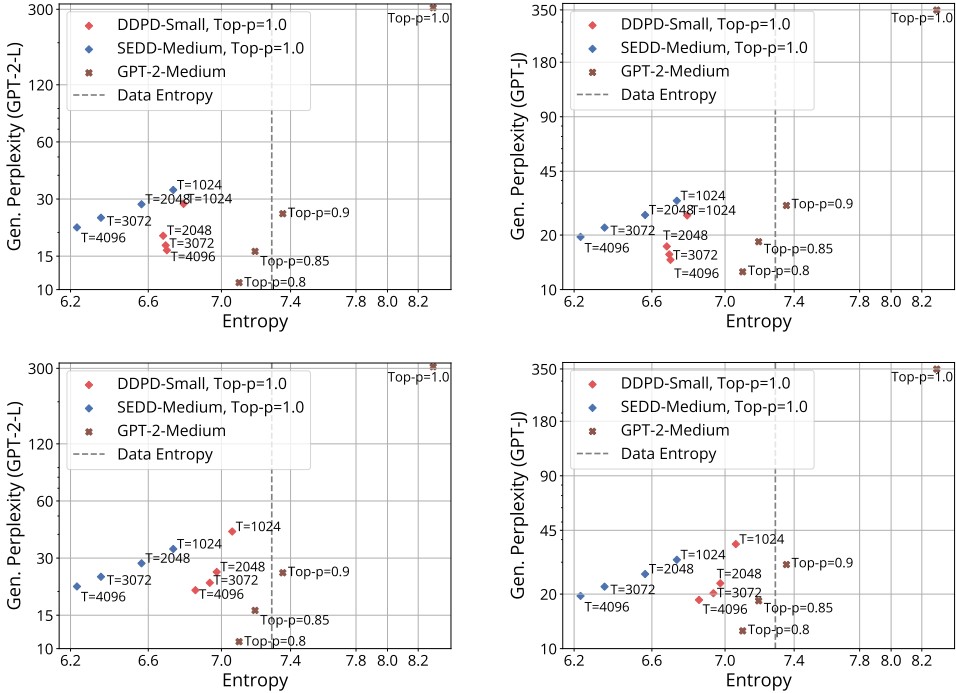

Figure 11: DDPD SEDD-small denoiser (90M) + Planner-small (90M) v.s. SEDD medium denoiser (320M) v.s. GPT-2-Medium (355M). DDPD with a smaller (less perfect) denoiser achieve better performance than simply using a larger (better) denoiser.

Table 7: Inception Scores ($\uparrow$) on ImageNet $256 \times 256$. MaskD refers to mask diffusion. The denoiser and parallel sampling schedule are kept the same as [45], without classifier-free guidance.

| | **No Logit Annealing** | | | **Logit temp** 0.6 | | | **Logit temp** $1.0 \rightarrow 0.0$ | | |
|---|---|---|---|---|---|---|---|---|---|
| **Steps** $T$ | MaskD | MaskGIT | DDPD | MaskD | MaskGIT | DDPD | MaskD | MaskGIT | DDPD |
| 8 | 33.56 | 199.83 | 149.98 | 149.28 | 271.73 | 201.67 | 157.19 | 249.86 | 213.03 |
| 16 | 39.36 | 248.88 | 178.17 | 179.85 | 281.73 | 173.48 | 164.01 | 263.47 | 185.25 |
| 32 | 43.30 | 266.17 | 169.49 | 200.33 | 281.36 | 156.22 | 170.73 | 268.88 | 158.14 |
| 64 | 45.06 | 274.56 | 160.74 | 206.06 | 281.14 | 146.27 | 171.62 | 269.45 | 145.95 |
| 128 | 45.56 | 276.45 | 152.61 | 210.27 | 278.88 | 138.55 | 142.40 | 272.73 | 137.19 |

Table 8: ImageNet $256 \times 256$ generation results

| Method | FID $\downarrow$ | Inception Score $\uparrow$ | Model size | # tokens | codebook |
|---|---|---|---|---|---|
| Taming-VQGAN [12] | 15.78 | 78.3 | 1.4B | 256 | 1024 |
| RQ-VAE [23] | 8.71 | 119.0 | 1.4B | 256 | 16384 |
| MaskGIT-VQGAN [8] | 6.18 | 182.1 | 177M | 256 | 1024 |
| ViT-VQGAN [43] | 4.17 | 175.1 | 1.7B | 1024 | 8192 |
| MAGVIT-v2 [44] | 3.65 | 200.5 | 307M | 2048 | 262144 |
| 1D-tokenizer [45] (annealing tricks) | 4.61 | 166.7 | 287M | 128 | 4096 |
| DDPD-1D-tokenizer (w/o annealing tricks) | 4.63 | 176.28 | 287M + 287M | 128 | 4096 |

annealing, it helps mask diffusion to achieve much better FID score. DDPD achieves similar FID scores but much better Inception Scores (which means higher aestheticlity). When logit temperature = 0.6, the inception score of Mask Diffusion is greatly improved, but the composition of logit annealing and classifier-free guidance leads to a worse FID score due to less diversity. The same applies to MaskGIT.

We note that the interaction of various sampling heuristics, particularly in configurations that incorporate logit annealing and classifier-free guidance (CFG), is intricate and nuanced. As detailed in Yu et al. [45], the best FID scores for MaskGIT are achieved using additional hyperparameter adjustments, specifically inflating the logit and confidence temperatures to 3.0, which are then linearly annealed to 0.0. These unconventional settings are tailored to mitigate the loss of diversity typically caused by the combination of logit annealing and CFG. However, while this approach improves FID scores, it adversely impacts inception scores, underscoring a fundamental quality-diversity trade-off.

To address this trade-off, Table 11 presents results for MaskGIT with the same configurations as Yu et al. [45], where sampling includes logit annealing and logit and confidence temperatures are initialized at 3.0. Interestingly, the initial tokens sampled under these higher temperatures (which tend to introduce more errors) are later corrected during the annealing process, allowing the decoder to reconstruct coherent and high-quality outputs. These techniques are specifically optimized for mask diffusion with CFG, but do not generalize directly to DDPD, which operates using uniform diffusion. Following this insight, we applied the same configuration to DDPD as an experiment (Table 11). While this approach improved inception scores, it worsened FID scores, further highlighting the quality-diversity trade-off. Upon analysis, we observed that the DDPD planner proactively identified some of the initially generated tokens (under high-temperature settings) as noisy, as they deviated significantly from the original distribution. These tokens were subsequently corrected in later denoising steps, leading to improved visual quality at the expense of reduced diversity. This behavior indicates that DDPD naturally balances quality and diversity differently from MaskGIT.

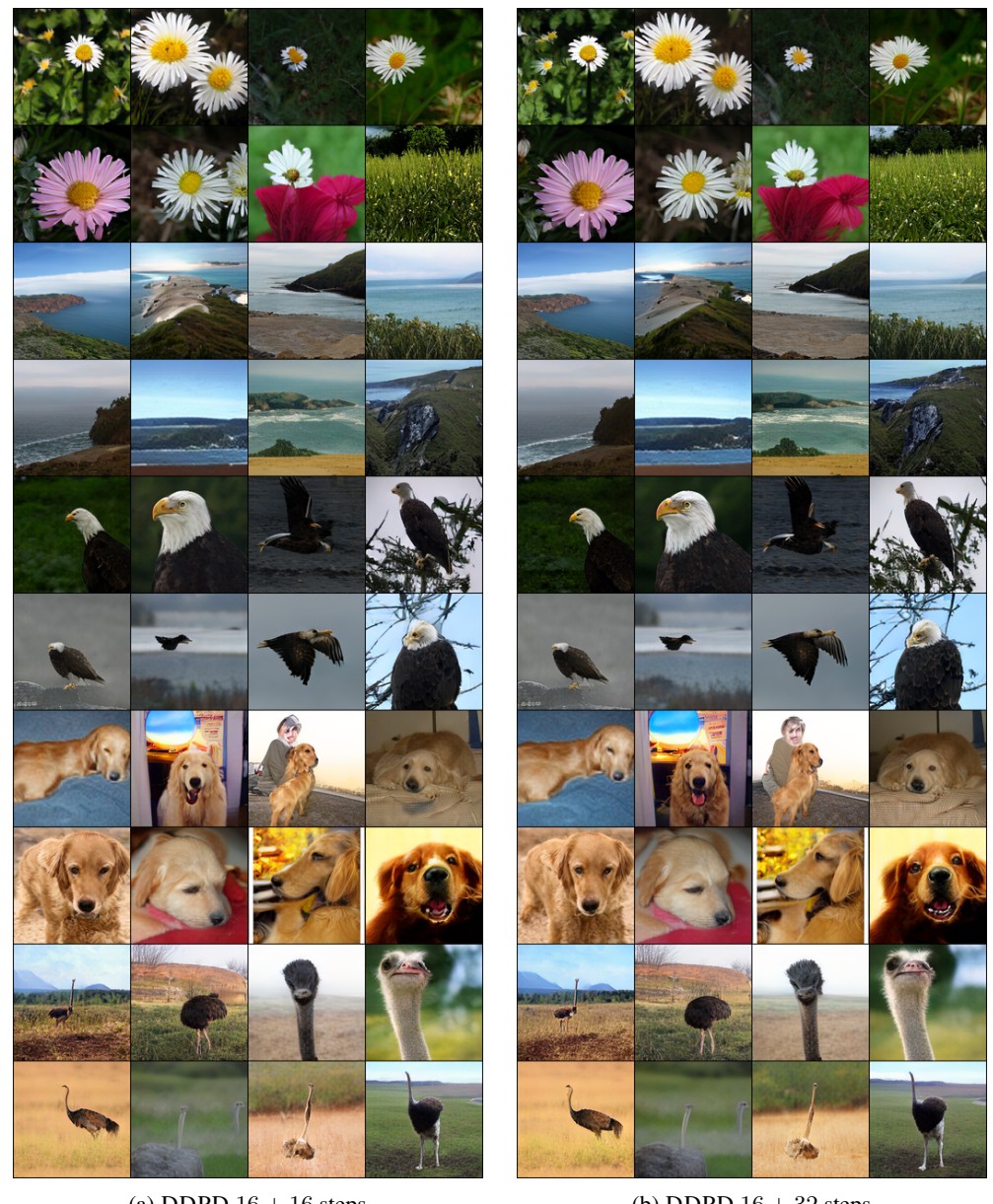



(a) DDPD $16 + 16$ steps          (b) DDPD $16 + 32$ steps



Figure 12: DDPD No Annealing, increasing number of refinement steps. The added refinement steps act as "touch-up" to improve the aesthetic quality without losing its original content.

Table 9: FID Scores on ImageNet $256 \times 256$. Increasing the number of refinement steps.

| Refinement Steps $T$ | Base steps $T = 8$ | | Base steps $T = 16$ | |
|:---:|:---:|:---:|:---:|:---:|
| | FID $\downarrow$ | Inception Score $\uparrow$ | FID $\downarrow$ | Inception Score $\uparrow$ |
| 8 | 5.12 | 178.17 | 5.12 | 161.17 |
| 16 | 4.92 | 187.59 | 4.75 | 169.49 |
| 32 | 4.93 | 192.93 | 4.63 | 176.28 |
| 48 | 4.94 | 192.99 | 4.71 | 176.22 |

Table 10: CFG scale 2.0, Logit Temp 1.0 or 0.6. MaskGIT anneals confidence noise temperature from 1.0 to 0.0. FID/Inception Scores ($\downarrow/\uparrow$) on ImageNet $256 \times 256$.

| Steps $T$ | No Logit Annealing | | | Logit Temp 0.6 | | |
|---|---|---|---|---|---|---|
| | MaskD | MaskGIT | DDPD | MaskD | MaskGIT | DDPD |
| 8 | 8.44 / 145.36 | 6.89 / 337.37 | 7.19 / 323.79 | 6.37 / 320.48 | 13.00 / 400.23 | 7.45 / 333.28 |
| 16 | 5.30 / 187.24 | 9.91 / 391.98 | 6.28 / 329.40 | 8.17 / 365.30 | 13.93 / 407.12 | 5.73 / 319.52 |
| 32 | 4.14 / 215.88 | 11.49 / 408.73 | 5.58 / 322.67 | 9.06 / 384.50 | 14.16 / 406.19 | 4.98 / 315.14 |
| 64 | 3.82 / 223.22 | 12.04 / 409.16 | 5.16 / 323.62 | 9.51 / 388.71 | 14.28 / 405.01 | 4.66 / 311.82 |
| 128 | 5.02 / 180.79 | 11.55 / 396.81 | 4.63 / 313.09 | 8.31 / 361.80 | 13.76 / 388.04 | 4.39 / 304.92 |

Table 11: When using the same hyperparameters as in [1] (8 steps are not enough for DDPD to converge). FID / Inception Scores ($\downarrow / \uparrow$).

| Steps $T$ | MaskGIT (FID / Inception) | DDPD (FID / Inception) |
|---|---|---|
| 16 | 2.48 / 284.20 | 7.93 / 359.69 |
| 32 | 2.74 / 299.53 | 6.11 / 343.63 |
| 64 | 2.86 / 309.62 | 5.11 / 328.56 |
| 128 | 2.93 / 205.26 | 4.63 / 313.09 |

E.5 Noise estimation error using independent noise output $p_\theta(z_t^d|x_t)$

We tested the assumption made in utilizing a pretrained mask diffusion denoiser by sampling joint noise latent variables using independent marginal prediction from a transformer for $p(z_t|x_t, z_t^d = N) \approx \prod_{d' \neq d} p_\theta(z_t^{d'}|x_t)$ in Table 12. We observe that the assumption holds almost perfectly in language modeling such as OpenWebText. On character modeling task text8, the assumption also holds most of the time, especially near the end of generation, but it is more complicated than word tokens due to a much smaller vocabulary. This is also discovered in Table 4 where we observe the two-step sampling introduces approximation errors and hence makes the log-likelihood for denoising lower.

Table 12: Accuracy on Mask Prediction for text8 and OpenWebText at fixed times. Mask accuracy measures if the independent sampling matches the joint noise variable values. Almost deterministic measures the assumption $p(z_t^{\bar{d}}|x_t, z_t^d = N) \approx 1$. We set the threshold to be $\text{logit.abs}() > 3.0$.

| Fixed Time $t = 1 \rightarrow 0$ | text8 | | OpenWebText | |
|---|---|---|---|---|
| | Mask Accuracy | If Deterministic | Mask Accuracy | If Deterministic |
| (Data) 1.0 | 0.9988 | 0.9975 | 0.9999 | 0.9997 |
| 0.95 | 0.9915 | 0.9864 | 0.9985 | 0.9960 |
| 0.8 | 0.9623 | 0.9238 | 0.9943 | 0.9851 |
| 0.6 | 0.8784 | 0.6789 | 0.9847 | 0.9585 |
| 0.4 | 0.7416 | 0.2261 | 0.9679 | 0.9100 |
| 0.2 | 0.7402 | 0.2125 | 0.9476 | 0.8466 |
| 0.05 | 0.8878 | 0.4800 | 0.9599 | 0.8817 |
| (Noise) 0.0 | 0.9465 | 0.5974 | 0.9975 | 0.9906 |

# F  GENERATION EXAMPLES

## F.1  GENERATED SAMPLES FROM MODELS TRAINED ON TEXT8

We compare samples between DFM and DDPD. For DDPD, we include samples from three models: 1) DDPD-DFM-Uni: planner and denoiser from a single uniform diffusion denoiser model $p_{1|t}^{\theta}(x_1^d|x_t)$ using Eq. (10) and Eq. (11); 2) DDPD-UniD: a planner network $p(z_t^d|x_t)$ and a uniform diffusion denoiser network $p_{1|t}(x_1^d|x_t, z_t^d = N)$; 3) DDPD-MaskD: a planner network $p(z_t^d|x_t)$ and a mask diffusion denoiser network $p_{1|t}(x_1^d|x_t, z_t^d = N)$.

---

Samples from DFM, $\eta = 15.0$, Temp 0.8:

```
are being damaged downtown plus the roads that are historical image map roads
roads through hong kong shin te kwun mun avenue tansua tai tonlin gouhan and
mengtusam there are several main freeways at the same station in the hong
kong through caches on the

ble everything basil mark to one nine two nine murmour about mirrored action
making me worth one nine three nine lecture bird man one nine three nine
the voice of law one nine three nine everything revived one nine three nine
people with fears one nine fou

n eight one nine nine two stemming from people s disaster one nine nine zero
one nine nine five author john chemerzi wakis pbs troupe witnessing impact
report one nine eight one jone bethy hopkins place wiley and jackson campaign
begins one nine eight eigh
```

---

Samples from DFM 2×, $\eta = 15.0$, Temp 0.8:

```
ero dinidol three one zero five two four three acritrine zero six zero
eight four eight zero zero two three acetic acid one zero one seven l not
aspirin in vinnol c two three nine hyproxyphenol zerolin references mda r two
virginia department of public saf

he often shared at least some of these suggestions the priesthood of all
the people of sparta hemischeres your father god and lord bound him among the
priesthood of the lord with such interference preaching the probes of tribute
and i cannot believe they w

in etc time relief belongs to the preparation of funeo the time silenium
and platarin the same rolled taste preparation by cooking wine from the gum
is sped back the wine that is produced per pell concentration is nine two the
taste is either rosin brac or
```

---

Samples from DDPD, Planner + Denoiser decomposed from a single uniform diffusion denoiser model $p(x_1^d|x_t)$, Temp 0.8:

```
dickey morris sam morris scott del man sam del man sam del del man brenda del
simon simon fred rogers gregg dickey david dicks marcus dusshinsky douglas
dickson steven dick douglas hartman harvey dickson scott kelly hartman mike
duncan daniel harvey micha

e prochet pink or peanut proc pinky proche pink pigmy pig proche pigmy pigmy
pig bear frog hornit horna horn wagon hornita wild fish hornit wagon griffon
germanic florahornit horna ostrich wild fish flora horna wild island chicken
horna horn winters winter

park kitchen comfort fort gorge fort hills castle bay hills lakeland fort
hamilton state park the park on ground hill bay forest reed brighton park
stanley innocence small park protector hillfort edgeside statue greenwood
woodfort st alban st columba stree
```

Samples from DDPD, Planner + Uniform Denoiser, Temp 0.8:

```
reported however in one eight six nine that a natural carnivore would be a
bad man daily utilized his newspaper the best lighting embeddings of quiner
sun warm and winter and warms france s illegal bubbles first said lighting
with quiner is greatly import

well the composition of the entire borough was to be by the boiling of free
communes of columbia were once filled and defined without separate antiphers
and were antiphers each of these twin princess mournings made the term death
as an antipher as a defin

in the work that is edward damascus across the cross from the flame of my
career and the king is why i like others here is there s a great aspect die
die crossing that precedence of the star die literary die an aspect of murder
may contact or rescue those
```

Samples from DDPD, Planner + Mask Denoiser, Temp 0.8:

```
of roy despite this in the franchise he was uncredited to put on club
chatterhead big morocco theater and in the winston team hockey shot out for
snap five sidekick notable player don allisto mike henning puncher jack may
founder of roy puncher five four f

lithography logoliths littoral nonfiction lilitus confusion lit little
dragon littorius love eye love crawlin love utopia lolita popula prose oracle
populae populae anticharia prophecy leonida popula lepheus mycenaean super
super dendron carbon lover dend

rrorists criminals in gold the timing expressing only insisted by the endings
of them rising six to ten days a population of one thousand guessing holding
potential risking dangers see falling and plasticizing goodman father of the
town of guam effect only
```

F.2  GENERATED SAMPLES FROM MODELS TRAINED ON OPENWEBTEXT

In general, samples from DDPD demonstrate a better ability to capture word correlations, leading to greater coherence compared to those generated by SEDD. However, both methods exhibit less coherence in longer contexts when compared to samples from GPT-2 models.

---

Samples from DDPD-Planner-Small-Denoiser-Small, 4096 steps:

```
tornadoes | Space.com
Read more:The disarray has reached an end, dragging the US political system into its turbulent
period.
Today looks to be the day of reckoning for Washington.
Facebook Twitter Pinterest Men and women make an attempt to enter the Capitol building, which is
part of the US State Department.  Photograph:  Nicky Boyce/AFP/Getty Images.
While there has been some infighting of late, the sometimes-jaded new US Congress has also
been shying from the usual trappings of parliamentary checks and balances, particularly on the
appointment of a secretary of state, and on immigration, as the Senate yesterday voted to vote ''no''
to a bill.
In the new session, Congress looks to seize the opportunity to form a new government, replacing
the old with a new one, something that has been done in other countries.
Work will start in April on a new law that will change dramatically the political landscape across
the country.
The law was amended more than half a dozen times over the two-week period, and will be announced in
advance of a private event hosted by Mr Obama.
The law creates a new legal system for states.  In the US system, it treats state and local
officials as the representatives of the people, with the federal government, including the
president, in the process of forming the new government.
Interior Senator Joe Lieberman, chairman of the so-called federal government lobby group, said it
''is time'' to come up with a new government.
''If this is the rule of law, that's not the way we've done it.  We have to think about that.  We
believe the result will be good for the people and the country,'' he said.
In the past, Mr Durbin has been vocal in his desire to form new government.  Firstly, he wanted
the bill to take effect in 2008, then he vetoed the amendment in 2010.
He was more outspoken in his desire to legislate further, just a week before the new session,
by arguing that lawmakers who failed to vote on the amendments to the law should have failed to
attempt to form a new government.
French foreign minister, Laurent Fabius said:  ''We welcome the formation of a new government and
we look forward to election of the new president and taking on the important task of forming a new
government,'' Fabius said.
''The announcement of the president's resignation from office, is seen as a sign of the election of
a new secretary of state.  Not a vote of confidence.  A vote of confidence and it will happen," he
added.
Earlier, Irish Labour Party leader Mairnín O'Naughin-Sullivan said the country's attention was
being diverted from immigration, saying:  ''I do think this issue is not on our agenda at all.  It
is only a short period of time, and we have to come up with a new government, especially in the
course of the new session.''
She added:  "I think it is important for the president and the US government, to not form a new
government in a different way.
''And we need to work on making amendments to the law so the US government will not form a new
government and not raise the specter of a new US government.''
Facebook Twitter Pinterest Concerns about immigration are rising in much of the country.
Photograph:  Marjorie Nougou/AFP/Getty Images.
Politics and constitutional issues
Ewen-Scott Brown, head of the US government under the Obama White House, had successfully pushed
for legislation to create a new legal system for states rather than the Republican-led federal
government, 10 years ago.
In the biggest political move in American history, Ewen-Scott Brown has described the issue as a
constitutional issue.
"I have said that the way it relates to a secretary of state position is no different than the
Secretary of State position.  It has nothing to do with the Constitution,'' she has said.
Deputy members of the administration, which included International Trade Secretary Michael Froman,
and Justice Secretary Carole Wray, formed a new congressional task force.
But by the time at which Mr Obama was elected the US president, public opinion had tumbled in the
opposite direction.
He had introduced a new immigration bill and became the first president ever to get the
immigration bill passed in the US Senate, but the House refused to act and he resigned from the
Senate in December 2011.
Health care

Mr Brown's position as leader of the nation's Republican Party has expanded under the Obama

administration.  He was one of only three members of the House to pass the health care reform

bill in 2009-10 and took a break from the Senate as head of the Department of Health and Human

Services.
```

Samples from SEDD-Small, 4096 steps:

to change,'' the second-in-one Cabinet minister, political correspondent Oliver North, told CNN on Sunday. This was the first such rally of the party control's campaign. ''Time to change course is now. A new politics will begin in 2020.''
Even party leaders and prominent Labour figures are alert to this shift, including Boris Johnson who has left the party for the first time since a rival candidate's nomination for president of the Republican National Committee, pledging in his Saturday speech that he had to work for the end of Thatcher's reign.
The media, meanwhile, had predicted that former minister Margaret Thatcher would not vote for her and backed her in the current right-handed coalition with Mr Miliband and attacked Labour leaders as without a party that could return them to the prime minister and at odds against across-the-ground austerity, which she personally has never said he wanted.
Read more from CNN on Twitter
In the absence of the party, Johnson has warned: ''We've kind of destroyed our country if we don't talk about our future, so David Miliband's attempts to replace our leader are selecting those who support Labour, who need to match voter turnout and are a real threat that could have that in 2020."
Corbyn's say for Corbyn
As Jeremy Corbyn addressed an exile party convention in West End, London on Saturday, Corbyn presided over the Tories in London who are in recent days leading the polls overall in the party. Corbyn, who used the black vote in his pool of 10 MPs to win theelections, also stood for Corbyn at a packed rally outside the Democratic Central party.
After London conference on the change in Labour's management of the Labour Party, Mr Corbyn said earlier this month that ''the greatest person ever to decide Sir Jeremy's former Labour leadership, the first woman to decide her leadership in more than 20 years'', and called Jeremy Corbyn at that conference in 2012 a ''remarkable individual''.
Mr Corbyn said in Westminster: ''I condemn the hate to your name'' and cited him as being tough on racists, who the Liberal Democrats say are just white men - shifting from white middle men, who already put 60 per cent and have stood down since the general election earlier this year.
Mr: "I condemn the hate to your name on the unWhiteList of #LabourCan. Yvette Caron... no one is racist" https://t.co/JkRs5ICHbA8 - Nicola Davidson (@GLGLa) September 14, 2017
David Cameron, the Corbyn leadership candidate, said having left the Carkey campaign opposed the real threat of racism. ''I'm a fan of this movement,'' he said after the press release, describing it as he'd like to see in Manchester, which is Britain's third biggest city. ''We've got it. 'Get you a ballot paper, and it will be a man, with no woman, with no women. It will be you, so step up and vote for it for the first time.''' Mr Corbyn appears to have been saying ''the Tories should stop being down against discrimination'' and ''The Greens should go against it.''
Locations for a Blue Brown memorial
Earlier this week, the Conservative parliamentary party released another statement about protests for the death of Mrs Brown, a young woman who carried opposition to ''a free-gout programme government'' into the black market. Brown and tens of thousands of pensioners walked to the streets after her death.
''The great political path forward over the past 60 years has come without Labour leaving Blue Brown, and including any ones involved in that distinct occasion,'' it said in a statement. Meanwhile, the Tories said in a statement she'm ''not left of politics'' and ''strong in my convictions.''
She stood for Jeremy Corbyn during Saturday's campaign: ''Democrats, in the last decade have been pushing their agenda and been trying to right the say they are complicit in sexism and racial inequality. The rest of the Democrats and Republicans have been playing similar roles in America's history for decades. I'm optimistic that a beacon for liberalism can be brought back into this country.''
But seeing Mr Corbyn as a father figure on the back of his election also moves her away from the party's platform as the dominant party in politics. Ms Corbyn has spent time across the country over the past 16 years and have seen public events as a means by which she and Corbyn have won them at one point or another.

Hillary, in her two years as Labour's founder and parliamentary frontrunner, named race not a factor in her election victories, but a core legacy

Samples from GPT-2-Small, 1024 steps:

Because he knew how dangerous it was to get there.  The two had been walking home, and when he came to, he saw two people standing outside the tree, each with a large knife and rifle.  The murderer stabbed them in the neck with a bladed weapon, and the two of them both died.
Why should I look at a movie when I could just see the actual movie?  I'll give you a clue.  The visual effects supervisor at the time, Donnie McCarthy, was not particularly interested in the movies, and decided that we should just get along with these actors, which is what this story follows.
When Charlie, the first person Charlie can actually meet, is brought up to him by his mother, he has to help her and reunite her with her children.  Now, she isn't about to be in a home, but she wants him to be reunited with her.  She wants Charlie to be safe, that he and his mother are safe, and that he doesn't need to go through so much trouble and regret for him and his mother.  It's a classic start to this story, so I'm not going to lie.  It's the complete opposite of what was made out there.
But when Charlie was brought up, the whole story about how to build his own life and stay in the state of Kentucky, he didn't need to do that.  He needed to make it up to his family.  In that way, he's made everything right.  That is, for him and his family.
It all started with what I first described in The Birth of a Nation, when a high school student named Emil Fowles tells him, "You can't be that way." He's no longer considered a teenager by the state, and he lives in Louisville, Kentucky.  He meets his father and father's parents, and he helps himself to his bag and goes to the Best Buy.  He keeps changing diapers.  He helps his uncle, who doesn't have money, or himself, who doesn't have a credit card.  He gives up cigarettes, and he starts to tell his dad about that, about wanting to make his own choices, that he's finally in a better place.
He starts this story with how Emil told him to help himself.  The world's aware of Emil's freedom, and now he can do his own thing, whether that means a house, a car, a school, or even helping his grandmother.  He then tells his mother, who is some time in the past, that he's leaving his country.  He has to, he says, to get out of here.
The rest of the story is very much about a woman who's trying to help herself.  In this story, Emil is really going to his mother.  That's something he doesn't want to do, but it's not what we're seeing in the movie.  It's more of his time with her, with his father, and he tries to get out of here, but at the end of the movie he tells her that he's going to make a big deal out of the fact that she's going to miss him.
It's a kind of adaptation of the "Selfish Girl" from the original series.  You don't get to see all the backstory and how these characters were raised, but the emotional moment is in the very end where Emil is helping his mother and now he's in a country where the government is corrupt, and his mother's toiling away, trying to make things better for her children.
To me, that is kind of my favorite part of the film, because it's almost like, "What if my mom is leaving me and I'm going to have to fight with her every day?" It's very rare, because I have been around, but I like being in a country where I have to fight for my kids.  And it's very very similar to the movie that I was involved with in The Birth of a Nation.
I was watching The Birth of a Nation in London back in the '60s, and I'm not sure if it was actually a true sequel or a remake, because I can't say, "We're not doing this because this is what we want to do." That's what I liked about that movie, because I knew that this world would always be different from any of the other movies in that time, and the people who were created in that world.  That was one of my most favorite aspects of the movie.
I mean, that's the thing about movies like The Birth of a Nation, and I thought, "There's something about it that really makes it, and that, at the same time, really pushes me to think about what I can do in this world and how I can make a world of my own." It's a perfect illustration of that.

I saw that movie a lot, and I knew that this was going to be one of the most beautiful movies of

all time.  It's

Samples from DDPD-Planner-Small-Denoiser-Medium, 4096 steps:

The app also allows you to select which channels to watch every time you open the app so you can watch those channels on Apple TV (iPhone and iPad only) or on your phone.
For example, HBO Go, Showtime, AMC and other channels don't have to be on Apple TV because the Roku app can be paired with Apple TV to watch them, according to Scott Robinson, vice president of business development for PlayStation Network, Inc.
7. The Appflix
With this app, you can connect your TV remote to your phone and watch new shows and movies that are being added to your TV collection, according to Amazon. The Appflix app lets you watch those channels on your phone without using the remote.
It will work well with the new Apple TV remote when it's released by Apple.
Roku
This was supposed to be a companion app for DirecTV.TV, but it's now being used by Roku.
You can't set up your Roku as a DVR if you have it on your set-top box. But Roku owners can use it as a hub for their TV so that you have multiple channels and apps so you can look for the best content available.
It also provides you with a split-screen streaming feature that allows you to watch multiple channels.
The app isn't connected to your TV if you have it on Apple TV, but you can use it on your Roku TV, or a Chromecast, Amazon Fire or any Android device.
Looking back at the apps released this week, this may just be all it takes to get some of the content from Roku on your TV.President Barack Obama speaks Thursday in Washington, accusing Moscow of undermining the alliance. (Photo11: SAUL LOEB, AFP/Getty Images)
MOSCOW - A German lawmaker said President Barack Obama should ask the United States to spend more money on military support and training in the region to prop up the Ukrainian government in the war against Russia.
In the end, NATO will have to stop the Russian aggression in eastern and central Ukraine, according to a statement from the German parliamentarian Robert Appel on Thursday from the alliance's headquarters in Vilnius, Lithuania.
Specifically, he said Washington needs to ask for more weapons and military assistance to fight the pro-Russia forces fighting in eastern Ukraine. That is a position some NATO leaders have not been comfortable with since the U.S. has called for military help.
The comments from the German lawmaker were put out in response to Obama's announcement this week of his plan to send more arms to the separatists in Ukraine and his call for the United States to join NATO.
U.S. and European leaders condemned the victory of the separatists in this month's elections.
"It is deeply disturbing to know that Ukraine elects a leader that the U.S. appreciates and shares with the U.S. government," said Vice President Joe Biden.
"The president was elected on the very platform that legitimates aggression in Ukraine," he added.
The U.N. Security Council is meeting to consider new ways to confront Russia, and Obama has urged President Vladimir Putin to pay attention to the issue.
The U.S. has said the move is a "serious threat" to the alliance.
Obama said he is "very serious about our security" and that while the alliance is seeking help from Moscow, there are limits to that.
"If the pattern of Russian aggression continues, it places NATO, the alliance, and ultimately the security of Europe and its allies in danger," said the president at a Thursday news conference.
"The Europeans face a security crisis," said Secretary of State John Kerry. "This is a serious security crisis in eastern Europe."
NATO in support of Ukraine
Appel focused his statement on the "turbulence factor," citing NATO's relationship with both Russia and the way in which counterinsurgency operations were waged against the Soviet Union.
In addition to the support of the European Union, he said, the UN Security Council is required to act against Russia.
"Ukraine is not a NATO member, and Russia as well, is not a NATO member," he said, referring to the alliance's membership under article 5 (a) of its constitution.
In February 2014, the Ukrainian government declared itself to be a member of the European Union, making it a NATO member.
Tipping away at the NATO alliance?
"You can raise the threat level with Russia," retired Adm. Mark Green, the top commander of the American forces in Europe, said on CBS's "This Week."
He said that if Russia ousted President Viktor Yanukovych's newly elected government, it would have to be supported by other members of the alliance "through the use of force," in such a "persistent way."

Green said the actions of the separatists

Samples from SEDD-Medium, 4096 steps:

said.
''To me, that's something that is going to happen to any of us.  You never know who is more
prepared.  If you give, the guys are up and out of there.  You know, if you get two guys off and
they're frustrated because they feel good, I don't think it's going to make the team any better.
''I'm not comfortable, and I should be, I'm competitive.  I'm just blocking every shot the way I do
everything I do.''
And that he plans to get even more aggressive out there.
''With each different knock to my body that happens against somebody in Tier 1, or better, I'm going
out there and pushing myself,'' Jordie said.  ''I appreciate that, and the better I get, the greater
an advantage it will make me feel because the schedule is better, so we'll see what happens.''
Since the injury finally more than a year ago, you can be sure he has worked slowly to get better,
but he has gone through his upsides as well.
''Initially they were pretty brutal before they happened, then they were pretty painful,'' McKenzie
said.  ''But it wasn't so bad.  I was figuring them out later.''
''He knows how to improve and get tougher,'' O'Regan added.  ''He's still a long way to get there, but
I don't think there's anything he's done before.
''This is probably the time that I worry he's going to have an issue like this from the off-season.
 "We're going to go to another one-on-one contact test on his body to see what happens - that's
going to be the only way that we can trust,'' O'Regan said.
As of now, McKenzie is still trying to determine how he is going to play his best.
''I'll put myself to the tests, but everyone has the same struggles, and I'm really just bad with
losing,'' he said, with a laugh.  ''That's when you're dealing with it you can't do anything else.
''That's how I'm kind of living.  I do need to deal with that, but it's always done it for me.  If
it wasn't hard, if I wasn't understand, I wouldn't be able to.
''The way I am now, I can walk six feet.  I'm kind of thinking that's all bad, but I don't know.  I
understand that's taking so much of your time and your trust.  I'm not Mr.  'The Ugly.'
''I'm not that kind of guy, and that's why I think I'm not as involved in camp as I need to still
be.  I'm damn hungry and I'm going out to work every night.  That's why I don't plan on staying
home like normally would.''
McKenzie said having the kind of type of recovery that he really wants was worth the scare he had
when he received from the start workout the morning the Morys started the trip.
"It kind of makes you sick thinking that," he said.  "That was an intense workout.  You know?
When you're dealing with that you can't do anything."
McKenzie said he underwent more than a massage and he continues to play a role in his body every
single day.  If the setup there was not successful for him, he knows that he won't be getting the
most of his time behind them from now on.
All told, McKenzie has a full season off his injury to be recovered and back into full-time hockey.
He still knows he is far from unstoppable.
"It's frustrating but I still am," McKenzie said.  "I know this has to give, and I can't give this
up right now."
But even then, McKenzie is at a loss for words, or really any words at all.
''In the dreamy way, you know, deal with that,'' he said.  ''I always keep that in mind.  I always
have.  But when it comes down to it I just know that I'm the best at what I do when I work through
it.  I just work harder and get better every day.''You'

Samples from GPT-2-Medium, 1024 steps:

A senior Russian prosecutor, summoned by President Vladimir Putin for questioning about his alleged links to the controversial bitcoin exchange Mt. Gox, has revealed he used the ill-gotten gains from the alleged theft of some $230m in the troubled digital currency to buy a holiday home in California.
Nassim Mikheev said he spent his weekend in California to buy a $60m three-bedroom house with a 6,900 sq ft kitchen, 600sq ft living room and 18ft ceilings and walls on luxury property in Malibu.
On Wednesday, the judge presiding over Mikheev's preliminary hearing asked him about his finances, telling him he had invested "roughly 200m roubles (£179m) in Bitcoin," using the digital currency to purchase a residence.
"I would like to make the remarks that when I spent money in Bitcoin and put it into my property, the property value increased tenfold," Mikheev replied.
The price increase, he said, came from Bitcoins used to buy the home, with a one-year buyer's contract for the property saying the currency had doubled in value.
Mikheev's lawyer said he intended to mount a defence of his client's conduct. "As we have already explained, we expect that even if this amount was a mistake, it was not justified," said lawyer Anatoly Semenov. "As he is not a defendant, we will not ask for any charges or a plea."
Mikheev is currently under house arrest. Russia's Federal Anti-Money Laundering Service (Banske) has named him and has requested access to all bank accounts in Russia and Kazakhstan, which cover at least half a billion dollars.
Mikheev said he first set up Mt Gox, a controversial bitcoin exchange, in 2009. There, the chief operating officer, Sergey Karpeles, said he bought about one million bitcoins and used the money to pay people for work they did for him and other members of the Bitcoin community.
After his father, Mikhail, the first owner of the Mt Gox company, sold the business to a second company, he became disillusioned and had Mt Gox hacked. At least $230m was allegedly stolen. Karpeles eventually left the company and told media that he would be "really happy to get it back, but in the meantime I can't take it, so what do I do."
His father then moved to Luxembourg, where Mikheev now lives. A group of investigators now believe he stole the money through hacking, and the company has since been under siege by authorities. The case remains under investigation by the German tax authority.
Citing the US criminal case against Mt Gox founder Mark Karpeles, Semenov said that "in this case it would be necessary to name and shame the cybercriminals who in exchange for stolen bitcoins gave them their services to discredit this investigation, of which the Mt. Gox CEO is a key figure."
He added: "If at the time of one of his actions the prosecutor decided to introduce them for questioning, this decision would have been completely based on the trust that this person feels in the prosecutors in general."
Mikheev said he would like to start a new company or take up another career, but is concerned he will not be able to remain in Kazakhstan.
"I've been lucky to live here for a year, and I want to continue this condition in the future. But my life is not going to be as easy as I thought," he said.
The 64-year-old's lawyer had urged the judge to immediately imprison him, adding that there are no limits to the amount of time a businessman can be jailed for acting in a "dirty manner."
Judge Ramy Azizov said his authority was constrained by an "expedited" schedule of prosecution against alleged bank thieves. Azizov added that a decision on the "witness' role" could take some time.
But Mikheev stressed that he would not be intimidated by authorities, suggesting his ruling may help prevent others from falling victim to the same actions. "I've heard of people who did things after acquiring the wrong identity documents, using aliases or in certain cases turning the wrong corner and, even if they used the correct documents, this can't mean that I've acted with gross negligence."
In response to Mikheev's remarks, a spokesman for the Mt Gox chief, Mark Karpeles, told the Russian press: "There are hundreds of people on our company payroll, some of them ex-employees. We don't comment on personal matters."

The case is "totally complex and complicated," said Ross Mrazoff, head of compliance at

KrebsOnSecurity, which monitors money-laundering cases. "It is a very important case, and the

decision was taken to speed up this investigation in

## F.3 GENERATED SAMPLES FROM MODELS TRAINED ON IMAGENET $256 \times 256$

In Figs. 13 to 15, we visualize samples of DDPD, Mask Diffusion and MaskGIT.

Without logit temperature annealing, Mask Diffusion captures diversity, but the sample quality suffers due to imperfections in the denoiser. On the other hand, MaskGIT's confidence-based strategy significantly improves sample quality, but at the cost of reduced diversity. DDPD trades off diversity v.s. quality naturally without the need for any annealing or confidence-based tricks.

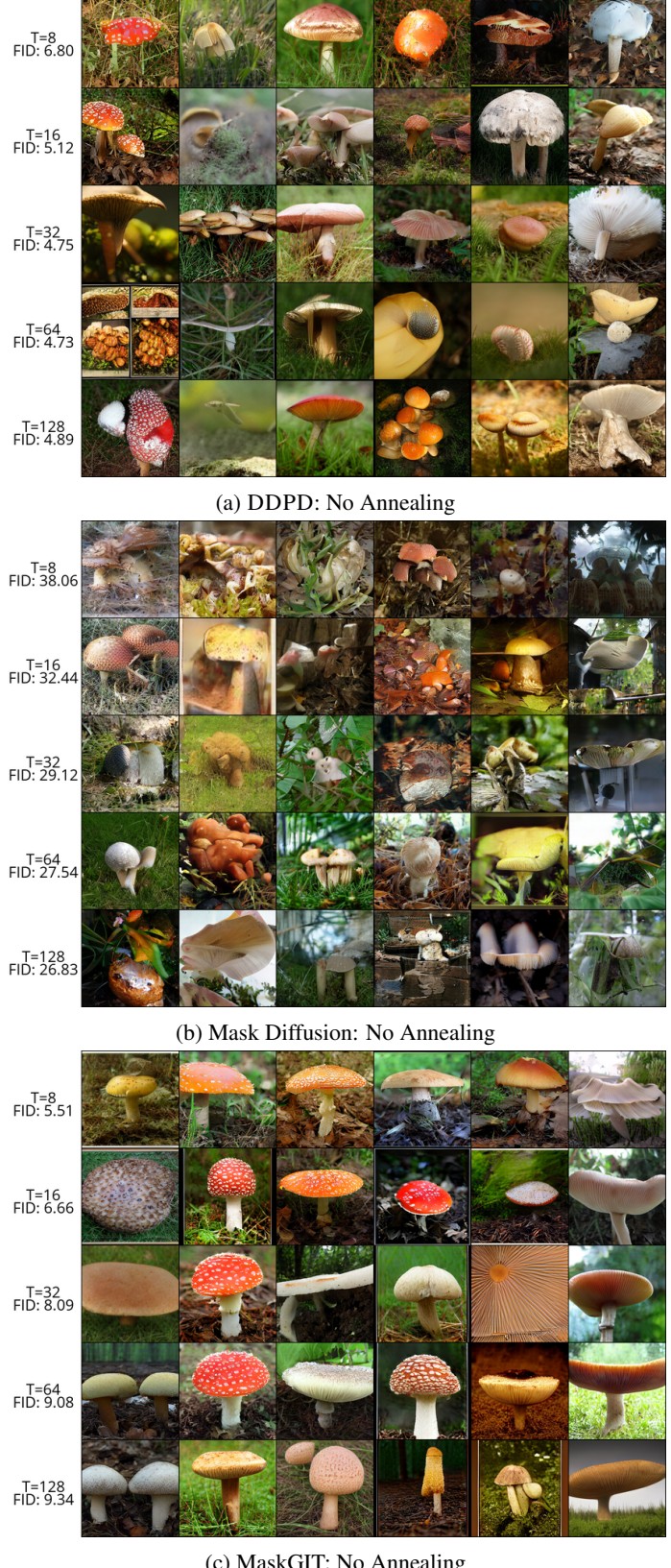

(a) DDPD: No Annealing

(b) Mask Diffusion: No Annealing

(c) MaskGIT: No Annealing

Figure 13: DDPD v.s. Mask Diffusion v.s. MaskGIT, No Logit Annealing

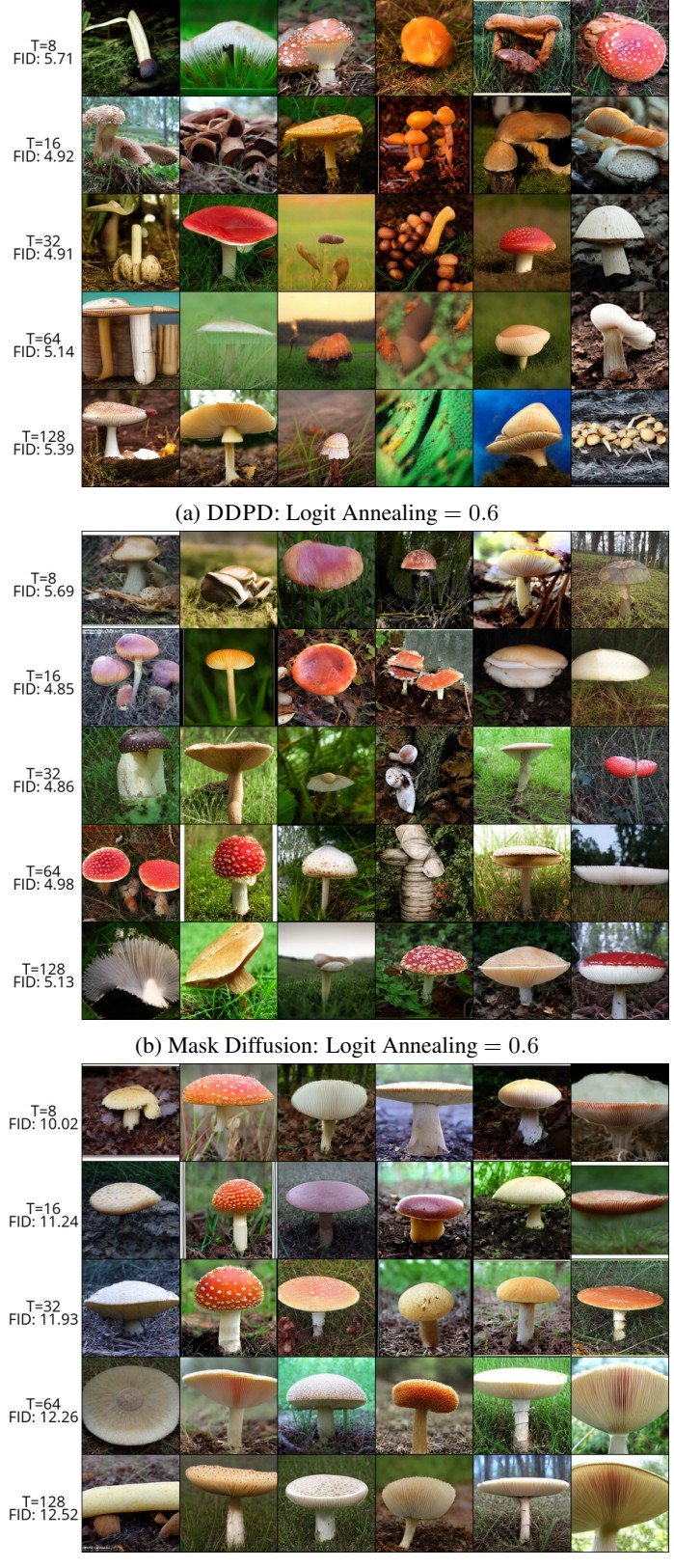

(a) DDPD: Logit Annealing = 0.6

(b) Mask Diffusion: Logit Annealing = 0.6

(c) MaskGIT: Logit Annealing = 0.6

Figure 14: DDPD v.s. Mask Diffusion v.s. MaskGIT, Logit Annealing = 0.6

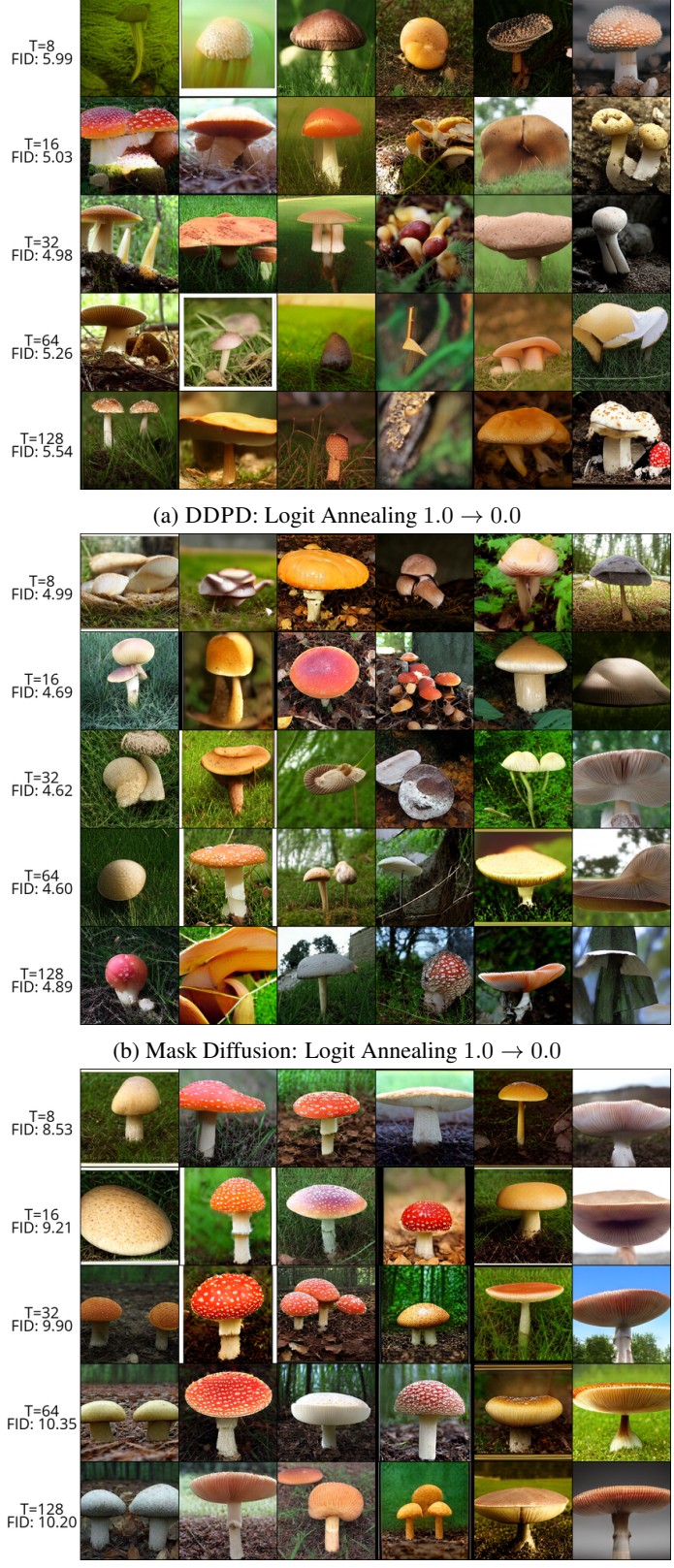

(a) DDPD: Logit Annealing $1.0 \rightarrow 0.0$

(b) Mask Diffusion: Logit Annealing $1.0 \rightarrow 0.0$

(c) MaskGIT: Logit Annealing $1.0 \rightarrow 0.0$

Figure 15: DDPD v.s. Mask Diffusion v.s. MaskGIT, Logit Annealing $1.0 \rightarrow 0.0$

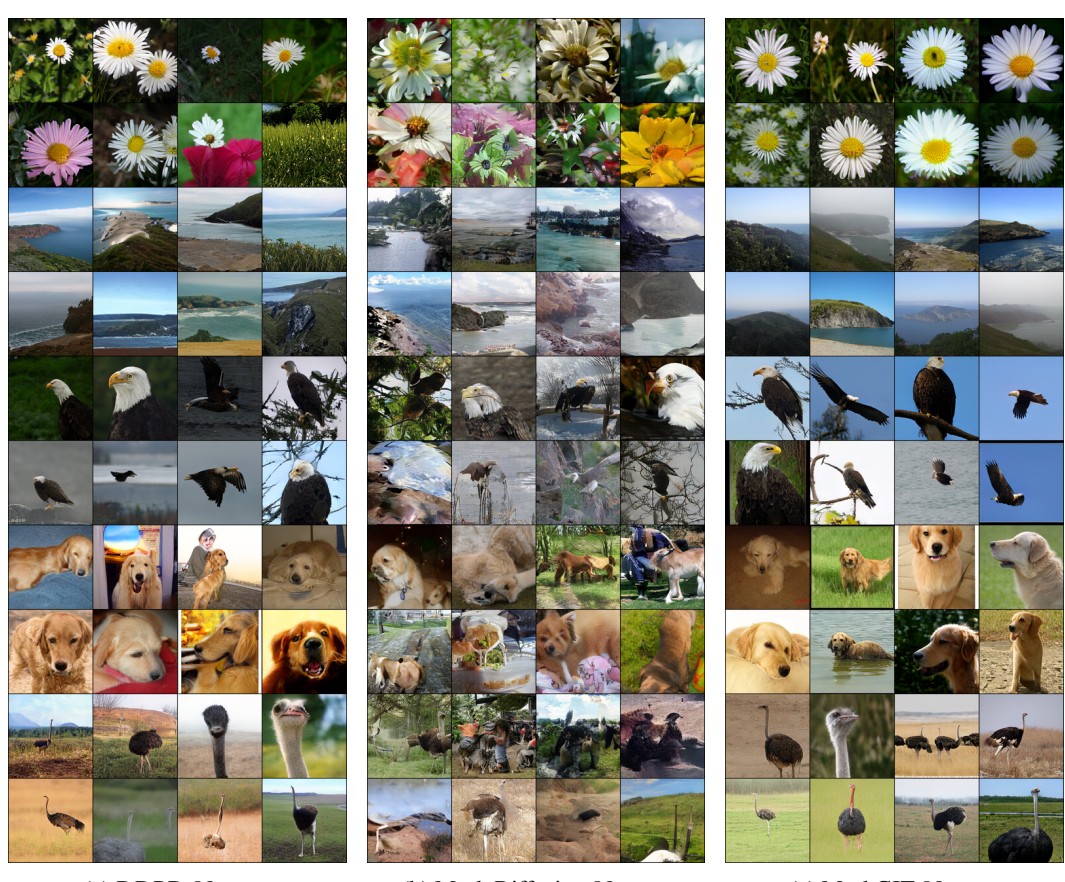

(a) DDPD 32 steps     (b) Mask Diffusion 32 steps     (c) MaskGIT 32 steps

Figure 16: DDPD v.s. Mask Diffusion v.s. MaskGIT, No Logit Annealing

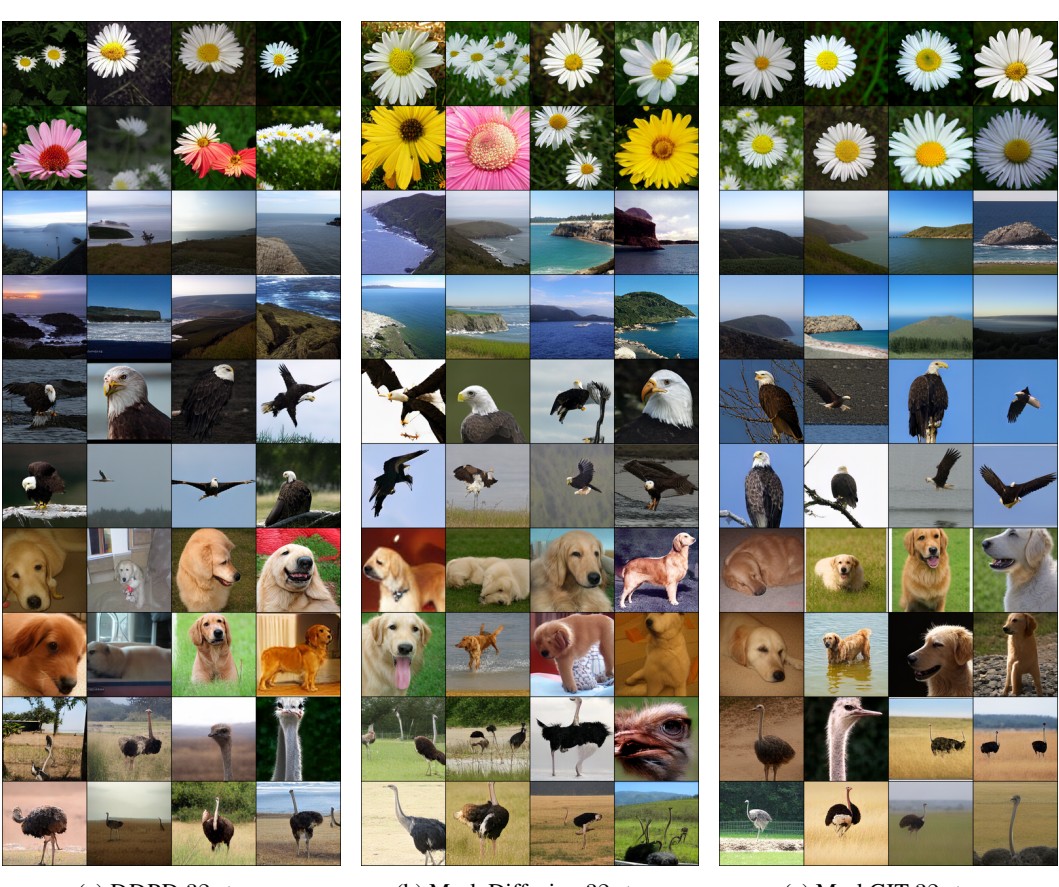

(a) DDPD 32 steps          (b) Mask Diffusion 32 steps          (c) MaskGIT 32 steps

Figure 17: DDPD v.s. Mask Diffusion v.s. MaskGIT, Logit Annealing $1.0 \rightarrow 0.0$

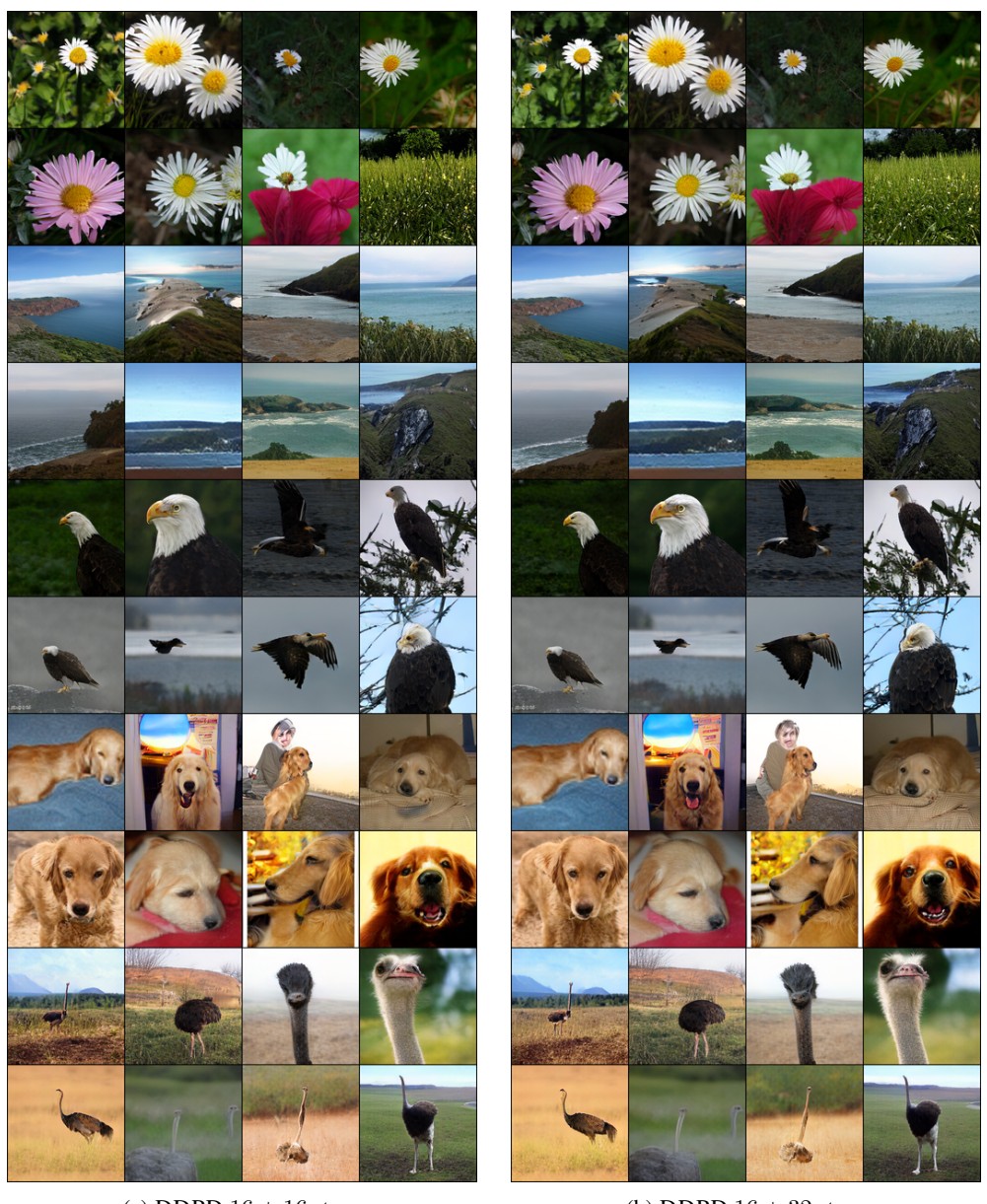

(a) DDPD $16 + 16$ steps                (b) DDPD $16 + 32$ steps

Figure 18: DDPD No Annealing, increasing number of refinement steps. The added refinement steps act as "touch-up" to improve the aesthetic quality without losing its original content.

