# OpenReview forum: "Think while You Generate: Discrete Diffusion with Planned Denoising"
_ICLR.cc/2025/Conference — ICLR 2025 Poster_

### Official Review · Reviewer_p5AT · 2024-10-29

**Soundness:** 3
**Presentation:** 2
**Contribution:** 3
**Rating:** 8
**Confidence:** 4

**Summary:**

This paper improves on the recent discrete diffusion generative models, with a focus on language modeling. This paper shows that the reverse (generative) rate can be decomposed in a planning and a denoising part. The proposed decomposition optionally allows training separate models for each part, which yields higher quality text samples, for a given inference budget. The decomposition also allows re-using a pre-trained masked discrete diffusion model to sample from a diffusion process with uniform stationary distribution. The training objective is theoretically sound as a decomposition of the ELBO.

**Strengths:**

- The presentation on Plan-and-Denoise decomposition is clear.
- The comparison of the sampling algorithm (Gillespie) and the comparison with tau-leaping is clear.
- Generally, Section 3 (method) is easy to follow and well-written.
- The note at the beginning of section 5 (experiment), that DDPD uses 2x the number of NFE versus regular absorbing diffusion is good to place there.
- While the main focus of the paper is on discrete diffusion for text generation, the authors have included experiments on images as well.

**Weaknesses:**

Overall, the method is sound, and the experiments are conducted fairly. The weaknesses noted here lie in the writing, which motivate my score. I have divided my comments into two groups: the first addresses more critical issues that I believe should be resolved, while the second includes suggestions that, while less urgent, could enhance the paper.


### More critical weaknesses
- Because of the description of the Gillespie's algorithm (Algorithm 1), it seems that each sampling iteration updates a single token at a time. But then, I'm confused on how the number of sampling steps can be lower than the number of tokens in Figure 2 (middle, DDPD). In particular, if you are generating sequences of 256 tokens on text8, how can you sample with 250 steps ? Indeed, there is a curve labeled DDPD-T250, which implies only 250 steps, so it feels like a piece of information might be missing here.

- Training Objective (lines 376 to the end of the page): The sentence on the 'coupling dynamics' and how it negatively affects backpropagation is unclear and should be rewritten.

- In Figure 2 (left), it appears that DFM-UNI, was only evaluated at a single temperature (shown by one dot) and others, like DDPD-MaskD, at just two temperatures (with only two dots). Could you confirm whether the models were indeed evaluated across all three temperature values? The missing points on the curve could be explained by missing experiments or dots that fall exactly on the same position for example. Therefore, I think there should be an explanation on this phenomenon.



- The meaning of the stochasticity parameter ($\eta$) on lines 431-432 (end of page 8, beginning of page 9) is not explained. While the context suggests that it comes from the DFM paper, a brief explanation of how it is used here is justified. I suggest adding a concise, high-level sentence about $\eta$, and if more detail is needed, directing readers to a section in the appendix.



### Less critical weaknesses
- Sampling with the Gillespie's algorithm requires sampling the holding time of the markov chain from an exponential distribution. However, for a reader that is not well-versed in simulation of markov chains, it might be unclear why the parameter of the exponential is chosen by summing over the rate for each dimension. I believe it would be good to include a paragraph on this, or at least a footnote to a precise chapter in a book such as the book of J.R. Norris on Markov Chains.

- For a reader short on time, it might be unclear what 'softmax selection' and 'proportional selection' refer to on line 476, as they haven’t been introduced previously. I assume these terms relate to the Gillespie sampler, where a dimension is chosen either by softmaxing the model output or dividing by the sum. To clarify, it would be helpful to label these explicitly. I would suggest mentioning "proportional selection" in bold in section 3.2, and mentioning that alternatively, one could perform "softmax selection" as well, again in section 3.2. This would make it easier for a reader that would search for "softmax selection"/"proportional selection" in the pdf.

- It would be helpful to include a timing comparison between DDPD, the diffusion baselines and the AR model. Indeed, a comparison of the computational resources vs generation quality is relevant for many applications.

### Final comment
I realize that implementing these suggestions would require adding content while you already reach the paper's page limit. However, I believe my requests are justified in order to improve the clarity of the paper.

**Questions:**

- Did you experience convergence issues during sampling? More precisely, what happens if you do not limit the maximal number of planning+denoising steps, but iterate until the planning module is sufficiently confident (e.g. assigns a probability at least 0.95, or 0.99) for all the tokens?
- Related to the above, say you are using a confidence threshold of 0.95 to decide on the convergence. What is the mean/std of the number of sampling (planning + denoising) steps that you observe when averaging over 10'000 examples?
- Do the authors have an explanation or hypothesis on why adding a planner seem detrimental to performance for images, but not for text (table 2 with logits scaling)?
- Have the authors experimented with other types of data already such as DNA sequences? I am curious about such experiments since the setup is relatively different (smaller vocabulary and longer sequences).

---

> ### Author Response · Authors · 2024-11-25
> **Response to p5AT (1/2)**
>
> Dear p5AT, thank you for your detailed review and thoughtful suggestions. We address your points below:
>
> ### More critical:
>
> > **“How can you sample with 250 steps for sequences of 256 tokens?”**
> >
>
> We start with a sequence of 256 random noise tokens, where 250 steps correspond to 250 denoising steps. We realize this might be confusing since 250 steps may not always suffice to completely denoise the sequence (even if the denoiser is perfect). Following your suggestion, we have updated our experiments to use 256, 512, 768, and 1024 steps to eliminate ambiguity. We thank you for pointing this out.
>
> > **Presentation suggestion on explaining “Coupling dynamics and backpropagation (lines 376–end of the page)”**
> >
>
> We appreciate your comment on the clarity of this section. We have revised the description to provide a clearer explanation:
>
> - The conflict or balancing between objectives might cause gradient interference, reducing the overall effectiveness of all models. This is a known challenge in multi-task learning [1].
> - Additionally, in setups where one network depends on the outputs of another, this can lead to unstable training dynamics, as instability in one network propagates to the other [2].
>
> [1] Yu, Tianhe, Saurabh Kumar, Abhishek Gupta, Sergey Levine, Karol Hausman, and Chelsea Finn. "Gradient surgery for multi-task learning." *NeurIPS* (2020).
> [2] Gulrajani, Ishaan, Faruk Ahmed, Martin Arjovsky, Vincent Dumoulin, and Aaron C. Courville. "Improved training of wasserstein gans." *NeurIPS* (2017).
>
> > **Clarification on missing points in “Figure 2 (left)”**
> >
>
> Thank you for the great suggestion! The reason only a subset of points is shown for some models is that the results for the three temperatures are similar. We plot only the Pareto-optimal points, as the other points are slightly less optimal. We will include this explanation in the paper.
>
> > **Presentation Suggestion on explaining “Stochasticity parameter ($\eta$)”**
> >
>
> We appreciate your suggestion. We will include a brief explanation of the stochasticity parameter ($\eta$) in the main text and direct readers to a detailed explanation in the appendix for more context.
>
> ### Less critical:
>
> > **Presentation suggestion on explaining “summing rates for the exponential distribution in Gillespie's algorithm”**
> >
>
> This is a great point. We will add a brief explanation to clarify that the parameter of the exponential distribution is the sum of rates over all dimensions, which corresponds to the total rate of transitions in the Markov chain. We will also reference Chapter 2 of J.R. Norris’ book *Markov Chains* for readers interested in further details.
>
> > **Presentation suggestion on “clarifying ‘softmax selection’ and ‘proportional selection’ (line 476)”**
> >
>
> Thank you for highlighting this. We will explicitly mention "proportional selection" and "softmax selection" in Section 3.2 and use bold text for these terms to make them easier to locate.
>
> > **“Timing comparison between models”**
> >
>
> This is an excellent suggestion. We have included a comparison of methods in terms of the number of NFE (Numerical Function Evaluations) in the beginning of Section 6. Since all models have the same inference cost per NFE, this plot provides a fair comparison of computational efficiency. We will also add wall-clock time comparison.

---

> > ### Author Response · Authors · 2024-11-25
> > **Response to p5AT (2/2)**
> >
> > ### Questions:
> >
> > > **“Convergence during sampling”**
> > >
> >
> > We observe that performance is robust and did not experience convergence issues in our experiments. Following your suggestion, we tried continue sampling for 1024 steps for text8 of sequence length 256. We observe that on average, number of predicted mask change is around 0.072 (out of 256), and number of token value change is around 0.162 (out of 256). The generative perplexity and entropy values also converge, with no change if continuing the sampling to 1280 steps.
> >
> > > **“Why is the planner detrimental for images but not for text (Table 2)?”**
> > >
> >
> > As noted in the experimental section, the accuracy of the image denoiser is very low (3% for images vs. 60% for text). As a result, logit annealing plays a critical role in improving image generation performance. Without logit annealing, Mask Diffusion performs poorly, while DDPD can correct earlier denoising mistakes. Hence, planning is shown to be super effective and necessary.
> >
> > With logit annealing, the accuracy of the denoiser is greatly improved (with potential sacrifice of diversity), both Mask Diffusion and DDPD achieve comparable FID scores. DDPD is sometimes slightly worse, potentially due to the lack of diversity in low temperature sampling outweigh the benefits of correcting mistakes with planning.
> >
> > > **“Have the authors experimented with other types of data, such as DNA sequences?”**
> > >
> >
> > While we have not explored DNA sequences in this work, we expect the method to be effective for tasks where accumulated denoising errors harm generative performance. Additionally, the planner prioritizes the most corrupted regions first, which may be particularly beneficial for tasks with long contexts or specialized structure, such as DNA sequences. We hope to explore these applications in future work.

---

> > > ### Comment · Reviewer_p5AT · 2024-11-25
> > >
> > > I am grateful to authors for engaging in the discussion.
> > >
> > > I would like to ask further clarification for my first question  (**“How can you sample with 250 steps for sequences of 256 tokens?”**):
> > >
> > > Since the sequence length is 256 and the authors sampled with 250 steps in the original submission, does that mean that some tokens were still noised after sampling? From remark 3.4 (*"such that one step leads to one token denoised"*), it seems that you need at least one step per token, but you might need more, hence it is still unclear what happened in the original experiment with 250 sampling steps.
> > >
> > > Finally, the wording of the rebuttal suggests that the authors have updated the manuscript. However, Figure 2 still shows the DFM-T250 experiment (which raised my first question). I would be grateful if you could upload an updated manuscript with changes clearly visible (e.g. in blue or red).

---

> ### Author Response · Authors · 2024-11-25
> **Further clarification on “How can you sample with 250 steps for sequences of 256 tokens?”**
>
> We appreciate your recognition of our efforts to address your questions. Thank you for the follow-up question.
>
> To clarify, we can use the sampling trajectory in Figure 1 as an example. The sequence starts with noisy tokens, and if the number of steps is insufficient to denoise everything correctly, the sequence will end with some tokens still remaining noisy. For instance, if only 4 steps are allowed, the sampling process may terminate early and produce imperfect samples, such as "BREDN" in Figure 1. However, in another scenario where sampling progresses more favorably, it is possible to arrive at "BREAD" at step 4, since "B" was already correct in the initialization. This process illustrates how we generated samples with 250 steps and evaluated their quality.
>
> We have updated the manuscript to include the revised Figure 2 reflecting the updated experiments with 256 steps instead.
>
> Additionally, we note that in practice, denoisers are not perfect and can introduce sampling errors. Even with 256 steps, sequences may still contain noisy tokens for both mask diffusion and DDPD.
>
> Thank you again for your follow-up, and we hope this clarifies your concerns. Please let us know if you have any additional questions or suggestions!

---

> > ### Comment · Reviewer_p5AT · 2024-11-25
> >
> > Thank you again for engaging in the discussion. The concerns from my original review have been addressed. I have updated my score accordingly.
> >
> > As a final suggestion for future submissions, please consider highlighting in red the changes done during the rebuttals, to make it easier for the reviewers to see the updates. Without such highlighting, looking for updates in the manuscript requires more time, care and attention from the reviewer.

---

> > > ### Author Response · Authors · 2024-11-25
> > > **Thank you for acknowledging our response clarifies your concerns and updating your score**
> > >
> > > Thank you once again for your valuable feedback and insightful discussions. We are in the process of revising the manuscript to incorporate the comments and suggestions provided by all reviewers. Once the updates are complete, we will promptly upload the revised version. Following your suggestion, we will ensure that the changes are clearly highlighted in the manuscript for easy reference.

---

### Official Review · Reviewer_4zHc · 2024-11-02

**Soundness:** 2
**Presentation:** 2
**Contribution:** 2
**Rating:** 3
**Confidence:** 3

**Summary:**

This paper rethinks limitations of the conventional discrete diffusion generation schema that the generated tokens cannot be further corrected and the sampling process cannot be controlled. The authors propose to disentangle the single diffusion denoising process into two models: a planner and a denoiser. This planner is capable of selecting which positions to denoise next by identifying the most corrupted positions in need of denoising, including both initially corrupted and those requiring additional refinement. The plan-and-denoise method allows for adaptive control of the denoising order for efficient reconstruction during generation. The experimental results across text and images domains demonstrate the efficacy of the proposed model.

**Strengths:**

1. The motivation behind the study is sound, as discrete diffusion models struggle with correcting generated tokens and exhibit inflexibility during the sampling process.
2. The proposed model to disentangle the generation process with a planer and a denoiser is simple.
3. The theoretical analysis appears robust and well-founded.

**Weaknesses:**

1. Additional experiments are necessary, such as applying the model to machine translation tasks for texts.
2. Although the planner model can adaptively organize the generation process, performance degradation becomes an issue as the number of sampling steps increases.
3. The narrative is somewhat difficult to follow; I recommend that the authors clarify the main storyline.

**Questions:**

see weakness

---

> ### Author Response · Authors · 2024-11-25
> **Response to 4zHc**
>
> Dear 4zHc, thank you for your review and constructive feedback. We address your points below:
>
> > **“Additional experiments are necessary, such as applying the model to machine translation tasks for texts.”**
> >
>
> We appreciate the suggestion to explore machine translation. While this would indeed be an interesting application, we chose to focus on experiments with text8 and OpenWebText, as these are the standard benchmarks for discrete diffusion works [1,2,3]. Additionally, we provide experiments on ImageNet256, which extend the evaluation scope of our method.
>
> Machine translation, as a task, involves providing a generative model with additional conditioning. In our work, we aim to evaluate the general strength of the proposed discrete generative framework, focusing on its capabilities without being tied to specific conditioning tasks.
>
> > **“The narrative is somewhat difficult to follow; I recommend that the authors clarify the main storyline.”**
> >
>
> Thank you for pointing this out. To improve clarity, we have restructured Section 1 to better highlight the key motivations and contributions of our work. Additionally, we revised Section 3 and 4 to make the explanations more concise and intuitive. We welcome further feedback on specific areas where the narrative might still seem unclear.
>
> > **“Although the planner model can adaptively organize the generation process, performance degradation becomes an issue as the number of sampling steps increases.”**
> >
>
> We believe there might be some misunderstanding here. The performance of our model generally improves as the number of sampling steps increases, as demonstrated in Figures 2 and 3. To make this trend clearer, we have added up and down arrows in the captions of Figures 2 and 3 to explicitly highlight the improvements.
>
> The confusion might arise from Table 2, which presents results from ImageNet experiments using parallel decoding (for text experiments it was one token at a time). In this case, performance does not necessarily increase monotonically with additional sampling steps. This is because parallel decoding, which is particularly effective for image generation, causes performance to improve with more steps initially but eventually plateau, which is observed for all baseline methods as well (for MaskGIT it hurts diversity too much and leads to worse FID scores).
>
> We hope these clarifications address your concern, and we are happy to provide further explanations if needed.
>
> [1] Lou, Aaron, Chenlin Meng, and Stefano Ermon. "Discrete diffusion language modeling by estimating the ratios of the data distribution." *ICML* (2024).
>
> [2] Campbell, Andrew, Jason Yim, Regina Barzilay, Tom Rainforth, and Tommi Jaakkola. "Generative flows on discrete state-spaces: Enabling multimodal flows with applications to protein co-design." *ICML* (2024).
>
> [3] Shi, Jiaxin, Kehang Han, Zhe Wang, Arnaud Doucet, and Michalis K. Titsias. "Simplified and Generalized Masked Diffusion for Discrete Data." *NeurIPS* (2024).

---

> > ### Author Response · Authors · 2024-11-28
> >
> > Dear 4zHc, we hope our clarifications, revisions to the storyline (as detailed in the updated manuscript), and explanations have addressed your concerns and resolved any potential misunderstandings. Please let us know if otherwise!

---

### Official Review · Reviewer_yRLR · 2024-11-04

**Soundness:** 3
**Presentation:** 3
**Contribution:** 3
**Rating:** 6
**Confidence:** 3

**Summary:**

This paper introduces DDPO, a novel method that separates the generation process of discrete diffusion into two models: a planner and a denoiser. The authors unify the sampling processes of masked diffusion and uniform diffusion in a plan-and-denoise manner and present the advantages of this approach over previous works. Additionally, the authors provide the evidence lower bound (ELBO) for DDPO. Experiments in both text generation and image generation validate the effectiveness of DDPO.

**Strengths:**

1. This paper focuses on an interesting and important problem in the sampling processes of discrete diffusion models.
2. The writing is clear. The authors logically present the sampling process of DDPO and its advantages compared to previous works. And even with the extra planner, the training objective of DDPO still corresponds to the evidence lower bound.
3. The experiments are thorough, demonstrating the effectiveness of DDPO in both text generation and image generation tasks.

**Weaknesses:**

1. I suggest the authors use different notations for single-dimensional and multi-dimensional cases, e.g., x and **x**.

2. The authors claim that each sampling step of previous works predicts single-dimension transitions but not joint transitions of all dimensions. However, based on Proposition 3.1, the proposed DDPM also just predicts single-dimension transitions.

3. As presented in [1], the best FID score on ImageNet 256×256 is 1.97. Why are all the results in Table 2, both for the baseline and the proposed DDPO, significantly worse than 1.97? Is it because classifier-free guidance was not used? If so, why are there no results that include classifier-free guidance? This is my main concern; I will improve my score if a clear response is presented.

[1] Yu et al. An image is worth 32 tokens for reconstruction and generation. NeurIPS 2024.

**Questions:**

1. The term $p_{1|t}(x_1=j|x_t, z_t=D)$ (in Line 192) indicates that once a dimension is identified as data by the planner (i.e., $z_t=D$), it will not be updated in this step, similar to masked diffusion. I am curious whether a dimension that is recognized as data by the planner early in the sampling process remains unchanged until the end of the sampling. Could the authors provide further theoretical and experimental insights on this issue?
2. DDPO employs a different training objective (i.e., Theorem 4.1). Will this new training objective lead to lower perplexity in language modeling?

---

> ### Author Response · Authors · 2024-11-25
> **Response to yRLR (1/2)**
>
> Dear yRLR, thank you for taking the time and efforts in reviewing our paper. We appreciate your recognition of our paper’s contribution and hope to clarify on the questions you raised.
>
> > **“I suggest the authors use different notations for single-dimensional and multi-dimensional cases, e.g., x and x.”**
> >
>
> Thanks for the feedback. We agree that it is sometimes causing confusion and will follow your suggestion.
>
> > **“The authors claim that each sampling step of previous works predicts single-dimension transitions but not joint transitions of all dimensions. However, based on Proposition 3.1, the proposed DDPM also just predicts single-dimension transitions.”**
> >
>
> We believe this is a misunderstanding. Current neural network architectures (transformers in this case) are inherently designed to model single-dimension transitions (rather than joint transitions in parallel), and our framework adheres to this same assumption. If one opts for parallel decoding to achieve faster generation—despite the trade-off of potentially introducing more denoising errors—our framework offers a significant advantage by identifying and correcting these errors, as demonstrated in our experiments on ImageNet256 token generation.
>
> > **“I am curious whether a dimension that is recognized as data by the planner early in the sampling process remains unchanged until the end of the sampling.”**
> >
>
> Thank you for raising this important question. Unlike traditional masked diffusion, where once a dimension is unmasked it cannot be revisited, our planner dynamically re-evaluates the probabilities of $z_t^i = D$ or $z_t^i = N$ for all positions at each step based on the updated $x_t$. This means that dimensions recognized as data ($z_t = D$) early in the sampling process are not fixed permanently but can later transition to $z_t = N$ if needed.
>
> This dynamic re-evaluation enables self-correction and ensures flexibility in the sampling process. For example, if a previously denoised token introduces errors due to noise or inaccuracies, the planner can reassign it as $z_t = N$ and allow the denoiser to revisit and correct it. This mechanism is a key advantage of our approach over methods like masked diffusion, where such corrections are not possible.
>
> In experiments, we observe that dimensions marked as $D$ can indeed revert to $N$ later in the process if the planner identifies the need for correction, and such reversals are instrumental in improving the overall quality of generated samples. To give an idea of how frequent those reversals happen during the sampling steps, we report the statistics on text8 in the updated manuscript. For text8 sequence of length 256, the average number of mask value change between step $t$ and $t+1$, starts with 40.40 at $t=50$ and converges to 0 when t gets to $1024$.

---

> ### Author Response · Authors · 2024-11-25
> **Response to yRLR (2/2)**
>
> > **“FID scores for ImageNet 256x256 v.s. 1.97 in [1]. Is it because classifier-free guidance was not used? If so, why are there no results that include classifier-free guidance?”**
> >
>
> Thanks for the suggestion. Yes, our results are without classifier-free guidance. We initially chose not to include it to eliminate the influence of an additional confounding factor. Moreover, the classifier-free guidance induces a different generative process than the original theoretically correct process. Therefore, we aimed to evaluate the effect of DDPD on the original diffusion process.
>
> For completeness, we have now added experiments with classifier-free guidance. The effect of classifier-free guidance is similar to temperature annealing. In the case of no logit annealing, it helps mask diffusion to achieve much better FID score. DDPD achieves similar FID scores but much better Inception Scores (which means aesthetically higher quality). When logit temperature = 0.6, the inception score of Mask Diffusion is greatly improved, but the composition of logit annealing and classifier-free guidance leads to worse FID score due to less diversity. The same applies to MaskGIT.
>
> ```python
> Table 1. CFG scale 2.0, Logit Temp 1.0 or 0.6, MaskGIT anneals confidence noise temperature from 1.0 to 0.0
> | Steps T | No Logit Annealing   (FID/Inception Score)     | Logit Temp 0.6     (FID/Inception Score)       |
> |---------|------------------------------------------------|------------------------------------------------|
> |         | MaskD         | MaskGIT         | DDPD         | MaskD         | MaskGIT         | DDPD         |
> | 8       | 8.44/145.36   | 6.89/337.37     | 7.19/323.79  | 6.37/320.48   | 13.00/400.23    | 7.45/333.28  |
> | 16      | 5.30/187.24   | 9.91/391.98     | 6.28/329.40  | 8.17/365.30   | 13.93/407.12    | 5.73/319.52  |
> | 32      | 4.14/215.88   | 11.49/408.73    | 5.58/322.67  | 9.06/384.50   | 14.16/406.19    | 4.98/315.14  |
> | 64      | 3.82/223.22   | 12.04/409.16    | 5.16/323.62  | 9.51/388.71   | 14.28/405.01    | 4.66/311.82  |
> | 128     | 5.02/180.79   | 11.55/396.81    | 4.63/313.09  | 8.31/361.80   | 13.76/388.04    | 4.39/304.92  |
> ```
>
> In [1], achieving the optimal FID score of 1.97 required balancing inception quality against diversity by introducing several heuristic hyperparameter strategies. These included MaskGIT sampling, starting with a high temperature (>1, e.g., 3.0 in [1]) and gradually anneals to 0, as well as logit annealing (0.6). While these heuristics deviate from the theoretical generative process, they achieve the best empirical performance when carefully combined and tuned. In contrast, our plan-and-denoise sampler provides a principled approach to reducing sampling errors. However, incorporating all the aforementioned heuristics falls outside the scope of this methodology-focused paper.
>
> > **Will this new training objective lead to lower perplexity in language modeling?**
> >
>
> Please see common response C.1.1 and C.1.2.

---

> ### Comment · Reviewer_yRLR · 2024-11-26
>
> Thank you for your response! Your reply addressed most of my questions effectively. However, I remain particularly concerned about Weakness 3.
>
> I respectfully disagree with your statement that *"incorporating all the aforementioned heuristics falls outside the scope of this methodology-focused paper."* In my view, an excellent methodology-focused paper not only presents a comprehensive theoretical framework but also demonstrates strong empirical results to support its claims.
>
> In addition, I found some aspects of your response a bit confusing. Could you clarify what CFG scale is used in the table? You mentioned that *"the composition of logit annealing and classifier-free guidance leads to worse FID scores."* However, in [1], the best FID score is achieved precisely through this combination. I believe the worse FID result might be due to the authors not properly adjusting the CFG scale and temperature settings.
>
> While I acknowledge that [1] employs multiple heuristics during sampling, my understanding is that these heuristics are plug-and-play compatible with DDPD. Is that correct? If so, what would the FID score of DDPD be if these heuristics were also applied? If the FID score of DDPD still falls far behind 1.97 even after these heuristics are applied, could you explain why this happens? Could it be that DDPD is inherently less robust to high CFG scales? If the authors explicitly point out that DDPD is less robust, this would represent a valuable and constructive contribution to the paper.
>
> I believe that addressing these points thoroughly would significantly enhance the clarity and impact of the discussion.

---

> ### Author Response · Authors · 2024-11-27
> **Further clarification on Image token sampling with CFG**
>
> We sincerely thank you for your detailed feedback and thoughtful suggestions. We appreciate the opportunity to clarify and expand upon the points raised, particularly regarding Weakness 3.
>
> > **“Could you clarify what CFG scale is used in the table?“**
> >
>
> In the table from our previous response, all methods use a CFG scale of 2.0 and a logit temperature of either 1.0 or 0.6. For MaskGIT sampling, the confidence temperature (a mechanism introducing stochasticity by adding noise to the logit confidence) is set to 1.0 and linearly annealed to 0.0.
>
> > **“You mentioned that *"the composition of logit annealing and classifier-free guidance leads to worse FID scores."* However, in [1], the best FID score is achieved precisely through this combination. “**
> >
>
> Thank you for pointing this out. We acknowledge that our previous response may have been unclear. As detailed in [1], the best FID scores are achieved using additional hyperparameter adjustments, specifically inflating the logit and confidence temperatures to 3.0, which are then linearly annealed to 0.0 ([implementation here](https://github.com/bytedance/1d-tokenizer/blob/6a8b33569d4feb68f59e7c96349a43e23d686e78/modeling/maskgit.py#L132)). This unconventional setting is tailored to counter the loss of diversity caused by the combination of logit annealing and CFG. While this approach improves FID, it negatively affects inception scores, reflecting a quality-diversity trade-off.
>
> To address this, Table 2 presents results with the same configurations from [1], including MaskGIT sampling with logit annealing, where the logit and confidence temperatures are set to start from 3.0.
>
> > **“my understanding is that these heuristics are plug-and-play compatible with DDPD. Is that correct? … Could it be that DDPD is inherently less robust to high CFG scales?”**
> >
>
> The interactions between all the heuristics are much more nuanced than they appear. For instance, in the configuration used in [1], to counterbalance the lack of diversity from using logit annealing + CFG, an artificially inflated higher temperature (>1, e.g., 3.0 for the linear unmasking schedule and 6.9 for the cosine schedule) is employed to achieve better coverage of diversity. Interestingly, the initial tokens sampled under higher temperatures (which are expected to contain more errors) combined with the later temperature-annealed tokens remain coherent and are correctly decoded by the decoder. These techniques are specifically designed and optimized for mask diffusion with CFG, whereas DDPD relies on uniform diffusion.
>
> Following your suggestion, we applied the same configuration to DDPD (in Table 2). This resulted in better inception scores but worse FID scores due to the trade-off between quality and diversity. Upon investigation, we observed that the planner proactively predicts some of the initially generated tokens under higher temperatures to be noise (as they deviate more from the original distribution). These tokens are subsequently corrected in later steps, leading to reduced diversity but improved visual quality. Exploring how to combine DDPD, CFG, and other heuristics more effectively remains an interesting open research question—for instance, through on-policy training with samples using higher temperatures or additional heuristics for tuning the planner's temperature and/or CFG.
>
> Our paper focuses on the methodology for sampling without CFG (as is commonly applied in most applications such as text and protein). Therefore, we leave these explorations to future work.
>
> Again, we agree that this is a very interesting and important point and we will include the above detailed discussion in our paper.
>
> ```jsx
>      Table 2: When using same hyperparameters in [1] (8 steps are not enough for DDPD to converge)
> | Steps T |        FID/Inception Score               |
> |---------|------------------------------------------|
> |         | MaskGIT           | DDPD                 |
> | 16      | 2.48/284.20       | 7.93/359.69          |
> | 32      | 2.74/299.53       | 6.11/343.63          |
> | 64      | 2.86/309.62       | 5.11/328.56          |
> | 128     | 2.93/205.26       | 4.63/313.09          |
> ```
>
> Thank you once again for your follow-up, and we hope this response has clarified your concerns. Please feel free to reach out if you have any further questions or suggestions!

---

> > ### Comment · Reviewer_yRLR · 2024-11-28
> >
> > Thanks for your clarification. I have adjusted my score accordingly from 5 to 6.

---

> > > ### Author Response · Authors · 2024-11-28
> > > **Thank you for acknowledging our response clarifies your concerns**
> > >
> > > We appreciate your recognition that our response addressed your concerns and adjusting your score. Thank you again for your thoughtful evaluation and valuable suggestions.

---

### Official Review · Reviewer_uHmQ · 2024-11-09

**Soundness:** 3
**Presentation:** 3
**Contribution:** 3
**Rating:** 6
**Confidence:** 5

**Summary:**

This paper introduces Discrete Diffusion with Planned Denoising (DDPD), unlike traditional discrete diffusion models which rely solely on a denoising network, DDPD separates the generation process into two distinct steps: planning and denoising. A planner network identifies the most corrupted positions within a sequence, prioritizing them for correction, while a denoiser network then predicts the correct value for these selected positions. This decomposition simplifies the learning process for each network and allows for a more efficient sampling algorithm. The paper proposes an adaptive sampling scheme based on the Gillespie algorithm, which dynamically adjusts the denoising step size based on the planner's assessment of the sequence's corruption level. This adaptive sampling, coupled with a time correction mechanism, allows DDPD to iteratively refine generated sequences by revisiting previously denoised positions if errors were made. The authors demonstrate experiments on text8, OpenWebText, and ImageNet that DDPD improves sample quality and diversity compared to existing discrete diffusion methods, particularly when inference compute budget is limited. Importantly, DDPD allows for leveraging pre-trained mask diffusion denoisers for uniform diffusion tasks, demonstrating improved performance by combining a strong mask denoiser with a separately trained planner.

**Strengths:**

- The core idea of decoupling the denoising process in discrete diffusion into separate planning and denoising stages is novel and very interesting. The adaptive sampling scheme based on the Gillespie algorithm, combined with the time correction mechanism, is also a creative contribution that addresses limitations of standard tau-leaping samplers.
- The paper provides a strong theoretical foundation for DDPD, grounding the training objectives in a clear derivation of the ELBO for discrete diffusion. The experimental results are comprehensive, comparing DDPD against strong baselines across diverse tasks including language modeling and image generation, showcasing its consistent performance gains. The ablation studies further strengthen the analysis by demonstrating the robustness of DDPD to imperfect training and the individual contributions of its proposed modifications.
- The paper is dense yet generally well-written and easy to follow.
- DDPD addresses a key challenge in discrete diffusion models: the efficient and effective use of the limited computational budget during sampling. By prioritizing denoising efforts through planning and allowing for self-correction, DDPD offers a promising path towards closing the performance gap between diffusion and autoregressive models in discrete generative modeling. The ability to leverage pre-trained mask diffusion denoisers further enhances the practical significance of DDPD, making it more accessible for researchers working with limited resources.

**Weaknesses:**

- The paper acknowledges the increased computational cost of DDPD compared to denoiser-only methods due to the additional planner network and the sequential nature of the Gillespie sampler.
- The clarity can be further improved by adding more description of the training algorithm/procedure of the model confirugrations considered in Sec.6, given that there are many variations and the forward/training objectives/planner&denoiser training can vary.
- Lack of empirical NLL/perplexity comparisons on OpenWebText dataset with baseline discrete diffusion models. While this work presents results and analysis in terms of generative perplexity on OWT, this metric is easier to be inflated and sensitive to the precision used in categorical sampling. The more commonly used validation perplexity is not reported, as well as the zero shot perplexity on other datasets.
- Missing reference: The proposed adaptive time correction is interesting and well-motivated. Similar ideas/trick was proposed in [1] with asymmetric time intervals for a different context but not discussed as related work.

[1] Chen, T., Zhang, R., & Hinton, G. (2022). Analog bits: Generating discrete data using diffusion models with self-conditioning. arXiv preprint arXiv:2208.04202.

**Questions:**

- What are authors' thoughts on the connections of this work  to the state-dependent masking schedule proposed in MD4?

---

> ### Author Response · Authors · 2024-11-25
> **Response to uHmQ**
>
> Dear uHmQ, thank you for your thoughtful review and valuable feedbacks.
>
> > **Suggestions on presentation: “The clarity can be further improved by adding more description of the training algorithm/procedure of the model configurations considered in Sec.6, given that there are many variations”**
> >
>
> This is a great suggestion, we will add details for the different configurations considered and update the manuscript towards the end of the discussion phase.
>
> > **"Missing reference on asymmetric time intervals in [1]"**
> >
>
> Thanks for the suggestion! The fixed time difference adjustment of [1] shares similar idea with the concurrent EDM paper [2] in slightly adjusting time backwards for continuous data, which is shown to be very effective in improving quality. We will include this discussion and compare our learnable adaptive time correction with the pre-defined time adjustment in [1][2].
>
> > **“Lack of test/valiation NLL/perplexity comparisons … generative perplexity is easier to be inflated and sensitive to the precision used in categorical sampling”**
> >
> Please see common response C.1.1 and C.1.2.
>
> > **"Thoughts on connection of DDPD to the state-dependent masking schedule in MD4 [3]"**
> >
>
> How they compare in terms of methodology:
>
> - Type of diffusion process: MD4 uses mask diffusion (data —> mask token), DDPD uses uniform diffusion (data —> uniform noise token).
> - Noise schedule: The state-dependent masking schedule in MD4 assumes each different token value has a learnable scalar schedule. In reconstruction, this means some token values should be generated first (for example punctuation marks such as comma, period etc.). In DDPD, we consider fixed noise schedule for simplicity. Reconstruction follows the standard $p(x_1^i|x_t)$, which does not have a preference on which token values to generate first.
> - Optimization of ELBO: In MD4, the learnable noise masking schedule and the denoiser need to be jointly optimized and requires the use of additional REINFORCE estimator to get unbiased gradients. The ELBO under DDPD decomposes to two separate parts and can be separately optimized using standard cross-entropy loss.
>
> How they perform in practice:
>
> - MD4’s state-dependent masking schedule leads to prioritizing generation certain token values first (such as punctuation marks), but each mask token still has same chance of being denoised. DDPD leads to choosing which dimension to denoise based on which tokens are noisier.
>
> In summary, they focus on orthogonal aspects of the discrete diffusion sampling problem and can be combined to complement each other.
>
> [1] Chen, T., Zhang, R., & Hinton, G. (2022). Analog bits: Generating discrete data using diffusion models with self-conditioning. arXiv preprint arXiv:2208.04202.
>
> [2] Karras, Tero, Miika Aittala, Timo Aila, and Samuli Laine. "Elucidating the design space of diffusion-based generative models." *Advances in neural information processing systems* 35 (2022): 26565-26577.
>
> [3] Shi, Jiaxin, Kehang Han, Zhe Wang, Arnaud Doucet, and Michalis K. Titsias. "Simplified and Generalized Masked Diffusion for Discrete Data." *NeurIPS* (2024).

---

> > ### Author Response · Authors · 2024-11-30
> >
> > Dear uHmQ, thank you again for your valuable feedbacks. We have included clarifications and experiments on test perplexity evaluation, as well as discussion on time-adjustment in relevant work [1] and connections to MD4. We hope our response have addressed your concerns and answered your questions. Please let us know if otherwise!

---

### Author Response · Authors · 2024-11-25
**Common response**

We sincerely thank the reviewers for their valuable feedback and insightful suggestions. We appreciate the reviewers’ recognition of our work's novelty, the importance of the addressed problem, clear-writing, strong theoretical foundation and thorough experiments. This general response aims at clarifying and addressing common questions.

> **C.1.1 including comparison for validation/test/zero-shot perplexity (uHmQ and yRLR)**
>
- We have included a comparison of ELBO on test data across methods in Line 451, with details in Tables 4–6 (in Appendix). Appendix C.1 explains ELBO derivations for uniform and mask diffusion. Uniform diffusion and DDPD have worse evidence lower bound in terms of BPC on test data when compared to mask diffusion, which is expected and aligned with previous observations in [1]. In other words, the uniform diffusion type methods have a lower compression rate.
- The ELBO reflects the likelihood of the generative process that exactly follows the test data. However, it does not count for the accumulated discretization errors and denoising errors that occur during actual sampling in practice. And this is exactly what DDPD sampler (with time adjustment) is trying to improve on: to adaptively fix sampling errors accumulated in previous steps. To give an example, the ELBO on test data is the same for 1) Uniform Diffusion with tau-leaping sampler (DFM-Uni, $\tau$-leaping) v.s. 2) DDPD sampler without time adjustment (DDPD-DFM-Uni, A) v.s. 3) DDPD sampler with time adjustment (DDPD-DFM-Uni, A+B or A+B+C), since they use the same uniform diffusion model. But due to differences in sampling, using planner-induced time-adjustment leads to a huge improvement in terms of sample quality v.s. diversity trade-off (see Figure 7, top-left shows the generative ppl v.s. entropy). Similar phenomenon are observed in [2] in mask diffusion when tuning the stochasticity $\eta$ in sampling.
Therefore metric like generative perplexity (eval under a stronger model) or FID score is a fairer metric that better reflects the quality of the generation.

> **C.1.2 generative perplexity is easier to be inflated and sensitive to the precision used in categorical sampling (uHmQ)**
>
- We are aware of the pitfalls of generative perplexity [2] and categorial sampling precision effects [3]. In particular, generative perplexity can be easily gamed by generating repeated tokens of high gen. ppl. with reduced entropy. Therefore, we have been careful by following [2] to measure both generative perplexity and entropy. And we do not see the generative perplexity being gamed with low diversity of the samples in DDPD.
- In terms of sampling effect caused by precision in [3], the truncation effect occurs when a very fine discretization $\Delta t$ is used, and sampling next-step transitions follows $\text{next step probabilities} \times \Delta t$ which results in underflow issues and reduces diversity in sampling. However, in DDPD’s implementation that uses Gillespie sampler (see Algo. 1 and Listing 1), we use the planner to pick the dimension first (and implicitly simulating a $\Delta t$ until the next state transition), and then use the denoiser $p(x_1^d|x_t)$ to sample the state transition. Therefore, DDPD does not have the precision issue in sampling.

[1] Austin, Jacob, Daniel D. Johnson, Jonathan Ho, Daniel Tarlow, and Rianne Van Den Berg. "Structured denoising diffusion models in discrete state-spaces." *Advances in Neural Information Processing Systems* 34 (2021): 17981-17993.

[2] Campbell, Andrew, Jason Yim, Regina Barzilay, Tom Rainforth, and Tommi Jaakkola. "Generative flows on discrete state-spaces: Enabling multimodal flows with applications to protein co-design." *ICML* (2024).

[3] Zheng, Kaiwen, Yongxin Chen, Hanzi Mao, Ming-Yu Liu, Jun Zhu, and Qinsheng Zhang. "Masked diffusion models are secretly time-agnostic masked models and exploit inaccurate categorical sampling." *arXiv preprint arXiv:2409.02908* (2024).

---

### Author Response · Authors · 2024-12-03
**Summary of author response and paper revision**

We sincerely thank all reviewers for their thoughtful and constructive feedback. As the author response period concludes, we provide a concise summary of our individual responses. Our paper revision includes additional results and explanations that address major concerns. Suggestions on improving presentation will be incorporated into the manuscript in subsequent updates, given current page limit.

Reviewer uHmQ:

- *Lack of test/validation NLL/perplexity*: Addressed in Common Responses C1.1 and C1.2. Also refer to Appendix C.1.
- *Method details for different configurations*: Expanded details in Appendix D.1.
- *Discussion on time interval adjustment in [1] and connection with MD4*: Detailed discussion provided in our response.

Reviewer yRLR:

- *FID score with CFG, comparisons with [2]*: Additional experiments and explanations on heuristic interplay are discussed and updated in Appendix E.4.1.
- *Clarification on single-dimension transition*: Addressed in our response.
- *If data latent states remain unchanged*: Further experiments and explanations confirm planner effectiveness in correcting (reverting) latent state if necessary; see Figure 8.
- *Test perplexity***:** Addressed in Common Response C1.1.

Reviewer 4zHc:

- *Performance with increased sampling steps*: Clarified in our response.
- *Hard-to-follow narrative*: Restructured and improved the introduction section for better readability.
- *Additional experiments in machine translation*: As our work focuses on standard benchmarks from prior discrete diffusion methods [3, 4, 5], we did not include machine translation experiments.

Reviewer p5AT:

- *Sampling steps fewer than sequence length*: Explanation and updated experiments provided in Figure 2.
- *Missing points in Figure 2*: Clarified in our response.
- *Convergence during sampling*: New experiments added in Figure 8.
- *Presentation suggestions*: Incorporated where feasible; further refinements planned due to page limit.

---

### Meta-Review · Area_Chair_849b · 2024-12-21

**Metareview:**

This paper proposes to train two networks for discrete diffusion: one for denoising, the other for identifying the positions to be denoised. The idea is valid and intuitive, and is clearly presented in the paper. All but one reviewer are in favor of accept, however after going through the reviews and rebuttal, I found that reviewer's 4zHc negative comments are not well supported -- eg there seems to be a serious misunderstanding regarding number of sample steps and that the request of machine translation tasks doesn't seem necessary. The rebuttal is very strong, and I believe it addresses most serious issues. I recommend acceptance based on these considerations.

**Additional Comments On Reviewer Discussion:**

The authors did a good job in addressing the concerns, mostly in evaluation protocols and some presentation clarity issues. Reviewer 4zHc did not participate in the discussion period but I went through the author responses and believed that they sufficiently addressed the questions.

---

### Decision · Program_Chairs · 2025-01-22

Accept (Poster)